# Provably Efficient UCB-type Algorithms For Learning Predictive State Representations

Ruiquan Huang[*]            Yingbin Liang[†]            Jing Yang[*]

## Abstract

The general sequential decision-making problem, which includes Markov decision processes (MDPs) and partially observable MDPs (POMDPs) as special cases, aims at maximizing a cumulative reward by making a sequence of decisions based on a history of observations and actions over time. Recent studies have shown that the sequential decision-making problem is statistically learnable if it admits a low-rank structure modeled by predictive state representations (PSRs). Despite these advancements, existing approaches typically involve oracles or steps that are computationally intractable. On the other hand, the upper confidence bound (UCB) based approaches, which have served successfully as computationally efficient methods in bandits and MDPs, have not been investigated for more general PSRs, due to the difficulty of optimistic bonus design in these more challenging settings. This paper proposes the first known UCB-type approach for PSRs, featuring a novel bonus term that upper bounds the total variation distance between the estimated and true models. We further characterize the sample complexity bounds for our designed UCB-type algorithms for both online and offline PSRs. In contrast to existing approaches for PSRs, our UCB-type algorithms enjoy computational tractability, last-iterate guaranteed near-optimal policy, and guaranteed model accuracy.

## 1 Introduction

As a general framework of reinforcement learning (RL), the sequential decision-making problem aims at maximizing a cumulative reward by making a sequence of decisions based on a history of observations and actions over time. This framework is powerful to include and generalize Markov Decision Processes (MDPs) and Partially Observable Markov Decision Processes (POMDPs), and captures a wide range of real-world applications such as recommender systems (Li et al., 2010; Wu et al., 2021), business management (De Brito & Van Der Laan, 2009), economic simulation (Zheng et al., 2020), robotics (Akkaya et al., 2019), strategic games (Brown & Sandholm, 2018; Vinyals et al., 2019), and medical diagnostic systems (Hauskrecht & Fraser, 2000).

However, tackling POMDPs alone presents significant challenges, not to mention general sequential decision-making problems. Many hardness results have been developed (Mossel & Roch, 2005; Mundhenk et al., 2000; Papadimitriou & Tsitsiklis, 1987; Vlassis et al., 2012; Krishnamurthy et al., 2016), showing that learning POMDPs is already computationally and statistically intractable in the worst case. The reason is that the non-Markovian property of these problems implies that the sufficient statistics or belief about the current environmental state encompasses all observations and actions from past interactions with the environment. This dramatically increases the computational burden and statistical complexity, since even for a finite observation-action space, the possibilities of the beliefs are exponentially large in terms of the number of observation-action pairs.

In order to tackle these challenges, recent research has introduced various structural conditions for POMDPs and general sequential decision-making problems, such as reactiveness (Jiang et al., 2017), decodability (Du et al., 2019), revealing conditions (Liu et al., 2022a), hindsight observability (Lee et al., 2023), and low-rank representations with regularization conditions (Zhan et al., 2022; Liu et al., 2022b; Chen et al., 2022). These conditions have opened up new possibilities for achieving

---

[*]Penn State University, State College, PA 16801, USA. {`rzh514,yangjing`}`@psu.edu`
[†]Ohio State University, Columbus, OH 43210, USA. `liang.889@osu.edu`

polynomial sample complexities in general sequential decision-making problems. Among them, predictive state representations (PSRs) (Littman & Sutton, 2001) has been proved to capture and generalize a rich subclass of sequential decision-making problems such as MDPs and observable POMDPs. Yet, most existing solutions for PSRs involve oracles that might not be computationally tractable. For instance, Optimistic MLE (OMLE) (Liu et al., 2022b) involves a step that maximizes the optimal value function over a confidence set of models. Typically, such a confidence set does not exhibit advantageous structures, resulting in a potentially combinatorial search within the set. Another popular posterior sampling based approach (Agarwal & Zhang, 2022; Zhong et al., 2022) requires to maintain a distribution over the entire set of models, which is highly memory inefficient. In addition, most existing results lack a last-iterate guarantee and produce only mixture policies, which often exhibit a very large variance in practical applications. On the other hand, the upper confidence bound (UCB) based approach has been proved to be computationally efficient and provide last-iterate guarantee in many decision-making problems such as bandits (Auer et al., 2002) and MDPs (Ménard et al., 2021). However, due to the non-Markovian property of POMDPs and general sequential decision-making problems, designing an explicit UCB is extremely challenging. To the best of our knowledge, such a design has been seldomly explored in POMDPs and beyond. Thus, we are motivated to address the following important open question:

*Q1: Can we design a UCB-type algorithm for learning PSRs that (a) is both computationally tractable and statistically efficient, and (b) enjoys the last-iterate guarantee?*

Another important research direction in RL is offline learning (Lange et al., 2012), where the learning agent has access to a pre-collected dataset and aims to design a favorable policy without any interaction with the environment. While offline MDPs have been extensively studied (Jin et al., 2021b; Xiong et al., 2022a; Xie et al., 2021), there exist very limited studies of offline POMDPs from the theoretical perspective (Guo et al., 2022; Lu et al., 2022). To our best knowledge, offline learning for a more general model of PSRs has never been explored. Thus, we will further address the following research question:

*Q2: Can we design a UCB-type algorithm for **offline** learning of PSRs with guaranteed policy performance and sample efficiency?*

**Main Contributions:** We provide affirmative answers to both aforementioned questions by making the following contributions.

- We introduce the first known UCB-type approach to learning PSRs with only a regularization assumption, characterized by a novel bonus term that upper bounds the total variation distance between the estimated and true models. The bonus term is designed based on a new confidence bound induced by a new model estimation guarantee for PSRs, and is computationally tractable.

- We theoretically characterize the performance of our UCB-type algorithm for learning PSRs online, called PSR-UCB. In contrast to existing approaches, PSR-UCB is computationally tractable with only supervised learning oracles, guarantees a near-optimal policy in the last iteration, and ensures model accuracy. When the rank of the PSR is small, our sample complexity matches the best known upper bound in terms of the rank and the accuracy level.

- We further extend our UCB-type approach to the offline setting, and propose the PSR-LCB algorithm. We then develop an upper bound on the performance difference between the output policy of PSR-LCB and any policy covered by the behavior policy. The performance difference scales in $O(C_\infty/\sqrt{K})$, where $C_\infty$ is the coverage coefficient and $K$ is the size of the offline dataset. This is the first known sample complexity result on offline PSRs.

- Technically, we develop two key properties for PSRs to establish the sample complexity guarantees: (a) a new estimation guarantee on the distribution of future observations conditioned on empirical samples, enabled by the *stable* model estimation step, and (b) a new relationship between the empirical UCB and the ground-truth UCB. We believe these insights advance the current understanding of PSRs, and will benefit future studies on this topic.

## 2 RELATED WORK

**Learning MDPs and POMDPs.** The MDP is a basic model in RL that assumes Markovian property in the model dynamics, i.e. the distribution of the future states only depends on the current system state. Researchers show that learning tabular MDPs (with finite state and action spaces) is

both computationally and statistically efficient in both online setting (Auer et al., 2008; Azar et al., 2017; Dann et al., 2017; Agrawal & Jia, 2017; Jin et al., 2018; Li et al., 2021) and offline setting (Jin et al., 2021b; Rashidinejad et al., 2021; Yin et al., 2021; Chang et al., 2021). Learning MDPs with function approximations is also well-studied by establishing favorable statistical complexity and computation efficiency (Jin et al., 2020; Wagenmaker & Jamieson, 2022; Zanette et al., 2020; Zhou et al., 2021; Agarwal et al., 2020; Uehara et al., 2021; Du et al., 2021; Foster et al., 2021; Jin et al., 2021a; Jiang et al., 2017; Wang et al., 2020; Jin et al., 2021b; Xiong et al., 2022a; Xie et al., 2021). Notably, several algorithms designed for learning MDPs with general function approximations can be extended to solve a subclass of POMDPs. In particular, OLIVE (Jiang et al., 2017) and GOLF (Jin et al., 2021a), which are originally designed for MDPs with low Bellman rank and low Bellman-Eluder dimension, respectively, can efficiently learn reactive POMDPs, where the optimal policy only depends on the current observation. Besides, Du et al. (2019); Efroni et al. (2022) study decodable RL where the observations determine the underlying states, and Kwon et al. (2021) investigate latent MDPs where there are multiple MDPs determined by some latent variables.

To directly address the partial observability in POMDPs, some works assume exploratory data or reachability property and provide polynomial sample complexity for learning these POMDPs (Guo et al., 2016; Azizzadenesheli et al., 2016; Xiong et al., 2022b). Others tackle the challenge of exploration and exploitation tradeoff in POMDPs by considering various sub-classes of POMDPs such as low-rank POMDPs (Wang et al., 2022), observable POMDPs (Golowich et al., 2022a;b), hindsight observability (Lee et al., 2023), and weakly-revealing POMDPs (Liu et al., 2022c;a). Furthermore, Liu et al. (2022c); Uehara et al. (2022a) propose computationally efficient algorithms for POMDPs with deterministic latent transitions. Notably, Golowich et al. (2022a) propose a provably efficient algorithm for learning observable tabular POMDPs without computationally intractable oracles. Finally, Lu et al. (2022) study the offline POMDPs in the presence of confounders, and Guo et al. (2022) provide provably efficient algorithm for offline linear POMDPs. After the initial submission of this work, we notice that a UCB-type algorithm has been studied by Guo et al. (2023) under low-rank $L$-step decodable POMDPs, which is a subclass of the PSRs considered here.

**Learning PSRs and general sequential decision-making problems.** The PSR is first introduced by Littman & Sutton (2001); Singh et al. (2012) and considered as a general representation to model dynamic systems. A line of research (Boots et al., 2011; Hefny et al., 2015; Jiang et al., 2018; Zhang et al., 2022) obtain polynomial sample complexity with observability assumption and spectral techniques. Later, Zhan et al. (2022) demonstrate that learning regular PSRs is sample efficient and can avoid poly$(|\mathcal{O}|^m)$ in the sample complexity. Uehara et al. (2022b) propose a PO-bilinear class that captures a rich class of tractable RL problems with partial observations, including weakly revealing POMDPs and PSRs and design an actor-critic style algorithm. For the works most closely related to ours, Liu et al. (2022b) propose a universal algorithm known as OMLE, which is capable of learning PSRs and its generalizations under certain conditions; Chen et al. (2022) enhance the sample complexity upper bounds for three distinct algorithms, including OMLE, the model-based posterior sampling, and the estimation-to-decision type algorithm (Foster et al., 2021); Zhong et al. (2022) address the general sequential decision-making problem by posterior sampling under a newly proposed low generalized Eluder coefficient. However, as elaborated in Section 1, those approaches are not efficient in terms of computational complexity or memory.

## 3 PRELIMINARIES

**Problem Setting.** We consider a finite horizon episodic sequential decision-making problem, defined by a tuple $\mathsf{P} = (\mathcal{O}, \mathcal{A}, H, \mathbb{P}, R)$, where $\mathcal{O}$ represents the observation space, $\mathcal{A}$ is a finite action space, $H$ is the number of time steps within an episode, $\mathbb{P} = \{\mathbb{P}_h\}$ determines the model dynamics, i.e., $\mathbb{P}_h(o_h|o_1, \ldots, o_{h-1}, a_1, \ldots, a_{h-1})$, where $o_t \in \mathcal{O}$ is the observation at time step $t$, and $a_t \in \mathcal{A}$ is the action taken by the agent at time step $t$ for all $t \in \{1, \ldots, h\}$, and $R : (\mathcal{O} \times \mathcal{A})^H \to [0, 1]$ is the reward function defined on trajectories of one episode. We denote a historical trajectory at time step $h$ as $\tau_h := (o_1, a_1, \ldots, o_h, a_h)$, and denote a future trajectory as $\omega_h := (o_{h+1}, a_{h+1}, \ldots, o_H, a_H)$. The set of all $\tau_h$ is denoted by $\mathcal{H}_h = (\mathcal{O} \times \mathcal{A})^h$ and the set of all future trajectories is denoted by $\Omega_h = (\mathcal{O} \times \mathcal{A})^{H-h}$. In addition, let $\omega_h^o = (o_{h+1}, \ldots, o_H)$ and $\omega_h^a = (a_{h+1}, \ldots, a_H)$ be the observation sequence and the action sequence contained in $\omega_h$, respectively. Similarly, for a history $\tau_h$, we denote $\tau_h^o$ and $\tau_h^a$ as the observation and action sequences in $\tau_h$, respectively. Notably, the general framework of sequential decision-making problem subsume not only fully observable

MDPs but also POMDPs as special cases, because in MDPs, $\mathbb{P}_h(o_h|\tau_{h-1}) = \mathbb{P}_h(o_h|o_{h-1}, a_{h-1})$ and in POMDPs, $\mathbb{P}_h(o_h|\tau_{h-1})$ can be factorized as $\mathbb{P}_h(o_h|\tau_{h-1}) = \sum_s \mathbb{P}_h(o_h|s)\mathbb{P}_h(s|\tau_{h-1})$, where $s$ represents unobserved states.

The interaction between an agent and P proceeds as follows. At the beginning of each episode, the environment initializes a fixed observation $o_1$ at time step 1. After observing $o_1$, the agent takes action $a_1$, and the environment transits to $o_2$, which is sampled according to the distribution $\mathbb{P}_1(o_2|o_1, a_1)$. Then, at any time step $h \geq 2$, due to the non-Markovian nature of the problem, the agent takes action $a_h$ based on all past information $(\tau_{h-1}, o_h)$, and the environment transits to $o_{h+1}$, sampled from $\mathbb{P}_h(o_{h+1}|\tau_h)$. The interaction terminates after time step $H$.

The policy $\pi = \{\pi_h\}$ of the agent is a collection of $H$ distributions where $\pi_h(a_h|\tau_{h-1}, o_h)$ is the probability of choosing action $a_h$ at time step $h$ given the history $\tau_{h-1}$ and the current observation $o_h$. For simplicity, we use $\pi(\tau_h) = \pi(a_h|o_h, \tau_{h-1}) \cdots \pi(a_1|o_1)$ to denote the probability of the sequence of actions $\tau_h^a$ given the observations $\tau_h^o$. We denote $\mathbb{P}^\pi$ as the distribution of trajectories induced by policy $\pi$ under dynamics $\mathbb{P}$. The value of a policy $\pi$ under $\mathbb{P}$ and the reward $R$ is denoted by $V_{\mathbb{P},R}^\pi = \mathbb{E}_{\tau_H \sim \mathbb{P}^\pi}[R(\tau_H)]$.

The goal of the agent is to find an $\epsilon$-optimal policy $\hat{\pi}$ that satisfies $\max_\pi V_{\mathbb{P},R}^\pi - V_{\mathbb{P},R}^{\hat{\pi}} \leq \epsilon$. Since finding a near-optimal policy for a general decision-making problem incurs exponentially large sample complexity in the worst case, in this paper we follow the line of research in Zhan et al. (2022); Chen et al. (2022); Zhong et al. (2022) and focus on the *low-rank* class of problems. To define a low-rank problem, we introduce the dynamic matrix $\mathbb{D}_h \in \mathbb{R}^{|\mathcal{H}_h| \times |\Omega_h|}$ for each $h$, where the entry at the $\tau_h$-th row and $\omega_h$-th column of $\mathbb{D}_h$ is $\mathbb{P}(\omega_h^o, \tau_h^o|\tau_h^a, \omega_h^a)$.

**Definition 1 (Rank-$r$ sequential decision-making problem)** *A sequential decision-making problem is rank $r$ if for any $h$, the model dynamic matrix $\mathbb{D}_h$ has rank $r$.*

**Predictive State Representation (PSR).** To exploit the low-rank structure, we assume that for each $h$, there exists a set of future trajectories, namely, core tests (known to the agent) $\mathcal{Q}_h = \{\mathbf{q}_h^1, \ldots, \mathbf{q}_h^{d_h}\} \subset \Omega_h$, such that the submatrix restricted to these tests $\mathbb{D}_h[\mathcal{Q}_h]$ has rank $r$, where $d_h \geq r$ is a positive integer. This special set $\mathcal{Q}_h$ allows the system dynamics to be factorized as $\mathbb{P}(\omega_h^o, \tau_h^o|\tau_h^a, \omega_h^a) = \mathbf{m}(\omega_h)^\top \psi(\tau_h)$, where $\mathbf{m}(\omega_h), \psi(\tau_h) \in \mathbb{R}^{d_h}$ and the $\ell$-th coordinate of $\psi(\tau_h)$ is the joint probability of $\tau_h$ and the $\ell$-th core test $\mathbf{q}_h^\ell$. Mathematically, if we use $\mathbf{o}_h^\ell$ and $\mathbf{a}_h^\ell$ to denote the observation sequence and the action sequence of $\mathbf{q}_h^\ell$, respectively, then $\mathbb{P}(\mathbf{o}_h^\ell, \tau_h^o|\tau_h^a, \mathbf{a}_h^\ell) = [\psi(\tau_h)]_\ell$. By Theorem C.1 in Liu et al. (2022b), any low-rank decision-making problem admits a (self-consistent) predictive state representation $\theta = \{\phi_h, \mathbf{M}_h\}_{h=1}^H$ given core tests $\{\mathcal{Q}_h\}_{h=0}^{H-1}$, such that for any $\tau_h \in \mathcal{H}_h, \omega_h \in \Omega_h$,

$$\psi(\tau_h) = \mathbf{M}_h(o_h, a_h) \cdots \mathbf{M}_1(o_1, a_1)\psi_0, \quad \mathbf{m}(\omega_h)^\top = \phi_H^\top \mathbf{M}_H(o_H, a_H) \cdots \mathbf{M}_{h+1}(o_{h+1}, a_{h+1})$$

$$\sum_{o_{h+1}} \phi_{h+1}^\top \mathbf{M}_{h+1}(o_{h+1}, a_{h+1}) = \phi_h^\top, \quad \mathbb{P}(o_h, \ldots, o_1|a_1, \ldots, a_h) = \phi_h^\top \psi(\tau_h),$$

where $\mathbf{M}_h : \mathcal{O} \times \mathcal{A} \to \mathbb{R}^{d_h \times d_{h-1}}$, $\phi_h \in \mathbb{R}^{d_h}$, and $\psi_0 \in \mathbb{R}^{d_0}$. For ease of presentation, we assume $\psi_0$ is known to the agent[1]. Notably, the normalized version of $\psi(\tau_h)$ with respect to $\phi_h^\top \psi(\tau_h)$, denoted as $\bar{\psi}(\tau_h) = \psi(\tau_h)/\phi_h^\top \psi(\tau_h)$, is known as the prediction vector (Littman & Sutton, 2001) or prediction feature of $\tau_h$, since $[\bar{\psi}(\tau_h)]_\ell = \mathbb{P}(\mathbf{o}_h^\ell|\tau_h, \mathbf{a}_h^\ell)$. As illustrated in Appendices C and D, the prediction feature plays an important role in our algorithm design.

In the following context, we use $\mathbb{P}_\theta$ to indicate the model determined by the representation $\theta = \{\phi_h, \mathbf{M}_h\}_{h=1}^H$. For simplicity, we denote $V_{\mathbb{P}_\theta, R}^\pi = V_{\theta, R}^\pi$. Moreover, let $\mathcal{Q}_h^A = \{\mathbf{a}_h^\ell\}_{\ell=1}^{d_h}$ be the set of action sequences that are part of core tests, constructed by eliminating any repeated action sequence. $\mathcal{Q}_h^A$, known as the set of core action sequences, plays a crucial role during the exploration process. Selecting from these sequences is sufficient to sample core tests, leading to accurate estimate of $\theta$.

We further assume that the PSRs studied in this paper are well-conditioned, as specified in Assumption 1. Such an assumption and its variants are commonly adopted in the study of PSRs (Liu et al., 2022b; Chen et al., 2022; Zhong et al., 2022).

---

[1]The sample complexity of learning $\psi_0$ if it is unknown is relatively small compared with learning other parameters.

**Assumption 1 ($\gamma$-well-conditioned PSR)** *A PSR $\theta$ is said to be $\gamma$-well-conditioned if*

$$\forall h, \quad \max_{x \in \mathbb{R}^{d_h}: \|x\|_1 \leq 1} \max_{\pi} \max_{\tau_h} \sum_{\omega_h} \pi(\omega_h|\tau_h)|\mathbf{m}(\omega_h)^\top x| \leq \frac{1}{\gamma}. \tag{1}$$

Assumption 1 requires that the error of estimating $\theta$ does not significantly blow up when the estimation error $x$ of estimating the probability of core tests is small.

**Notations.** We denote the complete set of model parameters as $\Theta$ and the true model parameter as $\theta^*$. For a vector $x$, $\|x\|_A$ stands for $\sqrt{x^\top A x}$, and the $i$-th coordinate of $x$ is represented as $[x]_i$. For functions $\mathbb{P}$ and $\mathbb{Q}$ (not necessarily probability measures) over a set $\mathcal{X}$, the total variation distance between them is $\mathtt{D}_{\mathrm{TV}}(\mathbb{P}(x), \mathbb{Q}(x)) = \sum_x |\mathbb{P}(x) - \mathbb{Q}(x)|$, while the hellinger-squared distance is defined as $\mathtt{D}_{\mathrm{H}}^2(\mathbb{P}(x), \mathbb{Q}(x)) = \frac{1}{2}\sum_x (\sqrt{\mathbb{P}(x)} - \sqrt{\mathbb{Q}(x)})^2$. Note that our definition of the total variation distance is slightly different from convention by a constant factor. We define $d = \max_h d_h$ and $Q_A = \max_h |\mathcal{Q}_h^A|$. We use $\nu_h(\pi, \pi')$ to denote the policy that takes $\pi$ at the initial $h-1$ steps and switches to $\pi'$ from the $h$-th step. Lastly, $\mathtt{u}_{\mathcal{X}}$ represents the uniform distribution over the set $\mathcal{X}$.

## 4 Online Learning for Predivtive State Representations

In this section, we propose a model-based algorithm PSR-UCB, which features three main novel designs: (a) a stable model estimation step controlling the quality of the estimated model such that its prediction features of the empirical data are useful in the design of UCB, (b) an upper confidence bound that captures the uncertainty of the estimated model, and (c) a termination condition that guarantees the last-iterate model is near-accurate and the corresponding greedy policy is near-optimal.

### 4.1 Algorithm

The pseudo-code for the PSR-UCB algorithm is presented in Algorithm 1. We highlight the key idea of PSR-UCB: in contrast to the design of UCB-type algorithms in MDPs that leverage low-rank structures, we exploit the physical meaning of the prediction feature $\bar{\psi}(\tau_h)$ to design bonus terms that enable efficient exploration. In particular, "physical meaning" refers to that each coordinate of a prediction feature of $\tau_h$ represents the probability of visiting a core test conditioned on $\tau_h$ and taking the corresponding core action sequence. Therefore, it suffices to explore a set of $\tau_h$ whose prediction features can span the entire feature space, and use core action sequences to learn those "base" features. We next elaborate the main steps of the algorithm in greater detail as follows.

**Exploration.** At each iteration $k$, PSR-UCB constructs a greedy policy $\pi^{k-1}$ based on a previous dataset $\mathcal{D}^{k-1} = \{\mathcal{D}_h^{k-1}\}_{h=0}^{H-1}$, together with an estimated model $\hat{\theta}^{k-1}$. Intuitively, to enable efficient exploration, $\pi^{k-1}$ is expected to sample $\tau_h$ that "differs" the most from previous collected samples $\tau_h \in \mathcal{D}_h^{k-1}$. How to quantify such differences forms the foundation of our algorithm design and will be elaborated later.

Then, for each $h \in [H]$, PSR-UCB first uses $\pi^{k-1}$ to get a sample $\tau_{h-1}^{k,h}$, then follows the policy $\mathtt{u}_{\mathcal{Q}_{h-1}^{\exp}}$ to get $\omega_{h-1}^{k,h}$, where $\mathcal{Q}_{h-1}^{\exp} = (\mathcal{A} \times \mathcal{Q}_h^A) \cup \mathcal{Q}_{h-1}^A$. Here, superscripts $k, h$ represent the index of episodes. In other words, PSR-UCB adopts the policy $\nu_h(\pi^{k-1}, \mathtt{u}_{\mathcal{Q}_{h-1}^{\exp}})$ to collect a sample trajectory $(\tau_{h-1}^{k,h}, \omega_{h-1}^{k,h})$, which, together with $\mathcal{D}_{h-1}^{k-1}$, forms the dataset $\mathcal{D}_{h-1}^k$.

The importance of uniformly selecting actions from $\mathcal{Q}_{h-1}^{\exp}$ can be explained as follows. First, the actions in $\mathcal{Q}_{h-1}^A$ assist us to learn the prediction feature $\bar{\psi}^*(\tau_{h-1}^{k,h})$ as its $\ell$-th coordinate equals to $\mathbb{P}_{\theta^*}(\mathbf{o}_{h-1}^\ell|\tau_{h-1}^{k,h}, \mathbf{a}_{h-1}^\ell)$. Second, the action sequence $(a_h, \mathbf{a}_h^\ell) \in \mathcal{A} \times \mathcal{Q}_h^A$ helps us to estimate $\mathbf{M}_h^*(o_h, a_h)\bar{\psi}^*(\tau_{h-1}^{k,h})$ because its $\ell$-th coordinate represents $\mathbb{P}_{\theta^*}(\mathbf{o}_h^\ell, o_h|\tau_{h-1}^{k,h}, a_h, \mathbf{a}_h^\ell)$. Therefore, by uniform exploration on $\mathcal{Q}_{h-1}^{\exp}$ given that $\tau_{h-1}^{k,h}$ differs from previous dataset $\mathcal{D}_{h-1}^{k-1}$, PSR-UCB collects the most informative samples for estimating the true model $\theta^*$.

**Stable model estimation.** With the updated dataset $\mathcal{D}^k = \{\mathcal{D}_h^k\}$, we estimate the model by maximizing the log-likelihood functions with constraints. Specifically, PSR-UCB extracts any model $\hat{\theta}^k$

from $\mathcal{B}^k$ defined as:

$$\Theta_{\min}^k = \left\{ \theta : \forall h, (\tau_h, \pi) \in \mathcal{D}_h^k, \ \mathbb{P}_\theta^\pi(\tau_h) \geq p_{\min} \right\},$$

$$\mathcal{B}^k = \left\{ \theta \in \Theta_{\min}^k : \sum_{(\tau_H, \pi) \in \mathcal{D}^k} \log \mathbb{P}_\theta^\pi(\tau_H) \geq \max_{\theta' \in \Theta_{\min}^k} \sum_{(\tau_H, \pi) \in \mathcal{D}^k} \log \mathbb{P}_{\theta'}^\pi(\tau_H) - \beta \right\}, \quad (2)$$

where the estimation margin $\beta$ is a pre-defined constant. Here the threshold probability $p_{\min}$ is sufficiently small to guarantee that, with high probability, $\theta^* \in \Theta_{\min}^k$. Note that compared to existing vanilla maximum likelihood estimators (Liu et al., 2022b;a; Chen et al., 2022), PSR-UCB has an additional constraint $\Theta_{\min}^k$. This is a crucial condition, as $\tau_h^{k,h+1}$ is sampled to infer the conditional probability $\mathbb{P}_{\theta^*}(\omega_h^o | \tau_h^{k,h+1}, \omega_h^a)$ or the prediction feature $\bar{\psi}^*(\tau_h^{k,h+1})$, and learning this feature is useless or even harmful if $\mathbb{P}_{\hat{\theta}^k}^{\pi^{k-1}}(\tau_h^{k,h+1})$ is too small, as it is proved that $\theta^* \in \Theta_{\min}^k$ (Proposition 7).

**Design of UCB with prediction features.** From the discussion of the previous two steps, we see that the prediction features are vital since (a) actions in $\mathcal{Q}_h^{\exp}$ can efficiently explore the coordinates of these features, and (b) constraint $\Theta_{\min}^k$ ensures the significance of the learned features of the collected samples. Therefore, in the next round of exploration, our objective is to sample $\tau_h$ whose prediction feature exhibits the greatest dissimilarity compared with those of the previously collected samples $\tau_h \in \mathcal{D}_h^k$. Towards that, PSR-UCB constructs an upper confidence bound $V_{\hat{\theta}^k, \hat{b}^k}^\pi$ for $\mathsf{D}_{\mathrm{TV}}(\mathbb{P}_{\hat{\theta}^k}^\pi(\tau_H), \mathbb{P}_\theta^\pi(\tau_H))$, where the bonus $\hat{b}^k$ is defined as:

$$\hat{U}_h^k = \lambda I + \sum_{\tau_h' \in \mathcal{D}_h^k} \bar{\hat{\psi}}^k(\tau_h') \bar{\hat{\psi}}^k(\tau_h')^\top, \quad \hat{b}^k(\tau_H) = \min \left\{ \alpha \sqrt{\sum_{h=0}^{H-1} \left\| \bar{\hat{\psi}}^k(\tau_h) \right\|_{(\hat{U}_h^k)^{-1}}^2}, 1 \right\}, \quad (3)$$

with pre-specified regularizer $\lambda$ and UCB coefficient $\alpha$. Note that large $\hat{b}^k(\tau_H)$ indicates that $\tau_H$ is "perceived" to be under explored since the *estimated* prediction feature $\bar{\hat{\psi}}^k(\tau_h)$ is significantly different from $\{\bar{\hat{\psi}}^k(\tau_h)\}_{\tau_h \in \mathcal{D}_h^k}$.

---

**Algorithm 1** Learning Predictive State Representation with Upper Confidence Bound (PSR-UCB)

---

1: **Input:** threshold probability $p_{\min}$, estimation margin $\beta$, regularizer $\lambda$, UCB coefficient $\alpha$.
2: **for** $k = 1, \ldots, K$ **do**
3:      **for** $h = 1, ..., H$ **do**
4:          Use $\nu(\pi^{k-1}, \mathbf{u}_{\mathcal{Q}_{h-1}^{\exp}})$ to collect data $\tau_H^{k,h} = (\omega_{h-1}^{k,h}, \tau_{h-1}^{k,h})$.
5:          $\mathcal{D}_{h-1}^k \leftarrow \mathcal{D}_{h-1}^{k-1} \cup \{(\tau_H^{k,h}, \nu_h(\pi^{k-1}, \mathbf{u}_{\mathcal{Q}_{h-1}^{\exp}}))\}$.
6:      **end for**
7:      $\mathcal{D}^k = \{\mathcal{D}_h^k\}_{h=0}^{H-1}$. Extract any $\hat{\theta}^k \in \mathcal{B}^k$ according to Equation (2).
8:      Define bonus function $\hat{b}^k(\tau_H)$ according to Equation (3).
9:      Solve $\pi^k = \arg\max_\pi V_{\hat{\theta}^k, \hat{b}^k}^\pi$.
10:      **if** $V_{\hat{\theta}^k, \hat{b}^k}^{\pi^k} \leq \epsilon/2$ **then**
11:          $\theta^\epsilon = \hat{\theta}^k$, **break.**
12:      **end if**
13: **end for**
14: **Output:** $\bar{\pi} = \arg\max_\pi V_{\theta^\epsilon, R}^\pi$.

---

**Design of the greedy policy and last-iterate guaranteed termination.** The construction of UCB implies that $\hat{\theta}^k$ is highly uncertain on the trajectories $\tau_H$ with large bonuses. Thus, PSR-UCB finds a greedy policy $\pi^k = \arg\max_\pi V_{\hat{\theta}^k, \hat{b}^k}^\pi$ and terminates if $V_{\hat{\theta}^k, \hat{b}^k}^\pi$ is sufficiently small, indicating that the estimated model $\hat{\theta}^k$ is sufficiently accurate on any trajectory. Otherwise, $\pi^k$ serves for the next iteration $k+1$, as it tries to sample the most dissimilar $\bar{\hat{\psi}}^k(\tau_h)$ compared with the previous samples $\bar{\hat{\psi}}^k(\tau_h^{k,h+1})$ for efficient exploration. We remark that the termination condition favors the last-iterate guarantee, where a single model and a greedy policy are identified. Compared with algorithms that output a mixture policy (e.g. uniform selection in a large policy set), this guarantee may present lower variance in practical applications.

**Remark 1 (Computation)** *Our algorithm only calls the MLE oracle $H$ times in each episode to construct the confidence model class in Equation (2). Similar to Guo et al. (2023), our UCB based planning avoids searching for an optimistic policy that maximizes the optimal value over a large model class, thus is more amendable to practical implementation.*

**Remark 2 (Reward-free PSRs)** *Our algorithm naturally handles reward-free PSRs since during the exploration, no reward information is needed. In the reward-free setting, Algorithm 1 exhibits greater advantage compared with existing reward-free algorithms (Liu et al., 2022b; Chen et al., 2022) for PSRs, which require a potentially combinatorial optimization oracle over* a pair *of models in the (non-convex) confidence set and examination of the total variance distance[2], and are computationally intractable in general (Golowich et al., 2022a).*

## 4.2 THEORETICAL RESULTS

In this section, we present the theoretical results for PSR-UCB. To make a general statement, we first introduce the notion of the optimistic net of the parameter space.

**Definition 2 (Optimistic net)** *Consider two bounded functions $\mathbb{P}$ and $\bar{\mathbb{P}}$ over a set $\mathcal{X}$. Then, $\bar{\mathbb{P}}$ is $\varepsilon$-optimistic over $\mathbb{P}$ if (a) $\bar{\mathbb{P}}(x) \geq \mathbb{P}(x), \forall x \in \mathcal{X}$, and (b) $D_{TV}(\mathbb{P}, \bar{\mathbb{P}}) \leq \varepsilon$. The $\varepsilon$-optimistic net of $\Theta$ is a smallest finite space $\bar{\Theta}_\varepsilon$ so that for all $\theta \in \Theta$, there exists $\bar{\theta} \in \bar{\Theta}_\varepsilon$ when $\mathbb{P}_{\bar{\theta}}$ is $\varepsilon$-optimistic over $\mathbb{P}_\theta$.*

Note that if $\Theta$ is the parameter space for tabular PSRs (including finite observation and action spaces) with rank $r$, we have $|\bar{\Theta}_\varepsilon| \leq r^2|\mathcal{O}||\mathcal{A}|H^2 \log \frac{H|\mathcal{O}||\mathcal{A}|}{\varepsilon}$ (see Proposition 3 or Theorem 4.7 in Liu et al. (2022b)). Now, we are ready to present the main theorem for PSR-UCB.

**Theorem 1** *Suppose Assumption 1 holds. Let $p_{\min} = O(\frac{\delta}{KH|\mathcal{O}|^H|\mathcal{A}|^H})^3$, $\beta = O(\log|\bar{\Theta}_\varepsilon|)$, where $\varepsilon = O(\frac{p_{\min}}{KH})$, $\lambda = \frac{\gamma|\mathcal{A}|^2 Q_A \beta \max\{\sqrt{r}, Q_A\sqrt{H}/\gamma\}}{\sqrt{dH}}$, and $\alpha = O\left(\frac{Q_A\sqrt{Hd}}{\gamma^2}\sqrt{\lambda} + \frac{|\mathcal{A}|Q_A\sqrt{\beta}}{\gamma}\right)$. Then, with probability at least $1 - \delta$, PSR-UCB outputs a model $\theta^\epsilon$ and a policy $\bar{\pi}$ that satisfy*

$$V_{\theta^*, R}^{\pi^*} - V_{\theta^*, R}^{\bar{\pi}} \leq \epsilon, \text{ and } \forall \pi, \ D_{TV}\left(\mathbb{P}_{\theta^\epsilon}^\pi(\tau_H), \mathbb{P}_{\theta^*}^\pi(\tau_H)\right) \leq \epsilon. \tag{4}$$

*In addition, PSR-UCB terminates with a sample complexity of*

$$\tilde{O}\left(\left(r + \frac{Q_A^2 H}{\gamma^2}\right)\frac{rdH^3|\mathcal{A}|^2 Q_A^4 \beta}{\gamma^4 \epsilon^2}\right).$$

The following remarks highlight a few insights conveyed by Theorem 1. **First,** Theorem 1 explicitly states that PSR-UCB features last-iterate guarantee, i.e., the guaranteed performance is on the last output of the algorithm. This is in contrast to the previous studies (Liu et al., 2022a;b; Chen et al., 2022) on POMDP and/or PSRs, where the performance guarantee is on a mixture of policies obtained over the entire execution of algorithms. Such policies often have a large variance. **Second,** in the regime with a low PSR rank such that $r < \frac{Q_A^2 H}{\gamma^2}$, our result matches the best known sample complexity (Chen et al., 2022) in terms of rank $r$ and $\epsilon$. **Third,** thanks to the explicitly constructed bonus function $\hat{b}$, whose computation complexity is polynomial in terms of the iteration number and the dimension $d$, PSR-UCB is computationally tractable, provided that there exist oracles for planning in POMDPs (e.g. Line 9 in PSR-UCB) (Papadimitriou & Tsitsiklis, 1987; Guo et al., 2023) and maximum likelihood estimation.

*Proof Sketch of Theorem 1.* The proof relies on three main steps corresponding to three new technical developments, respectively. **(a) A new MLE guarantee.** Due to the novel *stable* model estimation design, we establish a new MLE guarantee (which has not been developed in the previous studies) that the total variation distance between the conditional distributions of future trajectories conditioned on $(\tau_h, \pi) \in \mathcal{D}_h^k$ is small. Mathematically, $\sum_{(\tau_h, \pi) \in \mathcal{D}_h^k} D_{TV}(\mathbb{P}_{\hat{\theta}^k}^\pi(\omega_h|\tau_h), \mathbb{P}_{\theta^*}^\pi(\omega_h|\tau_h))$

---

[2]Computation complexity of calculating total variation distance is still an open question (Bhattacharyya et al., 2022) and only some specific forms can be approximated in polynomial time.

[3]$|\mathcal{O}|$ can be the cardinality of $\mathcal{O}$ if it is finite, or the measure of $\mathcal{O}$ if it is a measurable set with positive and bounded measure.

is upper bounded by a constant. **(b) A new confidence bound.** Based on (a) and the observation that the estimation error of the prediction feature $\hat{\bar{\psi}}^k(\tau_h)$ is upper bounded by the total variation distance $D_{TV}(\mathbb{P}^\pi_{\hat{\theta}^k}(\omega_h|\tau_h), \mathbb{P}^\pi_{\theta^*}(\omega_h|\tau_h))$, the estimation error of other prediction features is captured by the function $\|\hat{\bar{\psi}}^k(\tau_h)\|_{(\hat{U}^k_h)^{-1}}$ up to some constant, leading to the valid UCB design. **(c) A new relationship between the empirical bonus and the ground-truth bonus.** To characterize the sample complexity, we need to show that $V^{\pi^k}_{\hat{\theta}^k,\hat{b}^k}$ can be small for some $k$. One approach is to prove that $\sum_k V^{\pi^k}_{\hat{\theta}^k,\hat{b}^k}$ is sub-linear. We validate this sub-linearity by establishing $\hat{b}^k \le O\left(\sum_h \|\bar{\psi}^*(\tau_h)\|_{(U^k_h)^{-1}}\right)$, where $U^k_h = \lambda I + \sum_{\tau_h \in \mathcal{D}^k_h} \bar{\psi}^*(\tau_h)\bar{\psi}^*(\tau_h)^\top$. This inequality bridges the empirical and the ground-truth bonuses, and yields the final result when combined with the elliptical potential lemma (Carpentier et al., 2020). The complete proof can be found in Appendix C.

When PSR is specialized to $m$-step decodable POMDPs (Liu et al., 2022b), as elaborated in Appendix C.5, the sample complexity of PSR-UCB does not depend on $d$ with slight modification of the algorithm.

**Corollary 1** *When PSR is specialized to $m$-step decodable POMDPs, for each $k$, there exists a set of matrices $\{\hat{\mathbf{G}}^k_h\}^H_{h=1} \subset \mathbb{R}^{d\times r}$ such that if we replace $\hat{\bar{\psi}}^k(\tau_h)$ by $(\hat{\mathbf{G}}^k_{h+1})^\dagger \hat{\bar{\psi}}^k(\tau_h)$ in Equation* (3) *for any $\tau_h$, then PSR-UCB terminates with a sample complexity of poly$(r, 1/\gamma, Q_A, H, |\mathcal{A}|, \beta)/\epsilon^2$.*

# 5 OFFLINE LEARNING FOR PREDICTIVE STATE REPRESENTATIONS

In this section, we develop a computationally tractable algorithm PSR-LCB to learn PSRs in the offline setting. In offline PSRs, a dataset $\mathcal{D}$ contains $K$ pre-collected trajectories that are independently sampled from a behavior policy $\pi^b$. We slightly generalize the learning goal to be finding a policy that can compete with any target policy whose coverage coefficient is finite. We highlight that PSR-LCB is the first offline algorithm for learning PSRs. When specialized to POMDPs, our proposed algorithm enjoys better computational complexity than confidence region based algorithms (Guo et al., 2022).

## 5.1 ALGORITHM

We design a PSR-LCB algorithm for offline PSRs, whose pseudo-code is presented in Algorithm 2. In the offline setting, we also leverage the prediction feature $\bar{\psi}(\tau_h)$ to design lower confidence bound (LCB) of the true value function $V^\pi_{\theta^*,r}$. We explain the main steps of the algorithm in detail as follows.

**stable model estimation.** Inspired by PSR-UCB, where the prediction features $\bar{\psi}(\tau_h)$ at each step $h$ are learned through separate datasets, PSR-LCB first randomly and evenly divides $\mathcal{D}$ into $H$ datasets $\mathcal{D}_0, \ldots, \mathcal{D}_{H-1}$. The goal of this division is to separately learn $\bar{\psi}^*(\tau_h)$ for each $h \in \{0, 1, \ldots, H-1\}$. Then, PSR-LCB extracts a model $\hat{\theta}$ from $\mathcal{B}_\mathcal{D}$ defined as:

$$\Theta_{\min} = \left\{\theta : \forall h, \tau_h \in \mathcal{D}_h, \ \mathbb{P}^{\pi^b}_\theta(\tau_h) \ge p_{\min}\right\},$$

$$\mathcal{B}_\mathcal{D} = \left\{\theta \in \Theta_{\min} : \sum_{\tau_H \in \mathcal{D}} \log \mathbb{P}^{\pi^b}_\theta(\tau_H) \ge \max_{\theta' \in \Theta_{\min}} \sum_{\tau_H \in \mathcal{D}} \log \mathbb{P}^{\pi^b}_{\theta'}(\tau_H) - \hat{\beta}\right\}, \qquad (5)$$

where the estimation margin $\hat{\beta}$ is a pre-specified constant, and $p_{\min}$ guarantees that, with high probability, $\theta^* \in \Theta_{\min}$.

Following similar reasons of the design of PSR-UCB, the constraint $\Theta_{\min}$ controls the quality of the estimated model such that the prediction features of the behavior samples are non-negligible and useful for learning.

**Design of LCB with prediction features.** Given the estimated model $\hat{\theta}$, PSR-LCB constructs an upper confidence bound of $D_{\text{TV}}\left(\mathbb{P}_{\hat{\theta}}^{\pi}(\tau_H), \mathbb{P}_{\theta}^{\pi}(\tau_H)\right)$ in the form of $V_{\hat{\theta}, \hat{b}}^{\pi}$, where $\hat{b}(\tau_H)$ is defined as:

$$\hat{U}_h = \hat{\lambda} I + \sum_{\tau_h \in \mathcal{D}_h} \bar{\bar{\psi}}(\tau_h) \bar{\bar{\psi}}(\tau_h)^\top, \quad \hat{b}(\tau_H) = \min\left\{\hat{\alpha}\sqrt{\sum_h \left\|\bar{\bar{\psi}}(\tau_h)\right\|_{(\hat{U}_h)^{-1}}^2}, 1\right\}, \tag{6}$$

with pre-defined regularizer $\hat{\lambda}$ and LCB coefficient $\hat{\alpha}$. Note that we refer to $\hat{\alpha}$ as the LCB coefficient, as we adopt the pessimism principle in offline learning. Differently from PSR-UCB, we aim to select a policy where the estimated model exhibits the least uncertainty. Therefore, the output policy should allocate a high probability to $\tau_H$ if $\hat{b}(\tau_H)$ is small.

**Output policy design.** Building on the discussion above, PSR-LCB outputs a policy $\bar{\pi} = \arg\max_\pi V_{\hat{\theta}, R}^{\pi} - V_{\hat{\theta}, \hat{b}}^{\pi}$, which maximizes a *lower confidence bound* of $V_{\theta^*, R}^{\pi}$.

---

**Algorithm 2** Offline Predictive State Representations with Lower Confidence Bound (PSR-LCB)

1: **Input:** Offline dataset $\mathcal{D}$, threshold probability $p_{\min}$, estimation margin $\hat{\beta}$, regularizer $\hat{\lambda}$, LCB coefficient $\hat{\alpha}$.
2: Estimate model $\hat{\theta} \in \mathcal{B}$ according to Equation (5).
3: Construct $\hat{b}$ according to Equation (6).
4: Output $\hat{\pi} = \arg\max_\pi V_{\hat{\theta}, R}^{\pi} - V_{\hat{\theta}, \hat{b}}^{\pi}$.

---

### 5.2 THEORETICAL RESULTS

In this section, we develop the theoretical guarantee for PSR-LCB. To capture the distribution shift in offline PSRs, we require that the $\ell_\infty$ norm of the ratio between the probabilities over the entire trajectory under the target policy and under the behavior policy is finite. Mathematically, if $\pi$ is the target policy, then, $C_{\pi^b, \infty}^{\pi} := \max_h \max_{\tau_h} \frac{\mathbb{P}_{\theta^*}^{\pi}(\tau_h)}{\mathbb{P}_{\theta^*}^{\pi^b}(\tau_h)} < \infty$. We note that, for general sequential decision-making problems, no Markovian property or hidden state is assumed. Therefore, defining the coverage assumption on a single observation-action pair as in Xie et al. (2021) for offline MDP, or on hidden states as in Guo et al. (2022) for offline POMDP, is not suitable for PSRs.

**Theorem 2** *Suppose Assumption 1 holds. Let* $\iota = \min_{\mathbf{a}_h \in \mathcal{Q}_h^{\exp}} \pi^b(\mathbf{a}_h)$, $p_{\min} = O(\frac{\delta}{KH(|\mathcal{O}||\mathcal{A}|)^H})$, $\varepsilon = O(\frac{p_{\min}}{KH})$, $\hat{\beta} = O(\log|\bar{\Theta}_\varepsilon|)$, $\hat{\lambda} = \frac{\gamma C_{\pi^b, \infty}^{\pi} \hat{\beta} \max\{\sqrt{r}, Q_A \sqrt{H}/\gamma\}}{\iota^2 Q_A \sqrt{dH}}$, *and* $\hat{\alpha} = O\left(\frac{Q_A \sqrt{dH}}{\gamma^2}\sqrt{\hat{\lambda}} + \frac{\sqrt{\beta}}{\iota\gamma}\right)$. *Then, with probability at least* $1 - \delta$, *the output* $\bar{\pi}$ *of Algorithm 2 satisfies that*

$$\forall \pi, \ V_{\theta^*, R}^{\pi} - V_{\theta^*, R}^{\bar{\pi}} \leq \tilde{O}\left(\left(\sqrt{r} + \frac{Q_A\sqrt{H}}{\gamma}\right)\frac{C_{\pi^b, \infty}^{\pi} Q_A H^2 \sqrt{d}}{\iota\gamma^2}\sqrt{\frac{r\hat{\beta}}{K}}\right). \tag{7}$$

Theorem 2 states that for any target policy with finite coverage coefficient $C_{\pi^b, \infty}^{\pi}$, the performance degradation of the output policy $\bar{\pi}$ with respect to the target policy $\pi$ is at most $\tilde{O}\left(\frac{C_{\pi^b, \infty}^{\pi}}{\sqrt{K}}\right)$, which is negligible if the size of the offline dataset $K$ is sufficiently large. Thee full proof is in Appendix D.

## 6 CONCLUSION

We studied learning predictive state representations (PSRs) for low-rank sequential decision-making problems. We developed a novel upper confidence bound for the total variation distance of the estimated model and the true model that enables both computationally tractable with MLE oracles and statistically efficient learning for PSRs with only supervised learning oracles. Specifically, we proposed PSR-UCB for online learning for PSRs with last-iterate guarantee, i.e. producing not only a near-optimal policy, but also a near-accurate model. The statistical efficiency was validated by a polynomial sample complexity in terms of the model parameters. In addition, we extended this UCB-type approach to the offline setting and proposed PSR-LCB. Our theoretical result offers an initial perspective on offline PSRs by demonstrating that PSR-LCB outputs a policy that can compete with any policy with finite coverage coefficient.

ACKNOWLEDGMENTS

The work of R. Huang and J. Yang was supported in part by the U.S. National Science Foundation under the grants CNS-1956276, CNS-2003131 and CNS-2030026. The work of Y. Liang was supported in part by the U.S. National Science Foundation under the grants RINGS-2148253, CCF-1900145, and CNS-2112471.

IMPACT STATEMENT

This paper presents work whose goal is to advance the field of Machine Learning. There are many potential societal consequences of our work, none which we feel must be specifically highlighted here.

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

# Provably Efficient UCB-type Algorithms For Learning PSRs: Supplementary Materials

## A    PROPERTIES OF PSRS

In this section, we present a few important properties of PSRs, which will be intensively used in the algorithm analysis.

First, for any model $\theta = \{\phi_h, \mathbf{M}_h(o_h, a_h)\}$, we have the following identity

$$\mathbf{M}_h(o_h, a_h)\bar{\psi}(\tau_{h-1}) = \mathbb{P}_\theta(o_h|\tau_{h-1})\bar{\psi}(\tau_h). \tag{8}$$

The following proposition is directly adapted from Lemma C.3 in Liu et al. (2022b). Note that $\psi_0$ is known to the agent.

**Proposition 1 (TV-distance ≤ Estimation error)** *Consider two $\gamma$-well-conditioned PSRs $\theta, \hat{\theta} \in \Theta$. We have*

$$\mathrm{D_{TV}}\left(\mathbb{P}_{\hat{\theta}}^\pi, \mathbb{P}_\theta^\pi\right) \leq \sum_{h=1}^H \sum_{\tau_H} \left|\hat{\mathbf{m}}(\omega_h)^\top \left(\hat{\mathbf{M}}(o_h, a_h) - \mathbf{M}(o_h, a_h)\right) \psi(\tau_{h-1})\right| \pi(\tau_H),$$

$$\mathrm{D_{TV}}\left(\mathbb{P}_{\hat{\theta}}^\pi, \mathbb{P}_\theta^\pi\right) \leq \sum_{h=1}^H \sum_{\tau_H} \left|\mathbf{m}(\omega_h)^\top \left(\hat{\mathbf{M}}(o_h, a_h) - \mathbf{M}(o_h, a_h)\right) \hat{\psi}(\tau_{h-1})\right| \pi(\tau_H).$$

The next proposition characterizes well-conditioned PSRs (Zhong et al., 2022), which is obtained by noting that $\mathbf{m}(\mathbf{q}_h^\ell)^\top$ is the $\ell$-th row of $\mathbf{M}_{h+1}(o_{h+1}, a_{h+1})$.

**Proposition 2** *For well-conditioned (self-consistent) PSR $\theta$, we have*

$$\max_{x \in \mathbb{R}^{d_{h-1}}:\|x\|_1=1} \max_\pi \sum_{\omega_h} \pi(\omega_h) \|\mathbf{M}_{h+1}(o_{h+1}, a_{h+1})x\|_1 \leq \frac{Q_A}{\gamma}.$$

The following proposition characterizes the log-cardinality of the minimal optimistic net of the rank-$r$ PSRs. We note that this notion is closely related to the bracketing number, a typical complexity measure in the maximum likelihood estimation (MLE) anslysis (Geer, 2000). The proof follows directly from Theorem C.9 in Liu et al. (2022b).

**Proposition 3 (Optimistic net for tabular PSRs)** *Given any $\epsilon$, there exists a finite parameter space $\bar{\Theta}_\epsilon$ satisfying the following property: for any $\theta \in \Theta$, we can find a $\bar{\theta} \in \bar{\Theta}_\epsilon$ associated with a measure $\mathbb{P}_{\bar{\theta}}$ such that*

$$\forall \pi, h, \quad \mathbb{P}_{\bar{\theta}}^\pi(\tau_h) \geq \mathbb{P}_\theta^\pi(\tau_h),$$

$$\forall \pi, h, \quad \sum_{\tau_h} \left|\mathbb{P}_{\bar{\theta}}^\pi(\tau_h) - \mathbb{P}_\theta^\pi(\tau_h)\right| \leq \epsilon.$$

*Moreover, $\log |\bar{\Theta}_\epsilon| \leq 2r^2 OAH^2 \log \frac{OA}{\epsilon}$.*

## B    GENERAL MLE ANALYSIS

In this section, we present four general propositions that characterize the performance of MLE.

We start with a proposition that states that the log-likelihood of the true model is relatively high compared to any model.

**Proposition 4** *Fix $\varepsilon < \frac{1}{KH}$. With probability at least $1 - \delta$, for any $\bar{\theta} \in \bar{\Theta}_\varepsilon$ and any $k \in [K]$, the following two inequalities hold:*

$$\forall \bar{\theta} \in \bar{\Theta}_\varepsilon, \quad \sum_h \sum_{(\tau_h, \pi) \in \mathcal{D}_h} \log \mathbb{P}_{\bar{\theta}}^\pi(\tau_h) - 3\log \frac{K|\bar{\Theta}_\varepsilon|}{\delta} \leq \sum_h \sum_{(\tau_h, \pi) \in \mathcal{D}_h^k} \log \mathbb{P}_{\theta^*}^\pi(\tau_h),$$

$$\forall \bar{\theta} \in \bar{\Theta}_\varepsilon, \quad \sum_{(\tau_H, \pi) \in \mathcal{D}^k} \log \mathbb{P}_{\bar{\theta}}^\pi(\tau_H) - 3\log \frac{K|\bar{\Theta}_\varepsilon|}{\delta} \leq \sum_{(\tau_H, \pi) \in \mathcal{D}} \log \mathbb{P}_{\theta^*}^\pi(\tau_H).$$

**Proof:** We start with the first inequality. Suppose the data in $\mathcal{D}_h^k$ is indexed by $t$. Then,

$$\mathbb{E}\left[\exp\left(\sum_h \sum_{(\tau_h,\pi)\in\mathcal{D}_h^k} \log\frac{\mathbb{P}_{\bar\theta}^\pi(\tau_h)}{\mathbb{P}_{\theta^*}^\pi(\tau_h)}\right)\right]$$

$$= \mathbb{E}\left[\prod_{t\leq k}\prod_h \frac{\mathbb{P}_{\bar\theta}^{\pi^t}(\tau_h^t)}{\mathbb{P}_{\theta^*}^{\pi^t}(\tau_h^t)}\right]$$

$$= \mathbb{E}\left[\prod_{t\leq k-1}\prod_h \frac{\mathbb{P}_{\bar\theta}^{\pi^t}(\tau_h^t)}{\mathbb{P}_{\theta^*}^{\pi^t}(\tau_h^t)}\mathbb{E}\left[\frac{\mathbb{P}_{\bar\theta}^{\pi^k}(\tau_h^k)}{\mathbb{P}_{\theta^*}^{\pi^k}(\tau_h^k)}\right]\right]$$

$$= \mathbb{E}\left[\prod_{t\leq k-1}\prod_h \frac{\mathbb{P}_{\bar\theta}^{\pi^t}(\tau_h^t)}{\mathbb{P}_{\theta^*}^{\pi^t}(\tau_h^t)}\prod_h\sum_{\tau_h}\mathbb{P}_{\bar\theta}^{\pi^k}(\tau_h)\right]$$

$$\overset{(a)}{\leq} (1+\varepsilon)^H \mathbb{E}\left[\prod_{t\leq k-1}\prod_h \frac{\mathbb{P}_{\bar\theta}^{\pi^t}(\tau_h^t)}{\mathbb{P}_{\theta^*}^{\pi^t}(\tau_h^t)}\right]$$

$$\leq (1+\varepsilon)^{KH}$$

$$\overset{(b)}{\leq} e,$$

where $(a)$ follows because $\sum_{\tau_H}|\mathbb{P}_{\bar\theta}^\pi(\tau_h) - \mathbb{P}_{\bar\theta}^\pi(\tau_h)| \leq \varepsilon$, and $(b)$ follows because $\varepsilon \leq \frac{1}{KH}$.

By the Chernoff bound and the union bound over $\bar\Theta_\epsilon$ and $k\in[K]$, with probability at least $1-\delta$, we have,

$$\forall k\in[K], \bar\theta\in\bar\Theta_\epsilon, \quad \sum_h\sum_{(\tau_h,\pi)\in\mathcal{D}_h^k}\log\frac{\mathbb{P}_{\bar\theta}^\pi(\tau_h)}{\mathbb{P}_{\theta^*}^\pi(\tau_h)} \leq 3\log\frac{K|\bar\Theta_\epsilon|}{\delta},$$

which yields the first result of this proposition.

To show the second inequality, we follow an argument similar to that for the first inequality. We have

$$\mathbb{E}\left[\exp\left(\sum_{(\tau_H,\pi)\in\mathcal{D}^k}\log\frac{\mathbb{P}_\theta^\pi(\tau_H)}{\mathbb{P}_{\theta^*}^\pi(\tau_H)}\right)\right] \leq \mathbb{E}\left[\exp\left(\sum_{(\tau_H,\pi)\in\mathcal{D}^k}\log\frac{\mathbb{P}_{\bar\theta}^\pi(\tau_H)}{\mathbb{P}_{\theta^*}^\pi(\tau_H)}\right)\right]$$

$$\overset{(a)}{\leq} (1+\varepsilon)^{KH} \leq e,$$

where $(a)$ follows from the tower rule of the expectation and because $\sum_{\tau_H}\mathbb{P}_{\bar\theta}^\pi(\tau_H) \leq \varepsilon$.

Thus, with probability at least $1-\delta$, for any $k\in[K]$ and any $\bar\theta\in\Theta$, the following inequality holds

$$\sum_{(\tau_H,\pi)\in\mathcal{D}^k}\log\frac{\mathbb{P}_{\bar\theta}^\pi(\tau_H)}{\mathbb{P}_{\theta^*}^\pi(\tau_H)} \leq 3\log\frac{K|\bar\Theta_\epsilon|}{\delta},$$

which completes the proof. ∎

The following proposition upper bounds the total variation distance between the conditional distributions over the future trajectory conditioned on the empirical history trajectories. This proposition is crucial to ensure that the model estimated by PSR-UCB is accurate on those sample trajectories.

**Proposition 5** *Fix $p_{\min}$ and $\varepsilon \leq \frac{p_{\min}}{KH}$. Let $\Theta_{\min}^k = \{\theta : \forall h, (\tau_h,\pi)\in\mathcal{D}_h^k, \ \mathbb{P}_\theta^\pi(\tau_h)\geq p_{\min}\}$. Consider the following event*

$$\mathcal{E}_\omega = \left\{\forall k\in[K], \forall\theta\in\Theta_{\min}^k, \quad \sum_h\sum_{(\tau_h,\pi)\in\mathcal{D}_h^k}\mathrm{D}_{\mathrm{TV}}^2\left(\mathbb{P}_\theta^\pi(\omega_h|\tau_h), \mathbb{P}_{\theta^*}^\pi(\omega_h|\tau_h)\right)\right.$$

$$\leq 6 \sum_{h} \sum_{(\tau_H, \pi) \in \mathcal{D}_h^k} \log \frac{\mathbb{P}_{\theta^*}^\pi(\tau_H)}{\mathbb{P}_\theta^\pi(\tau_H)} + 31 \log \frac{K \left| \bar{\Theta}_\varepsilon \right|}{\delta} \Bigg\}.$$

*Then,* $\mathbb{P}\left(\mathcal{E}_\omega\right) \geq 1 - \delta.$

**Proof:** We start with a general upper bound on the total variation distance between two conditional distributions. Note that for any $\theta, \theta' \in \Theta \cup \bar{\Theta}_\epsilon$ and fixed $(\tau_h, \pi)$, we have

$$
\begin{aligned}
& \mathsf{D}_{\mathrm{TV}}\left(\mathbb{P}_\theta^\pi(\omega_h | \tau_h), \mathbb{P}_{\theta'}^\pi(\omega_h | \tau_h)\right) \\
&= \sum_{\omega_h} \left| \frac{\mathbb{P}_{\theta'}^\pi(\omega_h, \tau_h) \mathbb{P}_\theta^\pi(\tau_h) - \mathbb{P}_\theta^\pi(\omega_h, \tau_h) \mathbb{P}_{\theta'}^\pi(\tau_h)}{\mathbb{P}_\theta^\pi(\tau_h) \mathbb{P}_{\theta'}^\pi(\tau_h)} \right| \\
&= \sum_{\omega_h} \left| \frac{\left(\mathbb{P}_{\theta'}^\pi(\omega_h, \tau_h) - \mathbb{P}_\theta^\pi(\omega_h, \tau_h)\right) \mathbb{P}_\theta^\pi(\tau_h) + \mathbb{P}_\theta^\pi(\omega_h, \tau_h) \left(\mathbb{P}_\theta^\pi(\tau_h) - \mathbb{P}_{\theta'}^\pi(\tau_h)\right)}{\mathbb{P}_\theta^\pi(\tau_h) \mathbb{P}_{\theta'}^\pi(\tau_h)} \right| \\
&\leq \frac{\left| \mathbb{P}_\theta^\pi(\tau_h) - \mathbb{P}_{\theta'}^\pi(\tau_h) \right|}{\mathbb{P}_{\theta'}^\pi(\tau_h)} + \frac{1}{\mathbb{P}_{\theta'}^\pi(\tau_h)} \sum_{\omega_h} \left| \left(\mathbb{P}_{\theta'}^\pi(\omega_h, \tau_h) - \mathbb{P}_\theta^\pi(\omega_h, \tau_h)\right) \right| \\
&\leq \frac{2}{\mathbb{P}_{\theta'}^\pi(\tau_h)} \mathsf{D}_{\mathrm{TV}}\left(\mathbb{P}_\theta^\pi(\tau_H), \mathbb{P}_{\theta'}^\pi(\tau_H)\right).
\end{aligned}
$$

By symmetry, we also have

$$\mathsf{D}_{\mathrm{TV}}\left(\mathbb{P}_\theta^\pi(\omega_h | \tau_h), \mathbb{P}_{\theta'}^\pi(\omega_h | \tau_h)\right) \leq \frac{2}{\max\left\{\mathbb{P}_\theta^\pi(\tau_h), \mathbb{P}_{\theta'}^\pi(\tau_h)\right\}} \mathsf{D}_{\mathrm{TV}}\left(\mathbb{P}_\theta^\pi(\tau_H), \mathbb{P}_{\theta'}^\pi(\tau_H)\right).$$

We replace $\theta'$ by a $\bar{\theta} \in \bar{\Theta}_\varepsilon$ that is $\varepsilon$-optimistic over $\theta$ (recall Definition 2), i.e. $\mathsf{D}_{\mathrm{TV}}\left(\mathbb{P}_\theta^\pi, \mathbb{P}_{\bar{\theta}}^\pi\right) \leq \varepsilon$, and $\mathbb{P}_{\bar{\theta}}^\pi(\tau_h) \geq \mathbb{P}_\theta^\pi(\tau_h)$ holds for any $\pi$ and $\tau_h$. Then, due the construction of $\Theta_{\min}^k$, we have

$$\forall (\tau_h, \pi) \in \mathcal{D}_h^k, \ \ \mathsf{D}_{\mathrm{TV}}\left(\mathbb{P}_\theta^\pi(\omega_h | \tau_h), \mathbb{P}_{\bar{\theta}}^\pi(\omega_h | \tau_h)\right) \leq \frac{2\epsilon}{p_{\min}} \leq \frac{2}{KH},$$

which implies

$$
\begin{aligned}
& \sum_h \sum_{(\tau_h, \pi) \in \mathcal{D}_h^k} \mathsf{D}_{\mathrm{TV}}^2\left(\mathbb{P}_\theta^\pi(\omega_h | \tau_h), \mathbb{P}_{\theta^*}^\pi(\omega_h | \tau_h)\right) \\
&\overset{(a)}{\leq} \sum_h \sum_{(\tau_h, \pi) \in \mathcal{D}_h^k} 2\mathsf{D}_{\mathrm{TV}}^2\left(\mathbb{P}_\theta^\pi(\omega_h | \tau_h), \mathbb{P}_{\bar{\theta}}^\pi(\omega_h | \tau_h)\right) + 2\mathsf{D}_{\mathrm{TV}}^2\left(\mathbb{P}_{\bar{\theta}}^\pi(\omega_h | \tau_h), \mathbb{P}_{\theta^*}^\pi(\omega_h | \tau_h)\right) \\
&\leq \frac{4}{KH} + 2 \sum_h \sum_{(\tau_h, \pi) \in \mathcal{D}_h^k} \mathsf{D}_{\mathrm{TV}}^2\left(\mathbb{P}_{\bar{\theta}}^\pi(\omega_h | \tau_h), \mathbb{P}_{\theta^*}^\pi(\omega_h | \tau_h)\right).
\end{aligned}
$$

Here $(a)$ follows because the total variation distance satisfies the triangle inequality and $(a + b)^2 \leq 2a^2 + 2b^2$.

Moreover, note that

$$
\begin{aligned}
& \mathsf{D}_{\mathrm{TV}}^2\left(\mathbb{P}_{\bar{\theta}}^\pi(\omega_h | \tau_h), \mathbb{P}_{\theta^*}^\pi(\omega_h | \tau_h)\right) \\
&\overset{(a)}{\leq} 4(2 + 2/(KH)) \mathsf{D}_{\mathrm{H}}^2\left(\mathbb{P}_{\bar{\theta}}^\pi(\omega_h | \tau_h), \mathbb{P}_{\theta^*}^\pi(\omega_h | \tau_h)\right) \\
&\leq 6 \left(1 + \frac{1}{KH} - \mathbb{E}_{\omega_h \sim \mathbb{P}_{\theta^*}^\pi} \sqrt{\frac{\mathbb{P}_{\bar{\theta}}^\pi(\omega_h | \tau_h)}{\mathbb{P}_{\theta^*}^\pi(\omega_h | \tau_h)}}\right) \\
&\overset{(b)}{\leq} -6 \log \mathbb{E}_{\omega_h \sim \mathbb{P}_{\theta^*}^\pi(\cdot | \tau_h)} \sqrt{\frac{\mathbb{P}_{\bar{\theta}}^\pi(\omega_h | \tau_h)}{\mathbb{P}_{\theta^*}^\pi(\omega_h | \tau_h)}} + \frac{6}{KH},
\end{aligned}
$$

where $(a)$ is due to Lemma 12 and $(b)$ follows because $1 - x \leq -\log x$ for any $x > 0$.

Thus, the summation of the total variation distance between conditional distributions conditioned on $(\tau_h, \pi) \in \mathcal{D}_h^k$ can be upper bounded by

$$\sum_h \sum_{(\tau_h, \pi) \in \mathcal{D}_h^k} \mathrm{D}_{\mathrm{TV}}^2 \left( \mathbb{P}_\theta^\pi(\omega_h | \tau_h), \mathbb{P}_{\theta^*}^\pi(\omega_h | \tau_h) \right)$$

$$\leq \frac{18}{KH} - 12 \sum_h \sum_{(\tau_h, \pi) \in \mathcal{D}_h^k} \log \mathop{\mathbb{E}}_{\omega_h \sim \mathbb{P}_{\theta^*}^\pi(\cdot | \tau_h)} \sqrt{\frac{\mathbb{P}_{\bar{\theta}}^\pi(\omega_h | \tau_h)}{\mathbb{P}_{\theta^*}^\pi(\omega_h | \tau_h)}}.$$

In addition, we have

$$\mathop{\mathbb{E}}_{\substack{\forall h, (\tau_h, \pi) \in \mathcal{D}_h^k, \\ \omega_h \sim \mathbb{P}_{\theta^*}^\pi(\cdot | \tau_h)}} \left[ \exp \left( \frac{1}{2} \sum_h \sum_{(\omega_h, \tau_h, \pi) \in \mathcal{D}_h^k} \log \frac{\mathbb{P}_{\bar{\theta}}^\pi(\omega_h | \tau_h)}{\mathbb{P}_{\theta^*}^\pi(\omega_h | \tau_h)} - \sum_h \sum_{(\tau_h, \pi) \in \mathcal{D}_h^k} \log \mathop{\mathbb{E}}_{\omega_h \sim \mathbb{P}_{\theta^*}^\pi(\cdot | \tau_h)} \sqrt{\frac{\mathbb{P}_{\bar{\theta}}^\pi(\omega_h | \tau_h)}{\mathbb{P}_{\theta^*}^\pi(\omega_h | \tau_h)}} \right) \right]$$

$$= \frac{\mathop{\mathbb{E}}_{\substack{\forall h, (\tau_h, \pi) \in \mathcal{D}_h^k, \\ \omega_h \sim \mathbb{P}_{\theta^*}^\pi(\cdot | \tau_h)}} \left[ \prod_h \prod_{(\omega_h, \tau_h) \in \mathcal{D}_h^k} \sqrt{\frac{\mathbb{P}_{\bar{\theta}}^\pi(\omega_h | \tau_h)}{\mathbb{P}_{\theta^*}^\pi(\omega_h | \tau_h)}} \right]}{\prod_h \prod_{(\tau_h, \pi) \in \mathcal{D}_h^k} \mathop{\mathbb{E}}_{\omega_h \sim \mathbb{P}_{\theta^*}^\pi(\cdot | \tau_h)} \left[ \sqrt{\frac{\mathbb{P}_{\bar{\theta}}^\pi(\omega_h | \tau_h)}{\mathbb{P}_{\theta^*}^\pi(\omega_h | \tau_h)}} \right]} = 1,$$

where the last equality is due to the conditional independence of $\omega_h \in \mathcal{D}_h^k$ given $(\tau_h, \pi) \in \mathcal{D}_h^k$.

Therefore, by the Chernoff bound, with probability $1 - \delta$, we have

$$-\sum_h \sum_{(\tau_h, \pi) \in \mathcal{D}_h^k} \log \mathop{\mathbb{E}}_{\omega_h \sim \mathbb{P}_{\theta^*}^\pi(\cdot | \tau_h)} \sqrt{\frac{\mathbb{P}_{\bar{\theta}}^\pi(\omega_h | \tau_h)}{\mathbb{P}_{\theta^*}^\pi(\omega_h | \tau_h)}} \leq \frac{1}{2} \sum_h \sum_{(\omega_h, \tau_h, \pi) \in \mathcal{D}_h^k} \log \frac{\mathbb{P}_{\theta^*}^\pi(\omega_h | \tau_h)}{\mathbb{P}_{\bar{\theta}}^\pi(\omega_h | \tau_h)} + \log \frac{1}{\delta}.$$

Taking the union bound over $\bar{\Theta}_\epsilon$, $k \in [K]$, and rescaling $\delta$, we have, with probability at least $1 - \delta$, $\forall k \in [K]$, the following inequality holds:

$$\sum_h \sum_{(\tau_h, \pi) \in \mathcal{D}_h^k} \mathrm{D}_{\mathrm{TV}}^2 \left( \mathbb{P}_\theta^\pi(\omega_h | \tau_h), \mathbb{P}_{\theta^*}^\pi(\omega_h | \tau_h) \right)$$

$$\leq \frac{18}{KH} + 6 \sum_h \sum_{(\omega_h, \tau_h, \pi) \in \mathcal{D}_h^k} \log \frac{\mathbb{P}_{\theta^*}^\pi(\omega_h | \tau_h)}{\mathbb{P}_{\bar{\theta}}^\pi(\omega_h | \tau_h)} + 12 \log \frac{K \left| \bar{\Theta}_\varepsilon \right|}{\delta}$$

$$\leq 6 \sum_h \sum_{(\omega_h, \tau_h, \pi) \in \mathcal{D}_h^k} \log \frac{\mathbb{P}_{\theta^*}^\pi(\omega_h, \tau_h)}{\mathbb{P}_{\bar{\theta}}^\pi(\omega_h, \tau_h)} + 6 \sum_h \sum_{(\tau_h, \pi) \in \mathcal{D}_h^k} \log \frac{\mathbb{P}_{\bar{\theta}}^\pi(\tau_h)}{\mathbb{P}_{\theta^*}^\pi(\tau_h)} + 13 \log \frac{K \left| \bar{\Theta}_\varepsilon \right|}{\delta}.$$

Note that, following from Proposition 4, with probability at least $1 - \delta$, we have for any $k \in [K]$,

$$\sum_h \sum_{(\tau_h, \pi) \in \mathcal{D}_h^k} \log \frac{\mathbb{P}_{\bar{\theta}}^\pi(\tau_h)}{\mathbb{P}_{\theta^*}^\pi(\tau_h)} \leq 3 \log \frac{K \left| \bar{\Theta}_\varepsilon \right|}{\delta}.$$

Hence, combining with the optimistic property of $\bar{\theta}$ and rescaling $\delta$, we have that the following inequality holds with probability at least $1 - \delta$:

$$\sum_h \sum_{(\tau_h, \pi) \in \mathcal{D}_h^k} \mathrm{D}_{\mathrm{TV}}^2 \left( \mathbb{P}_\theta^\pi(\omega_h | \tau_h), \mathbb{P}_{\theta^*}^\pi(\omega_h | \tau_h) \right) \leq 6 \sum_h \sum_{(\tau_H, \pi) \in \mathcal{D}_h^k} \log \frac{\mathbb{P}_{\theta^*}^\pi(\tau_H)}{\mathbb{P}_\theta^\pi(\tau_H)} + 31 \log \frac{K \left| \bar{\Theta}_\varepsilon \right|}{\delta},$$

which yields the final result. ∎

The following proposition is standard in the MLE analysis, and we provide the full analysis here for completeness.

**Proposition 6** *Fix $\varepsilon < \frac{1}{K^2 H^2}$. Define the following event:*

$$\mathcal{E}_\pi = \left\{ \forall \theta \in \Theta, \forall k \in [K], \quad \sum_{\pi \in \mathcal{D}^k} \mathrm{D}_{\mathrm{H}}^2(\mathbb{P}_\theta^\pi(\tau_H), \mathbb{P}_{\theta^*}^\pi(\tau_H)) \leq \frac{1}{2} \sum_{(\tau_H, \pi) \in \mathcal{D}^k} \log \frac{\mathbb{P}_{\theta^*}^\pi(\tau_H)}{\mathbb{P}_\theta^\pi(\tau_H)} + 2 \log \frac{K|\bar{\Theta}_\varepsilon|}{\delta} \right\}.$$

*We have $\mathbb{P}(\mathcal{E}_\pi) \geq 1 - \delta$.*

**Proof:** First, by the construction of $\bar{\Theta}_\varepsilon$, for any $\theta$, let $\bar{\theta}$ be optimistic over $\theta$, i.e., $\sum_{\tau_H} \left| \mathbb{P}_\theta^\pi(\tau_H) - \mathbb{P}_{\bar{\theta}}^\pi(\tau_H) \right| \leq \varepsilon$. We translate the distance between $\theta$ and $\theta^*$ to the distance between $\bar{\theta}$ and $\theta^*$ as follows.

$$\mathrm{D}_{\mathrm{H}}^2(\mathbb{P}_\theta^\pi(\tau_H), \mathbb{P}_{\theta^*}^\pi(\tau_H))$$

$$= 1 - \sum_{\tau_H} \sqrt{\mathbb{P}_\theta^\pi(\tau_H) \mathbb{P}_{\theta^*}^\pi(\tau_H)}$$

$$= 1 - \sum_{\tau_H} \sqrt{\mathbb{P}_{\bar{\theta}}^\pi(\tau_H) \mathbb{P}_{\theta^*}^\pi(\tau_H) + \left( \mathbb{P}_\theta^\pi(\tau_H) - \mathbb{P}_{\bar{\theta}}^\pi(\tau_H) \right) \mathbb{P}_{\theta^*}^\pi(\tau_H)}$$

$$\overset{(a)}{\leq} 1 - \sum_{\tau_H} \sqrt{\mathbb{P}_{\bar{\theta}}^\pi(\tau_H) \mathbb{P}_{\theta^*}^\pi(\tau_H)} + \sum_{\tau_H} \sqrt{\left| \mathbb{P}_\theta^\pi(\tau_H) - \mathbb{P}_{\bar{\theta}}^\pi(\tau_H) \right| \mathbb{P}_{\theta^*}^\pi(\tau_H)}$$

$$\overset{(b)}{\leq} - \log \mathbb{E}_{\tau_H \sim \mathbb{P}_{\theta^*}^\pi(\cdot)} \sqrt{\frac{\mathbb{P}_{\bar{\theta}}^\pi(\tau_H)}{\mathbb{P}_{\theta^*}^\pi(\tau_H)}} + \sqrt{\sum_{\tau_H} \left| \mathbb{P}_\theta^\pi(\tau_H) - \mathbb{P}_{\bar{\theta}}^\pi(\tau_H) \right|}$$

$$\leq - \log \mathbb{E}_{\tau_H \sim \mathbb{P}_{\theta^*}^\pi(\cdot)} \sqrt{\frac{\mathbb{P}_{\bar{\theta}}^\pi(\tau_H)}{\mathbb{P}_{\theta^*}^\pi(\tau_H)}} + \sqrt{\varepsilon},$$

where $(a)$ follows because $\sqrt{a+b} \geq \sqrt{a} - \sqrt{|b|}$ if $a > 0$ and $a + b > 0$, and $(b)$ follows from the Cauchy's inequality and the fact that $1 - x \leq -\log x$.

Hence, in order to upper bound $\sum_{\pi \in \mathcal{D}_k} \mathrm{D}_{\mathrm{H}}^2(\mathbb{P}_\theta^\pi(\tau_H), \mathbb{P}_{\theta^*}^\pi(\tau_H))$, it suffices to upper bound $\sum_{\pi \in \mathcal{D}_k} - \log \mathbb{E}_{\tau_H \sim \mathbb{P}_{\theta^*}^\pi(\cdot)} \sqrt{\frac{\mathbb{P}_{\bar{\theta}}^\pi(\tau_H)}{\mathbb{P}_{\theta^*}^\pi(\tau_H)}}$. To this end, we observe that,

$$\mathbb{E}\left[ \exp\left( \frac{1}{2} \sum_{(\tau_H, \pi) \in \mathcal{D}^k} \log \frac{\mathbb{P}_{\bar{\theta}}^\pi(\tau_H)}{\mathbb{P}_{\theta^*}^\pi(\tau_H)} - \sum_{\pi \in \mathcal{D}^k} \log \mathbb{E}_{\tau_H \sim \mathbb{P}_{\theta^*}^\pi(\cdot)} \sqrt{\frac{\mathbb{P}_{\bar{\theta}}^\pi(\tau_H)}{\mathbb{P}_{\theta^*}^\pi(\tau_H)}} \right) \right]$$

$$\overset{(a)}{=} \frac{\mathbb{E}\left[ \prod_{(\tau_H, \pi) \in \mathcal{D}^k} \sqrt{\frac{\mathbb{P}_{\bar{\theta}}^\pi(\tau_H)}{\mathbb{P}_{\theta^*}^\pi(\tau_H)}} \right]}{\mathbb{E}\left[ \prod_{(\tau_H, \pi) \in \mathcal{D}^k} \sqrt{\frac{\mathbb{P}_{\bar{\theta}}^\pi(\tau_H)}{\mathbb{P}_{\theta^*}^\pi(\tau_H)}} \right]} = 1,$$

where $(a)$ follows because $(\tau_H, \pi) \in \mathcal{D}^k$ form a filtration.

Then, by the Chernoff bound, we have

$$\mathbb{P}\left( \frac{1}{2} \sum_{(\tau_H, \pi) \in \mathcal{D}^k} \log \frac{\mathbb{P}_{\bar{\theta}}^\pi(\tau_H)}{\mathbb{P}_{\theta^*}^\pi(\tau_H)} - \sum_{\pi \in \mathcal{D}^k} \log \mathbb{E}_{\tau_H \sim \mathbb{P}_{\theta^*}^\pi(\cdot)} \sqrt{\frac{\mathbb{P}_{\bar{\theta}}^\pi(\tau_H)}{\mathbb{P}_{\theta^*}^\pi(\tau_H)}} \geq \log \frac{1}{\delta} \right)$$

$$= \mathbb{P}\left( \exp\left( \frac{1}{2} \sum_{(\tau_H, \pi) \in \mathcal{D}^k} \log \frac{\mathbb{P}_{\bar{\theta}}^\pi(\tau_H)}{\mathbb{P}_{\theta^*}^\pi(\tau_H)} - \sum_{\pi \in \mathcal{D}^k} \log \mathbb{E}_{\tau_H \sim \mathbb{P}_{\theta^*}^\pi(\cdot)} \sqrt{\frac{\mathbb{P}_{\bar{\theta}}^\pi(\tau_H)}{\mathbb{P}_{\theta^*}^\pi(\tau_H)}} \right) \geq \frac{1}{\delta} \right)$$

$$\overset{(a)}{\leq} \delta \mathbb{E}\left[ \exp\left( \frac{1}{2} \sum_{(\tau_H, \pi) \in \mathcal{D}^k} \log \frac{\mathbb{P}_{\bar{\theta}}^\pi(\tau_H)}{\mathbb{P}_{\theta^*}^\pi(\tau_H)} - \sum_{\pi \in \mathcal{D}^k} \log \mathbb{E}_{\tau_H \sim \mathbb{P}_{\theta^*}^\pi(\cdot)} \sqrt{\frac{\mathbb{P}_{\bar{\theta}}^\pi(\tau_H)}{\mathbb{P}_{\theta^*}^\pi(\tau_H)}} \right) \right]$$

$$\leq \delta,$$

where $(a)$ is due to the Markov inequality.

Finally, rescaling $\delta$ to $\delta/(K|\bar{\Theta}_\varepsilon|)$ and taking the union bound over $\bar{\Theta}_\epsilon$ and $k \in [K]$, we conclude that, with probability at least $1 - \delta$, $\forall \theta \in \Theta, k \in [K]$,

$$\sum_{\pi \in \mathcal{D}^k} \mathrm{D}_{\mathrm{H}}^2(\mathbb{P}_\theta^\pi(\tau_H), \mathbb{P}_{\theta^*}^\pi(\tau_H))$$

$$\leq KH\sqrt{\varepsilon} + \frac{1}{2} \sum_{(\tau_H, \pi) \in \mathcal{D}^k} \log \frac{\mathbb{P}_{\theta^*}^\pi(\tau_H)}{\mathbb{P}_{\bar{\theta}}^\pi(\tau_H)} + \log \frac{K|\bar{\Theta}_\epsilon|}{\delta}$$

$$\stackrel{(a)}{\leq} \frac{1}{2} \sum_{(\tau_H, \pi) \in \mathcal{D}^k} \log \frac{\mathbb{P}_{\theta^*}^\pi(\tau_H)}{\mathbb{P}_\theta^\pi(\tau_H)} + 2\log \frac{K|\bar{\Theta}_\epsilon|}{\delta},$$

where $(a)$ follows because $\varepsilon \leq \frac{1}{K^2 H^2}$. $\blacksquare$

The following proposition states that the constraint $\Theta_{\min}^k$ does not rule out the true model.

**Proposition 7** *Fix* $p_{\min} \leq \frac{\delta}{KH(|\mathcal{O}||\mathcal{A}|)^H}$. *Consider the following event:*

$$\mathcal{E}_{\min} = \left\{ \forall k \in [K], \forall h, (\tau_h, \pi) \in \mathcal{D}_h^k, \ \mathbb{P}_{\theta^*}^\pi(\tau_h) \geq p_{\min} \right\} = \bigcap_{k \in [K]} \left\{ \theta^* \in \Theta_{\min}^k \right\}.$$

*We have* $\mathbb{P}(\mathcal{E}_{\min}) \geq 1 - \delta$.

**Proof:** For any $k \in [K]$, $h \in \{0, 1, \ldots, H - 1\}$ and $(\tau_h, \pi) \in \mathcal{D}_h^k$, we have

$$\mathbb{P}\left(\mathbb{P}_{\theta^*}^\pi(\tau_h) < p_{\min}\right)$$

$$= \mathbb{E}\left[\mathbb{P}\left(\mathbb{P}_{\theta^*}^\pi(\tau_h) < p_{\min} \middle| \pi\right)\right]$$

$$= \mathbb{E}\left[\sum_{\tau_h} \mathbb{P}_{\theta^*}^\pi(\tau_h^t) \mathbb{1}\left\{\mathbb{P}_{\theta^*}^{\pi^t}(\tau_h^t) < p_{\min}\right\}\right]$$

$$\leq \mathbb{E}\left[\sum_{\tau_h} p_{\min}\right]$$

$$\leq \frac{\delta}{KH}.$$

Thus, taking the union bound over $k \in [K]$, $h \in \{0, 1, \ldots, H - 1\}$ and $(\tau_h, \pi) \in \mathcal{D}_h$, we conclude that

$$\mathbb{P}(\mathcal{E}_{\min}) \geq 1 - \delta.$$

$\blacksquare$

## C    PROOF OF THEOREM 1 (FOR ONLINE PSR-UCB)

In this section, we present the full analysis for the online algorithm PSR-UCB to show Theorem 1. In particular, the proof of Theorem 1 consists of three main steps. **Step 1.** We prove that the estimated model $\hat{\theta}^k$ is not only accurate over the exploration policies, but also accurate conditioned on emipircal samples $\tau_h \in \mathcal{D}_h^k$. **Step 2.** Building up on the first step, we are able to show that $V_{\hat{\theta}^k, \hat{b}^k}^\pi$ is a valid upper bound on the total variation distance between $\hat{\theta}^k$ and $\theta^*$. **Step 3.** Based on a newly developed inequality that translates $V_{\hat{\theta}^k, \hat{b}^k}^\pi$ to the ground-truth prediction feature $\bar{\psi}^*(\tau_h)$, we show that the summation of $V_{\hat{\theta}^k, \hat{b}^k}^\pi$ over the iteration $k$ grows sublinear in time. This finally characterizes the optimality of the output policy, the accuracy of the output model and the sample complexity of PSR-UCB, and finishes the proof.

Before we proceed, we use $\mathcal{E}$ to denote the event $\mathcal{E}_\omega \cap \mathcal{E}_\pi \cap \mathcal{E}_{\min}$, where these three events are defined in Appendix B. Due to Propositions 5 to 7 and union bound, we immediately have $\mathbb{P}(\mathcal{E}) \geq 1 - 3\delta$.

## C.1 STEP 1: ESTIMATION GUARANTEE

We show that the estimated model is accurate with the past exploration policies and dataset.

**Lemma 1** *Under event $\mathcal{E}$, the following two inequalities hold:*

$$
\begin{cases}
\displaystyle\sum_{h}\sum_{(\tau_h,\pi)\in\mathcal{D}_h^k} \mathtt{D}_{\mathrm{TV}}^2\left(\mathbb{P}_{\hat{\theta}^k}^\pi(\omega_h|\tau_h), \mathbb{P}_{\theta^*}^\pi(\omega_h|\tau_h)\right) \le 7\beta, \\
\displaystyle\sum_{\pi\in\mathcal{D}^k} \mathtt{D}_{\mathrm{H}}^2\left(\mathbb{P}_{\hat{\theta}^k}^\pi(\tau_H), \mathbb{P}_{\theta^*}^\pi(\tau_H)\right) \le 7\beta,
\end{cases}
$$

*where $\beta = 31\log\frac{K|\bar{\Theta}_\varepsilon|}{\delta}$ and $\varepsilon \le \frac{\delta}{K^2 H^2 (|\mathcal{O}||\mathcal{A}|)^H}$.*

**Proof:** To show the first inequality, note that by the selection of $\hat{\theta}^k$, we have $\hat{\theta}^k \in \mathcal{B}^k$ (defined in Equation (2)).

Following from Proposition 5, we have

$$
\sum_{h}\sum_{(\tau_h,\pi)\in\mathcal{D}_h^k} \mathtt{D}_{\mathrm{TV}}^2\left(\mathbb{P}_{\hat{\theta}^k}^\pi(\omega_h|\tau_h), \mathbb{P}_{\theta^*}^\pi(\omega_h|\tau_h)\right)
$$

$$
\le 6\sum_{h}\sum_{(\tau_H,\pi)\in\mathcal{D}_h^k} \log\mathbb{P}_{\theta^*}^\pi(\tau_H) - 6\sum_{h}\sum_{(\tau_H,\pi)\in\mathcal{D}_h^k} \log\mathbb{P}_{\hat{\theta}^k}^\pi(\tau_H) + 31\log\frac{K|\bar{\Theta}_\varepsilon|}{\delta}
$$

$$
\overset{(a)}{\le} 6\max_{\theta'\in\Theta_{\min}^k}\sum_{h}\sum_{(\tau_H,\pi)\in\mathcal{D}_h^k} \log\mathbb{P}_{\theta'}^\pi(\tau_H) - 6\sum_{h}\sum_{(\tau_H,\pi)\in\mathcal{D}_h^k} \log\mathbb{P}_{\hat{\theta}^k}^\pi(\tau_H) + 31\log\frac{K|\bar{\Theta}_\varepsilon|}{\delta}
$$

$$
\le 7\beta,
$$

where $(a)$ follows from $\theta^* \in \Theta_{\min}^k$ (Proposition 7).

To show the second inequality, Proposition 6 implies that

$$
\sum_{\pi\in\mathcal{D}^k} \mathtt{D}_{\mathrm{H}}^2\left(\mathbb{P}_{\hat{\theta}^k}^\pi(\tau_H), \mathbb{P}_{\theta^*}^\pi(\tau_H)\right)
$$

$$
\le \sum_{\pi\in\mathcal{D}^k}\log\mathbb{P}_{\theta^*}^\pi(\tau_H) - \sum_{\pi\in\mathcal{D}^k}\log\mathbb{P}_{\hat{\theta}^k}^\pi(\tau_H) + 2\log\frac{K|\bar{\Theta}_\varepsilon|}{\delta}
$$

$$
\overset{(a)}{\le} \max_{\theta'\in\Theta_{\min}^k}\sum_{\pi\in\mathcal{D}^k}\log\mathbb{P}_{\theta'}^\pi(\tau_H) - \sum_{\pi\in\mathcal{D}^k}\log\mathbb{P}_{\hat{\theta}^k}^\pi(\tau_H) + 2\log\frac{K|\bar{\Theta}_\varepsilon|}{\delta}
$$

$$
\le 7\beta,
$$

where $(a)$ follows from $\theta^* \in \Theta_{\min}^k$ (Proposition 7). ∎

## C.2 STEP 2: UCB FOR TOTAL VARIATION DISTANCE

Following from Proposition 1, the total variation distance between two PSRs is controlled by the estimation error. Hence, we first characterize the estimation error of $\mathbf{M}_h^*(o_h, a_h)$ in the following lemma.

**Lemma 2** *Under event $\mathcal{E}$, for any $k \in [K]$ and any policy $\pi$, we have*

$$
\sum_{\tau_H}\left|\mathbf{m}^*(\omega_h)^\top\left(\hat{\mathbf{M}}_h^k(o_h,a_h) - \mathbf{M}_h^*(o_h,a_h)\right)\hat{\psi}_{h-1}^k(\tau_{h-1})\right|\pi(\tau_H)
$$

$$
\le \mathbb{E}_{\tau_{h-1}\sim\mathbb{P}_{\hat{\theta}(k)}^\pi}\left[\alpha_{h-1}^k\left\|\bar{\hat{\psi}}_{h-1}^k(\tau_{h-1})\right\|_{(\hat{U}_{h-1}^k)^{-1}}\right], \tag{9}
$$

*where*

$$\hat{U}_{h-1}^k = \lambda I + \sum_{\tau_{h-1} \in \mathcal{D}_{h-1}^k} \left[ \bar{\hat{\psi}}^k(\tau_{h-1}) \bar{\hat{\psi}}^k(\tau_{h-1})^\top \right],$$

$$\alpha_{h-1}^k = \frac{4\lambda Q_A^2 d}{\gamma^4} + \frac{|\mathcal{A}|^2 Q_A^2}{\gamma^2} \sum_{\tau_{h-1} \in \mathcal{D}_{h-1}^k} \mathtt{D}_{\mathtt{TV}}^2 \left( \mathbb{P}_{\hat{\theta}^k}^{\mathbf{u}_{\mathcal{Q}_{h-1}^{\exp}}}(\omega_{h-1}|\tau_{h-1}), \mathbb{P}_{\theta^*}^{\mathbf{u}_{\mathcal{Q}_{h-1}^{\exp}}}(\omega_{h-1}|\tau_{h-1}) \right).$$

**Proof:** We index future trajectory $\omega_{h-1} = (o_h, a_h, \ldots, o_H, a_H)$ by $i$, and history trajectory $\tau_{h-1}$ by $j$. For simplicity, we denote $\mathbf{m}^*(\omega_h)^\top \left( \hat{\mathbf{M}}_h^{(k)}(o_h, a_h) - \mathbf{M}_h^*(o_h, a_h) \right)$ as $w_i^\top$, and $\bar{\hat{\psi}}_{h-1}^{(k)}(\tau_{h-1}) = \frac{\hat{\psi}_{h-1}^k(\tau_{h-1})}{\hat{\phi}_{h-1}^{k\top} \hat{\psi}_{h-1}^k(\tau_{h-1})}$ as $x_j$. We also denote $\pi(\omega_{h-1}|\tau_{h-1})$ as $\pi_{i|j}$.

Then, the LHS of Equation (9) can be written as

$$\begin{aligned}
\text{LHS} &= \sum_{\tau_H} \left| \mathbf{m}^*(\omega_h)^\top \left( \hat{\mathbf{M}}_h^k(o_h, a_h) - \mathbf{M}_h^*(o_h, a_h) \right) \hat{\psi}_{h-1}^k(\tau_{h-1}) \right| \pi(\tau_H) \\
&\overset{(a)}{=} \sum_i \sum_j |w_i^\top x_j| \pi_{i|j} \mathbb{P}_{\hat{\theta}^k}^\pi(j) \\
&= \sum_j \sum_i (\pi_{i|j} \cdot \mathtt{sgn}(w_i^\top x_j) \cdot w_i)^\top x_j \cdot \mathbb{P}_{\hat{\theta}^k}^\pi(j) \\
&= \sum_j \left( \sum_i \pi_{i|j} \cdot \mathtt{sgn}(w_i^\top x_j) \cdot w_i \right)^\top x_j \cdot \mathbb{P}_{\hat{\theta}^k}^\pi(j) \\
&\overset{(b)}{\leq} \mathbb{E}_{j \sim \mathbb{P}_{\hat{\theta}^k}^\pi} \left[ \|x_j\|_{(\hat{U}_{h-1}^k)^{-1}} \sqrt{\left\| \sum_i \pi_{i|j} \cdot \mathtt{sgn}(w_i^\top x_j) \cdot w_i \right\|_{\hat{U}_{h-1}^k}^2} \right],
\end{aligned}$$

where $(a)$ follows because $\mathbb{P}_\theta^\pi(\tau_h) = \phi_h^\top \psi(\tau_h)$, and $(b)$ follows from the Cauchy's inequality.

Fix $\tau_{h-1} = j_0$. We aim to upper bound the term $I_1 := \left\| \sum_i \pi_{i|j_0} \cdot \mathtt{sgn}(w_i^\top x_{j_0}) \cdot w_i \right\|_{\hat{U}_{h-1}^k}^2$. Then, we have

$$I_1 = \underbrace{\lambda \left\| \sum_i \pi_{i|j_0} \cdot \mathtt{sgn}(w_i^\top x_{j_0}) \cdot w_i \right\|_2^2}_{I_2} + \underbrace{\sum_{j \in \mathcal{D}_{h-1}^\tau} \left[ \left( \sum_i \pi_{i|j_0} \cdot \mathtt{sgn}(w_i^\top x_{j_0}) \cdot w_i \right)^\top x_j \right]^2}_{I_3}.$$

We first upper bound the first term $I_2$ as follows:

$$\begin{aligned}
\sqrt{I_2} &= \sqrt{\lambda} \max_{x \in \mathbb{R}^{d_{h-1}} : \|x\|_2 = 1} \left| \sum_i \pi_{i|j_0} \mathtt{sgn}(w_i^\top x_{j_0}) w_i^\top x \right| \\
&\leq \sqrt{\lambda} \max_{x \in \mathbb{R}^{d_{h-1}} : \|x\|_2 = 1} \sum_{\omega_{h-1}} \left| \mathbf{m}^*(\omega_h)^\top \left( \hat{\mathbf{M}}_h^k(o_h, a_h) - \mathbf{M}_h^*(o_h, a_h) \right) x \right| \pi(\omega_{h-1}|j_0) \\
&\overset{(a)}{\leq} \sqrt{\lambda} \max_{x \in \mathbb{R}^{d_{h-1}} : \|x\|_2 = 1} \sum_{\omega_{h-1}} \left| \mathbf{m}^*(\omega_h)^\top \hat{\mathbf{M}}_h^k(o_h, a_h) x \right| \pi(\omega_{h-1}|j_0) \\
&\quad + \sqrt{\lambda} \max_{x \in \mathbb{R}^{d_{h-1}} : \|x\|_2 = 1} \sum_{\omega_{h-1}} \left| \mathbf{m}^*(\omega_h)^\top \mathbf{M}_h^*(o_h, a_h) x \right| \pi(\omega_{h-1}|j_0) \\
&\overset{(b)}{\leq} \frac{\sqrt{\lambda}}{\gamma} \max_{x \in \mathbb{R}^{d_{h-1}} : \|x\|_2 = 1} \sum_{o_h, a_h} \left\| \hat{\mathbf{M}}_h^k(o_h, a_h) x \right\|_1 \pi(a_h|o_h, j_0) + \frac{\sqrt{\lambda}}{\gamma} \max_{x \in \mathbb{R}^{d_{h-1}} : \|x\|_2 = 1} \|x\|_1
\end{aligned}$$

$$\overset{(c)}{\leq} \frac{2Q_A\sqrt{d\lambda}}{\gamma^2},$$

where $(a)$ follows because $|a+b| \leq |a| + |b|$, $(b)$ follows from Assumption 1, and $(c)$ follows from Proposition 2 and because $\max_{x \in \mathbb{R}^{d_{h-1}}:\|x\|_2=1} \|x\|_1 \leq \sqrt{d}$.

We next upper bound the second term $I_3$ as follows.

$$I_3 \leq \sum_{\tau_{h-1} \in \mathcal{D}_{h-1}^k} \left( \sum_{\omega_{h-1}} \left| \mathbf{m}^*(\omega_h)^\top \left( \hat{\mathbf{M}}_h^k(o_h, a_h) - \mathbf{M}_h^*(o_h, a_h) \right) \bar{\bar{\psi}}^k(\tau_{h-1}) \right| \pi(\omega_{h-1}|j_0) \right)^2$$

$$\overset{(a)}{\leq} \sum_{\tau_{h-1} \in \mathcal{D}_{h-1}^k} \left( \sum_{\omega_{h-1}} \left| \mathbf{m}^*(\omega_h)^\top \left( \hat{\mathbf{M}}_h^k(o_h, a_h)\bar{\bar{\psi}}^k(\tau_{h-1}) - \mathbf{M}_h^*(o_h, a_h)\bar{\psi}^*(\tau_{h-1}) \right) \right| \pi(\omega_{h-1}|j_0) \right.$$

$$\left. + \sum_{\omega_{h-1}} \left| \mathbf{m}^*(\omega_h)^\top \mathbf{M}_h^*(o_h, a_h) \left( \bar{\bar{\psi}}^{(k)}(\tau_{h-1}) - \bar{\psi}^*(\tau_{h-1}) \right) \right| \pi(\omega_{h-1}|j_0) \right)^2$$

$$\overset{(b)}{\leq} \sum_{\tau_{h-1} \in \mathcal{D}_{h-1}^k} \left( \frac{1}{\gamma} \sum_{o_h, a_h} \left\| \mathbb{P}_{\hat{\theta}^k}(o_h|\tau_{h-1})\bar{\bar{\psi}}_h^k(\tau_h) - \mathbb{P}_\theta(o_h|\tau_{h-1})\bar{\psi}_h^*(\tau_h) \right\|_1 \pi(a_h|o_h, j_0) \right.$$

$$\left. + \frac{1}{\gamma} \sum_{\tau_{h-1} \in \mathcal{D}_{h-1}^\tau} \left\| \bar{\bar{\psi}}_{h-1}^k(\tau_{h-1}) - \bar{\psi}_{h-1}^*(\tau_{h-1}) \right\|_1 \right)^2$$

$$\overset{(c)}{=} \frac{1}{\gamma^2} \sum_{\tau_{h-1} \in \mathcal{D}_{h-1}^k} \left( \sum_{o_h, a_h} \sum_{\ell=1}^{|\mathcal{Q}_h|} \left| \mathbb{P}_{\hat{\theta}^k}(\mathbf{o}_h^\ell, o_h|\tau_{h-1}, a_h, \mathbf{a}_h^\ell) - \mathbb{P}_{\theta^*}(\mathbf{o}_h^\ell, o_h|\tau_{h-1}, a_h, \mathbf{a}_h^\ell) \right| \pi(a_h|o_h, j_0) \right.$$

$$\left. + \sum_{\ell=1}^{|\mathcal{Q}_{h-1}|} \left| \mathbb{P}_{\hat{\theta}^k}(\mathbf{o}_{h-1}^\ell|\tau_{h-1}, \mathbf{a}_{h-1}^\ell) - \mathbb{P}_{\theta^*}(\mathbf{o}_{h-1}^\ell|\tau_{h-1}, \mathbf{a}_{h-1}^\ell) \right| \right)^2$$

$$\leq \frac{1}{\gamma^2} \sum_{\tau_{h-1} \in \mathcal{D}_{h-1}^k} \left( \sum_{\mathbf{a}_{h-1} \in \mathcal{Q}_h^{\exp}} \sum_{\omega_{h-1}^o} \left| \mathbb{P}_{\hat{\theta}^k}(\omega_{h-1}^o|\tau_{h-1}, \mathbf{a}_{h-1}) - \mathbb{P}_{\theta^*}(\omega_{h-1}^o|\tau_{h-1}, \mathbf{a}_{h-1}) \right| \right)^2$$

$$\leq \frac{4|\mathcal{A}|^2 Q_A^2}{\gamma^2} \sum_{\tau_{h-1} \in \mathcal{D}_{h-1}^k} \mathrm{D}_{\mathrm{TV}}^2 \left( \mathbb{P}_{\hat{\theta}^k}^{\mathbf{u}_{\mathcal{Q}_{h-1}^{\exp}}}(\omega_{h-1}|\tau_{h-1}), \mathbb{P}_{\theta^*}^{\mathbf{u}_{\mathcal{Q}_{h-1}^{\exp}}}(\omega_{h-1}|\tau_{h-1}) \right),$$

where $(a)$ follows because $|a+b| \leq |a| + |b|$, $(b)$ follows from Equation (8) and Assumption 1, and $(c)$ follows from the physical meaning of the prediction feature, i.e. $[\bar{\psi}(\tau_h)]_\ell = \mathbb{P}_\theta(\mathbf{o}_h^\ell|\tau_h, \mathbf{a}_h^\ell)$.

By combining the upper bounds for $I_2$ and $I_3$, we conclude that

$$I_1 \leq \frac{4\lambda Q_A^2 d}{\gamma^4} + \frac{4|\mathcal{A}|^2 Q_A^2}{\gamma^2} \sum_{\tau_{h-1} \in \mathcal{D}_{h-1}^k} \mathrm{D}_{\mathrm{TV}}^2 \left( \mathbb{P}_{\hat{\theta}^k}^{\mathbf{u}_{\mathcal{Q}_{h-1}^{\exp}}}(\omega_{h-1}|\tau_{h-1}), \mathbb{P}_{\theta^*}^{\mathbf{u}_{\mathcal{Q}_{h-1}^{\exp}}}(\omega_{h-1}|\tau_{h-1}) \right) = \alpha_{h-1}^k,$$

which completes the proof. ∎

The following lemma validates that $V_{\hat{\theta}^k, \hat{b}^k}^\pi$ is an upper bound on the total variation distance between the estimated model $\hat{\theta}^k$ and the true model $\theta^*$.

**Lemma 3** *Under event $\mathcal{E}$, for any $\pi$, we have*

$$\mathrm{D}_{\mathrm{TV}} \left( \mathbb{P}_{\hat{\theta}^k}^\pi(\tau_H), \mathbb{P}_{\theta^*}^\pi(\tau_H) \right) \leq \alpha \mathbb{E}_{\tau_H \sim \mathbb{P}_{\hat{\theta}^k}^\pi} \left[ \sqrt{\sum_{h=0}^{H-1} \left\| \bar{\psi}^k(\tau_h) \right\|_{(\hat{U}_h^k)^{-1}}^2} \right], \tag{10}$$

*where*

$$\alpha^2 = \frac{4\lambda H Q_A^2 d}{\gamma^4} + \frac{28|\mathcal{A}|^2 Q_A^2 \beta}{\gamma^2}.$$

**Proof:** We proceed the proof as follows:

$$\sum_h (\alpha_{h-1}^k)^2 \leq \frac{4\lambda H Q_A^2 d}{\gamma^4} + \frac{4|\mathcal{A}|^2 Q_A^2}{\gamma^2} \sum_h \sum_{(\tau_{h-1},\pi)\in\mathcal{D}_{h-1}^k} \mathrm{D}_{\mathrm{TV}}^2 \left( \mathbb{P}_{\hat{\theta}^k}^\pi(\omega_{h-1}|\tau_{h-1}), \mathbb{P}_{\theta^*}^\pi(\omega_{h-1}|\tau_{h-1}) \right)$$

$$\overset{(a)}{\leq} \frac{4\lambda H Q_A^2 d}{\gamma^4} + \frac{28\beta|\mathcal{A}|^2 Q_A^2}{\gamma^2},$$

where $(a)$ follows from Lemma 1. The proof then follows directly from Proposition 1, Lemma 2 and the Cauchy's inequality. ∎

Note that the reward function $R$ is within $[0, 1]$. We hence obtain the following corollary.

**Corollary 2 (UCB)** *Under event $\mathcal{E}$, for any $k \in [K]$ and any reward $R$, we have*

$$\left| V_{\hat{\theta}^k, R}^\pi - V_{\theta^*, R}^\pi \right| \leq V_{\hat{\theta}^k, \hat{b}^k},$$

*where $\hat{b}^k(\tau_H) = \min \left\{ \alpha \sqrt{\sum_h \left\| \bar{\hat{\psi}}_h^k(\tau_h) \right\|_{(\hat{U}_h^k)^{-1}}^2}, 1 \right\}.$*

## C.3 Step 3: Sublinear Summation

To prove that $\sum_k V_{\hat{\theta}^k, \hat{b}^k}^{\pi^k}$ is sublinear, i.e., scales as $O(\sqrt{K})$, we first prove the following lemma that relates the estimated feature and the ground-truth feature via the total variation distance between the estimated model and the true model.

**Lemma 4** *Under the event $\mathcal{E}$, for any $k \in [K]$ and any policy $\pi$, we have*

$$\mathbb{E}_{\tau_H \sim \mathbb{P}_{\theta^*}^\pi} \left[ \sqrt{\sum_{h=0}^{H-1} \left\| \bar{\hat{\psi}}^k(\tau_h) \right\|_{(\hat{U}_h^k)^{-1}}^2} \right]$$

$$\leq \left( 1 + \frac{2|\mathcal{A}|Q_A\sqrt{7r\beta}}{\sqrt{\lambda}} \right) \sum_{h=0}^{H-1} \mathbb{E}_{\tau_h \sim \mathbb{P}_{\theta^*}^\pi} \left[ \left\| \bar{\psi}^*(\tau_h) \right\|_{(U_h^k)^{-1}} \right] + \frac{2HQ_A}{\sqrt{\lambda}} \mathrm{D}_{\mathrm{TV}} \left( \mathbb{P}_{\theta^*}^\pi(\tau_H), \mathbb{P}_{\hat{\theta}^k}^\pi(\tau_H) \right).$$

**Proof:** Recall that

$$\hat{U}_h^k = \lambda I + \sum_{\tau \in \mathcal{D}_h^k} \bar{\hat{\psi}}^k(\tau_h) \bar{\hat{\psi}}^k(\tau_h)^\top.$$

We define the ground-truth counterpart of $\hat{U}_h^k$ as follows:

$$U_h^k = \lambda I + \sum_{\tau \in \mathcal{D}_h^k} \bar{\psi}^*(\tau_h) \bar{\psi}^*(\tau_h)^\top.$$

Then, following from Lemma 13, we have

$$\sqrt{\sum_{h=0}^{H-1} \left\| \bar{\hat{\psi}}^k(\tau_h) \right\|_{(\hat{U}_h^k)^{-1}}^2}$$

$$\leq \sum_{h=0}^{H-1} \left\| \bar{\hat{\psi}}^k(\tau_h) \right\|_{(\hat{U}_h^k)^{-1}}$$

$$\leq \frac{1}{\sqrt{\lambda}} \sum_{h=0}^{H-1} \left\| \bar{\hat{\psi}}^k(\tau_h) - \bar{\psi}^*(\tau_h) \right\|_2 + \sum_{h=0}^{H-1} \left( 1 + \frac{\sqrt{r}\sqrt{\sum_{\tau_h \in \mathcal{D}_h^k} \left\| \bar{\hat{\psi}}^k(\tau_h) - \bar{\psi}^*(\tau_h) \right\|_2^2}}{\sqrt{\lambda}} \right) \left\| \bar{\psi}^*(\tau_h) \right\|_{(U_h^k)^{-1}}.$$

Furthermore, note that

$$\left\| \bar{\hat{\psi}}^k(\tau_h) - \bar{\psi}^*(\tau_h) \right\|_2 \leq \left\| \bar{\hat{\psi}}^k(\tau_h) - \bar{\psi}^*(\tau_h) \right\|_1$$
$$\overset{(a)}{\leq} 2|\mathcal{A}|Q_A \mathsf{D}_{\mathsf{TV}} \left( \mathbb{P}_{\hat{\theta}^k}^{\mathbf{u} \mathcal{Q}_h^{\exp}}(\omega_h | \tau_h), \mathbb{P}_{\theta^*}^{\mathbf{u} \mathcal{Q}_h^{\exp}}(\omega_h | \tau_h) \right),$$

where $(a)$ follows from the physical meaning of the prediction feature.

Following from Lemma 1, we conclude that

$$\sqrt{\sum_{h=0}^{H-1} \left\| \bar{\hat{\psi}}^k(\tau_h) \right\|_{(\hat{U}_h^k)^{-1}}^2}$$
$$\leq \frac{1}{\sqrt{\lambda}} \sum_{h=0}^{H-1} \left\| \bar{\hat{\psi}}^k(\tau_h) - \bar{\psi}^*(\tau_h) \right\|_2 + \left( 1 + \frac{2|\mathcal{A}|Q_A\sqrt{7r\beta}}{\sqrt{\lambda}} \right) \sum_{h=0}^{H-1} \left\| \bar{\psi}^*(\tau_h) \right\|_{(U_h^k)^{-1}}.$$

For the first term, taking expectation, we have

$$\sum_{h=0}^{H-1} \mathbb{E}_{\tau_h \sim \mathbb{P}_{\theta^*}^{\pi}} \left[ \left\| \bar{\hat{\psi}}^k(\tau_h) - \bar{\psi}^*(\tau_h) \right\|_1 \right]$$
$$\leq \sum_{h=0}^{H-1} \sum_{\tau_h} \left( \left\| \bar{\hat{\psi}}^k(\tau_h) \left( \mathbb{P}_{\theta}^{\pi}(\tau_h) - \mathbb{P}_{\hat{\theta}^k}^{\pi}(\tau_h) \right) + \bar{\hat{\psi}}^k(\tau_h)\mathbb{P}_{\hat{\theta}^k}^{\pi}(\tau_h) - \bar{\psi}^*(\tau_h)\mathbb{P}_{\theta^*}^{\pi}(\tau_h) \right\|_1 \right)$$
$$\leq \sum_{h=0}^{H-1} \sum_{\tau_h} \left( \left\| \bar{\hat{\psi}}^k(\tau_h) \right\|_1 \left| \mathbb{P}_{\theta}^{\pi}(\tau_h) - \mathbb{P}_{\hat{\theta}^k}^{\pi}(\tau_h) \right| + \left\| \hat{\psi}^k(\tau_h) - \psi^*(\tau_h) \right\|_1 \pi(\tau_h) \right)$$
$$\overset{(a)}{\leq} 2 \sum_{h=0}^{H-1} Q_A \mathsf{D}_{\mathsf{TV}} \left( \mathbb{P}_{\theta^*}^{\pi}(\tau_h), \mathbb{P}_{\hat{\theta}^k}^{\pi}(\tau_h) \right)$$
$$\leq 2HQ_A \mathsf{D}_{\mathsf{TV}} \left( \mathbb{P}_{\theta^*}^{\pi}(\tau_H), \mathbb{P}_{\hat{\theta}^k}^{\pi}(\tau_H) \right),$$

where $(a)$ follows from $\|\bar{\hat{\psi}}^k(\tau_h)\|_1 \leq |\mathcal{Q}_h^A| \leq Q_A$, and the physical meaning of $\psi(\tau_h)$.

Thus,

$$\mathbb{E}_{\tau_H \sim \mathbb{P}_{\theta^*}^{\pi}} \left[ \sqrt{\sum_{h=0}^{H-1} \left\| \bar{\hat{\psi}}^k(\tau_h) \right\|_{(\hat{U}_h^k)^{-1}}^2} \right]$$
$$\leq \left( 1 + \frac{2|\mathcal{A}|Q_A\sqrt{7r\beta}}{\sqrt{\lambda}} \right) \sum_{h=0}^{H-1} \mathbb{E}_{\tau_h \sim \mathbb{P}_{\theta^*}^{\pi}} \left[ \left\| \bar{\psi}^*(\tau_h) \right\|_{(U_h^k)^{-1}} \right] + \frac{2HQ_A}{\sqrt{\lambda}} \mathsf{D}_{\mathsf{TV}} \left( \mathbb{P}_{\theta^*}^{\pi}(\tau_H), \mathbb{P}_{\hat{\theta}^k}^{\pi}(\tau_H) \right).$$

∎

The following lemma can be proved via the $\ell_2$ Eluder argument (Chen et al., 2022; Zhong et al., 2022). Since we have a slightly different estimation oracle and guarantee, we provide the full proof here for completeness.

**Lemma 5** *Under event $\mathcal{E}$, for any $h \in \{0, \ldots, H-1\}$, we have*

$$\sum_k \mathsf{D}_{\mathsf{TV}} \left( \mathbb{P}_{\theta^*}^{\pi^k}(\tau_h), \mathbb{P}_{\hat{\theta}^k}^{\pi^k}(\tau_h) \right) \lesssim \frac{|\mathcal{A}|Q_A\sqrt{\beta}}{\gamma} \sqrt{rHK \log(1 + dQ_A K/\gamma^4)}.$$

*Here, $a \lesssim b$ indicates that there is an absolute positive constant $c$ such that $a \leq c \cdot b$.*

**Proof:** First, by the first inequality in Proposition 1, we have

$$D_{TV}\left(\mathbb{P}_{\theta^*}^{\pi^k}(\tau_h), \mathbb{P}_{\hat{\theta}^k}^{\pi^k}(\tau_h)\right) \le \sum_h \sum_{\tau_H} \left|\hat{\mathbf{m}}^k(\omega_h)^\top \left(\hat{\mathbf{M}}_h^k(o_h, a_h) - \mathbf{M}_h^*(o_h, a_h)\right)\psi^*(\tau_{h-1})\right| \pi^k(\tau_H).$$

It suffices to upper bound $\sum_{\tau_H} \left|\hat{\mathbf{m}}^k(\omega_h)^\top \left(\hat{\mathbf{M}}_h^k(o_h, a_h) - \mathbf{M}_h^*(o_h, a_h)\right)\psi^*(\tau_{h-1})\right| \pi(\tau_H)$ for any policy $\pi$. For simplicity, we use similar notations as in Lemma 2. We index the future trajectory $\omega_{h-1} = (o_h, a_h, \ldots, o_H, a_H)$ by $i$, and the history trajectory $\tau_{h-1}$ by $j$. We represent $\bar{\psi}^*(\tau_{h-1})$ as $x_j$, and $\hat{\mathbf{m}}^k(\omega_h)^\top \left(\hat{\mathbf{M}}_h^k(o_h, a_h) - \mathbf{M}_h^*(o_h, a_h)\right)$ as $w_i$. We also denote $\pi(\omega_{h-1}|\tau_{h-1})$ by $\pi_{i|j}$.

Let $\lambda_0$ be a constant determined later and define the matrix

$$\Lambda_h^k = \lambda_0 I + \sum_{t<k} \mathbb{E}_{j \sim \mathbb{P}_{\theta^*}^{\pi^t}}\left[x_j x_j^\top\right].$$

Then, for any $\pi$, we have

$$\sum_{\tau_H} \left|\hat{\mathbf{m}}^k(\omega_h)^\top \left(\hat{\mathbf{M}}_h^k(o_h, a_h) - \mathbf{M}_h^*(o_h, a_h)\right)\psi^*(\tau_{h-1})\right| \pi(\tau_H)$$

$$= \mathbb{E}_{j \sim \mathbb{P}_{\theta^*}^{\pi^k}}\left[\sum_i |\pi_{i|j} w_i^\top x_j|\right]$$

$$= \mathbb{E}_{j \sim \mathbb{P}_{\theta^*}^{\pi^k}}\left[\left(\sum_i \pi_{i|j}\,\mathrm{sgn}(w_i^\top x_j)w_i\right)^\top x_j\right]$$

$$\overset{(a)}{\le} \mathbb{E}_{j \sim \mathbb{P}_{\theta^*}^{\pi^k}}\left[\|x_j\|_{\Lambda_h^\dagger}\left\|\sum_i \pi_{i|j}\,\mathrm{sgn}(w_i^\top x_j)w_i\right\|_{\Lambda_h}\right],$$

where $(a)$ follows from the Cauchy's inequality.

Now we fix $j = j_0$ and consider the following term.

$$\left\|\sum_i \pi_{i|j_0}\,\mathrm{sgn}(w_i^\top x_{j_0})w_i\right\|_{\Lambda_h}^2$$

$$= \underbrace{\lambda_0 \left\|\sum_i \pi_{i|j_0}\,\mathrm{sgn}(w_i^\top x_{j_0})w_i\right\|_2^2}_{I_1} + \underbrace{\sum_{t<k}\mathbb{E}_{j \sim \mathbb{P}_{\theta^*}^{\pi^t}}\left[\left(\sum_i \pi_{i|j_0}\,\mathrm{sgn}(w_i^\top x_{j_0})w_i^\top x_j\right)^2\right]}_{I_2}$$

For the first term $I_1$, we have

$$\sqrt{I_1} = \sqrt{\lambda_0}\max_{x \in \mathbb{R}^{d_{h-1}}:\|x\|_2=1}\left|\sum_i \pi_{i|j_0}\,\mathrm{sgn}(w_i^\top x_{j_0})w_i^\top x\right|$$

$$\le \sqrt{\lambda_0}\max_{x \in \mathbb{R}^{d_{h-1}}:\|x\|_2=1}\sum_{\omega_{h-1}} \pi(\omega_{h-1}|j_0)\left|\hat{\mathbf{m}}^k(\omega_{h-1})^\top x\right|$$

$$\quad + \sqrt{\lambda_0}\max_{x \in \mathbb{R}^{d_{h-1}}:\|x\|_2=1}\sum_{\omega_{h-1}} \pi(\omega_{h-1}|j_0)\left|\hat{\mathbf{m}}^k(\omega_h)^\top \mathbf{M}_h^*(o_h, a_h)x\right|$$

$$\overset{(a)}{\le} \frac{\sqrt{d\lambda_0}}{\gamma} + \frac{Q_A\sqrt{d\lambda_0}}{\gamma^2}$$

$$\le \frac{2Q_A\sqrt{d\lambda_0}}{\gamma^2},$$

where $(a)$ follows from Assumption 1, Proposition 2, and the fact that $\max_{x \in \mathbb{R}^{d_{h-1}}:\|x\|_2=1}\|x\|_1 \le \sqrt{d_{h-1}} \le \sqrt{d}$.

For the second term $I_2$, we have

$$I_2 \leq \sum_{t<k} \mathop{\mathbb{E}}_{\tau_{h-1} \sim \mathbb{P}_{\theta^*}^{\pi^t}} \left[ \left( \sum_{\omega_{h-1}} \pi(\omega_{h-1}|j_0) \left| \hat{\mathbf{m}}^k(\omega_h)^\top \left( \hat{\mathbf{M}}_h^k(o_h, a_h) - \mathbf{M}_h^*(o_h, a_h) \right) \bar{\psi}^*(\tau_{h-1}) \right| \right)^2 \right]$$

$$\leq \sum_{t<k} \mathop{\mathbb{E}}_{\tau_{h-1} \sim \mathbb{P}_{\theta^*}^{\pi^t}} \left[ \left( \sum_{\omega_{h-1}} \pi(\omega_{h-1}|j_0) \left| \hat{\mathbf{m}}^k(\omega_{h-1})^\top \left( \bar{\psi}^*(\tau_{h-1}) - \bar{\hat{\psi}}^k(\tau_{h-1}) \right) \right| \right. \right.$$

$$\left. \left. + \sum_{\omega_{h-1}} \pi(\omega_{h-1}|j_0) \left| \hat{\mathbf{m}}^k(\omega_h)^\top \left( \hat{\mathbf{M}}_h^k(o_h, a_h)\bar{\hat{\psi}}^k(\tau_{h-1}) - \mathbf{M}_h^*(o_h, a_h)\bar{\psi}^*(\tau_{h-1}) \right) \right| \right)^2 \right]$$

$$\overset{(a)}{\leq} \sum_{t<k} \mathop{\mathbb{E}}_{\tau_{h-1} \sim \mathbb{P}_{\theta^*}^{\pi^t}} \left[ \left( \frac{1}{\gamma} \left\| \bar{\psi}^*(\tau_{h-1}) - \bar{\hat{\psi}}^k(\tau_{h-1}) \right\|_1 \right. \right.$$

$$\left. \left. + \frac{1}{\gamma} \sum_{o_h, a_h} \pi^k(a_h|o_h, j_0) \left\| \hat{\mathbf{M}}^k(o_h, a_h)\bar{\hat{\psi}}^k(\tau_{h-1}) - \mathbf{M}_h^*(o_h, a_h)\bar{\psi}^*(\tau_{h-1}) \right\|_1 \right)^2 \right]$$

$$= \frac{1}{\gamma^2} \sum_{t<k} \mathop{\mathbb{E}}_{\tau_{h-1} \sim \mathbb{P}_{\theta^*}^{\pi^t}} \left[ \left( \sum_{\ell=1}^{|\mathcal{Q}_{h-1}|} \left| \mathbb{P}_{\hat{\theta}^k}(\mathbf{o}_{h-1}^\ell|\tau_{h-1}, \mathbf{a}_{h-1}^\ell) - \mathbb{P}_{\theta^*}(\mathbf{o}_{h-1}^\ell|\tau_{h-1}, \mathbf{a}_{h-1}^\ell) \right| \right. \right.$$

$$\left. \left. + \sum_{o_h, a_h} \sum_{\ell=1}^{|\mathcal{Q}_h|} \pi^k(a_h|o_h, j_0) \left| \mathbb{P}_{\hat{\theta}^k}(\mathbf{o}_h^\ell, o_h|\tau_{h-1}, a_h, \mathbf{a}_h^\ell) - \mathbb{P}_{\theta^*}(\mathbf{o}_h^\ell, o_h|\tau_{h-1}, a_h, \mathbf{a}_h^\ell) \right| \right)^2 \right]$$

$$\leq \frac{|\mathcal{Q}_{h-1}^{\exp}|^2}{\gamma^2} \sum_{t<k} \mathop{\mathbb{E}}_{\tau_{h-1} \sim \mathbb{P}_{\theta^*}^{\pi^t}} \left[ \mathrm{D}_{\mathrm{TV}}^2 \left( \mathbb{P}_{\hat{\theta}^k}^{\mathbf{u}_{\mathcal{Q}_{h-1}^{\exp}}}(\omega_{h-1}|\tau_{h-1}), \mathbb{P}_{\theta^*}^{\mathbf{u}_{\mathcal{Q}_{h-1}^{\exp}}}(\omega_{h-1}|\tau_{h-1}) \right) \right]$$

$$\overset{(b)}{\leq} \frac{4|\mathcal{A}|^2 Q_A^2}{\gamma^2} \sum_{t<k} \mathrm{D}_{\mathrm{H}}^2 \left( \mathbb{P}_{\hat{\theta}^k}^{\nu_h(\pi^t, \mathbf{u}_{\mathcal{Q}_{h-1}^{\exp}})}(\tau_H), \mathbb{P}_{\theta^*}^{\nu_h(\pi^t, \mathbf{u}_{\mathcal{Q}_{h-1}^{\exp}})}(\tau_H) \right),$$

where $(a)$ follows from Assumption 1, and $(b)$ follows because $|\mathcal{Q}_{h-1}^{\exp}| = |\mathcal{A}||\mathcal{Q}_h^A| + |\mathcal{Q}_{h-1}^A| \leq 2|\mathcal{A}|Q_A$.

Thus, we have

$$\sum_{\tau_H} \left| \hat{\mathbf{m}}^k(\omega_h)^\top \left( \hat{\mathbf{M}}_h^k(o_h, a_h) - \mathbf{M}_h^*(o_h, a_h) \right) \psi^*(\tau_{h-1}) \right| \pi(\tau_H)$$

$$\leq \mathop{\mathbb{E}}_{\tau_{h-1} \sim \mathbb{P}_{\theta^*}^{\pi}} \left[ \tilde{\alpha}_{h-1}^k \left\| \bar{\psi}^*(\tau_{h-1}) \right\|_{\Lambda_{h-1}^\dagger} \right],$$

where

$$(\tilde{\alpha}_{h-1}^k)^2 = \frac{4\lambda_0 Q_A^2 d}{\gamma^4} + \frac{4|\mathcal{A}|^2 Q_A^2}{\gamma^2} \sum_{t<k} \mathrm{D}_{\mathrm{H}}^2 \left( \mathbb{P}_{\hat{\theta}^k}^{\nu_h(\pi^t, \mathbf{u}_{\mathcal{Q}_{h-1}^{\exp}})}(\tau_H), \mathbb{P}_{\theta^*}^{\nu_h(\pi^t, \mathbf{u}_{\mathcal{Q}_{h-1}^{\exp}})}(\tau_H) \right).$$

By choosing $\lambda_0 = \frac{\gamma^4}{4Q_A^2 d}$, and recalling Lemma 1, we further have

$$\sum_h (\tilde{\alpha}_{h-1}^k)^2 \leq \frac{28|\mathcal{A}|^2 Q_A^2 \beta}{\gamma^2} \triangleq \tilde{\alpha}.$$

Hence, with the Cauchy's inequality, we have

$$\mathrm{D}_{\mathrm{TV}} \left( \mathbb{P}_{\hat{\theta}^k}^{\pi^k}(\tau_H), \mathbb{P}_{\theta}^{\pi^k}(\tau_H) \right) \leq \min \left\{ \tilde{\alpha} \sqrt{\sum_{h=1}^{H} \mathop{\mathbb{E}}_{\tau_{h-1} \sim \mathbb{P}_{\theta^*}^{\pi^k}} \left[ \left\| \bar{\psi}^*(\tau_{h-1}) \right\|_{\Lambda_{h-1}^{-1}}^2 \right]}, 2 \right\}.$$

Taking the summation and applying the elliptical potential lemma (i.e., Lemma 14), we have

$$\sum_{k=1}^{K} \mathsf{D}_{\mathsf{TV}} \left( \mathbb{P}_{\hat{\theta}^k}^{\pi^k}(\tau_H), \mathbb{P}_{\theta^*}^{\pi^k}(\tau_H) \right)$$

$$\leq \sqrt{K} \sqrt{\sum_{k=1}^{K} \sum_{h=1}^{H} \min \left\{ \tilde{\alpha} \mathop{\mathbb{E}}_{\tau_{h-1} \sim \mathbb{P}_{\theta^*}^{\pi^k}} \left[ \left\| \bar{\psi}^*(\tau_{h-1}) \right\|_{\Lambda_{h-1}^{\dagger}}^{2} \right], 4 \right\}}$$

$$\lesssim \frac{|\mathcal{A}| Q_A}{\gamma} \sqrt{r H K \beta \log(1 + d Q_A K / \gamma)},$$

which yields the final result. ∎

The next lemma shows that the summation $\sum_{k=1}^{K} V_{\hat{\theta}^k, \hat{b}^k}^{\pi^k}$ grows sublinearly in $K$.

**Lemma 6** *Under the event $\mathcal{E}$, with probability at least $1 - \delta$, we have*

$$\sum_{k=1}^{K} V_{\hat{\theta}^k, \hat{b}^k}^{\pi^k} \lesssim \left( \sqrt{r} + \frac{Q_A \sqrt{H}}{\gamma} \right) \frac{|\mathcal{A}| Q_A^2 H \sqrt{d r H \beta K \beta_0}}{\gamma^2},$$

*where $\beta_0 = \max\{\log(1 + K/\lambda), \log(1 + d Q_A K / \gamma)\}$, and $\lambda = \frac{\gamma |\mathcal{A}|^2 Q_A \beta \max\{\sqrt{r}, Q_A \sqrt{H}/\gamma\}}{\sqrt{dH}}$.*

**Proof:** First, since $\hat{b}^k(\tau_H) \in [0, 1]$, we have

$$V_{\hat{\theta}^k, \hat{b}^k}^{\pi^k} \leq V_{\theta^*, \hat{b}^k}^{\pi^k} + \frac{1}{2} \mathsf{D}_{\mathsf{TV}} \left( \mathbb{P}_{\hat{\theta}^k}^{\pi^k}, \mathbb{P}_{\theta^*}^{\pi^k} \right).$$

Then, following from Lemma 4, we have

$$\sum_{k=1}^{K} V_{\theta^*, \hat{b}^k}^{\pi^k}$$

$$\leq \sum_{k=1}^{K} \min \left\{ \alpha \left( 1 + \frac{2|\mathcal{A}| Q_A \sqrt{7r\beta}}{\sqrt{\lambda}} \right) \sum_{h=0}^{H-1} \mathop{\mathbb{E}}_{\tau_h \sim \mathbb{P}_{\theta^*}^{\pi^k}} \left[ \left\| \bar{\psi}^*(\tau_h) \right\|_{(U_h^k)^{-1}} \right] + \sum_{k=1}^{K} \frac{\alpha H Q_A}{\sqrt{\lambda}} \mathsf{D}_{\mathsf{TV}} \left( \mathbb{P}_{\theta^*}^{\pi^k}(\tau_H), \mathbb{P}_{\hat{\theta}^k}^{\pi^k}(\tau_H) \right), 1 \right\}$$

$$\leq \underbrace{\sum_{k=1}^{K} \min \left\{ \alpha \left( 1 + \frac{2|\mathcal{A}| Q_A \sqrt{7r\beta}}{\sqrt{\lambda}} \right) \sum_{h=0}^{H-1} \mathop{\mathbb{E}}_{\tau_h \sim \mathbb{P}_{\theta^*}^{\pi^k}} \left[ \left\| \bar{\psi}^*(\tau_h) \right\|_{(U_h^k)^{-1}} \right], 1 \right\}}_{I_1} + \sum_{k=1}^{K} \frac{\alpha H Q_A}{\sqrt{\lambda}} \mathsf{D}_{\mathsf{TV}} \left( \mathbb{P}_{\theta^*}^{\pi^k}(\tau_H), \mathbb{P}_{\hat{\theta}^k}^{\pi^k}(\tau_H) \right).$$

We next analyze the term $I_1$. Recall that $U_h^k = \lambda I + \sum_{\tau_h \in \mathcal{D}_h^k} \bar{\psi}^*(\tau_h) \bar{\psi}^*(\tau_h)^\top$. Further note that

$$\left\{ \mathop{\mathbb{E}}_{\tau_h \sim \mathbb{P}_{\theta^*}^{\pi^k}} \left[ \left\| \bar{\psi}^*(\tau_h) \right\|_{(U_h^k)^{-1}} \right] - \left\| \bar{\psi}^*(\tau_h^{k+1, h+1}) \right\|_{(U_h^k)^{-1}} \right\}_{k=1}^{K}$$

forms a martingale. Applying the Azuma-Hoeffding inequality (see Lemma 11), we have, with probability at least $1 - \delta$,

$$I_1 \leq \sqrt{2K \log(2/\delta)} + \sum_{k=1}^{K} \min \left\{ \alpha \left( 1 + \frac{2|\mathcal{A}| Q_A \sqrt{7r\beta}}{\sqrt{\lambda}} \right) \sum_{h=0}^{H-1} \left\| \bar{\psi}^*(\tau_h^{k+1, h+1}) \right\|_{(U_h^k)^{-1}}, 1 \right\}$$

$$\overset{(a)}{\lesssim} \sqrt{2K \log(2/\delta)} + \alpha \left( 1 + \frac{2|\mathcal{A}| Q_A \sqrt{7r\beta}}{\sqrt{\lambda}} \right) H \sqrt{r K \log(1 + K/\lambda)}$$

$$\lesssim \alpha \left( 1 + \frac{|\mathcal{A}| Q_A \sqrt{r\beta}}{\sqrt{\lambda}} \right) H \sqrt{r K \log(1 + K/\lambda)},$$

where $(a)$ follows from Lemma 14.

Let $\beta := \max\{\log(1 + K/\lambda), \log(1 + dQ_A K/\gamma)\}$ for simplicity. By Lemma 5, we have,

$$\sum_k V_{\hat{\theta}^k, \hat{b}^k}^{\pi^k}$$

$$\lesssim \alpha \left(1 + \frac{|\mathcal{A}|Q_A\sqrt{r\beta}}{\sqrt{\lambda}}\right) H\sqrt{rK\beta_0} + \frac{\alpha H}{\sqrt{\lambda}} \frac{|\mathcal{A}|Q_A^2\sqrt{\beta}}{\gamma} \sqrt{rHK\beta_0}$$

$$\leq \alpha \left(1 + \frac{|\mathcal{A}|Q_A\sqrt{r\beta}}{\sqrt{\lambda}} + \frac{|\mathcal{A}|Q_A^2\sqrt{\beta H}}{\gamma\sqrt{\lambda}}\right) H\sqrt{rK\beta_0}$$

$$\lesssim \left(\frac{|\mathcal{A}|Q_A\sqrt{\beta}}{\gamma} + \frac{Q_A\sqrt{dH}}{\gamma^2}\sqrt{\lambda}\right) \left(1 + \frac{|\mathcal{A}|Q_A\sqrt{\beta}\max\{\sqrt{r}, Q_A\sqrt{H}/\gamma\}}{\sqrt{\lambda}}\right) H\sqrt{rK\beta_0}$$

$$= \left(1 + \frac{\sqrt{dH}}{|\mathcal{A}|\sqrt{\beta}\gamma}\sqrt{\lambda}\right) \left(1 + \frac{|\mathcal{A}|Q_A\sqrt{\beta}\max\{\sqrt{r}, Q_A\sqrt{H}/\gamma\}}{\sqrt{\lambda}}\right) \frac{|\mathcal{A}|Q_A H\sqrt{r\beta K\beta_0}}{\gamma}$$

$$\overset{(a)}{\lesssim} \left(1 + \frac{Q_A\sqrt{dH}\max\{\sqrt{r}, Q_A\sqrt{H}/\gamma\}}{\gamma}\right) \frac{|\mathcal{A}|Q_A H\sqrt{r\beta K\beta_0}}{\gamma}$$

$$\leq \left(\sqrt{r} + \frac{Q_A\sqrt{H}}{\gamma}\right) \frac{|\mathcal{A}|Q_A^2 H\sqrt{drH\beta K\beta_0}}{\gamma^2},$$

where $(a)$ follows by choosing

$$\lambda = \frac{\gamma|\mathcal{A}|^2 Q_A\beta\max\{\sqrt{r}, Q_A\sqrt{H}/\gamma\}}{\sqrt{dH}}.$$

This completes the proof. ■

## C.4 Proof of Theorem 1

**Theorem 3 (Restatement of Theorem 1)** *Suppose Assumption 1 holds. Let $p_{\min} = O(\frac{\delta}{KH|\mathcal{O}|^H|\mathcal{A}|^H})$, $\beta = O(\log|\bar{\Theta}_\varepsilon|)$, where $\varepsilon = O(\frac{p_{\min}}{KH})$, $\lambda = \frac{\gamma|\mathcal{A}|^2 Q_A\beta\max\{\sqrt{r}, Q_A\sqrt{H}/\gamma\}}{\sqrt{dH}}$, and $\alpha = O\left(\frac{Q_A\sqrt{Hd}}{\gamma^2}\sqrt{\lambda} + \frac{|\mathcal{A}|Q_A\sqrt{\beta}}{\gamma}\right)$. Then, with probability at least $1 - \delta$, PSR-UCB outputs a model $\theta^\epsilon$ and a policy $\bar{\pi}$ that satisfy*

$$V_{\theta^*, R}^{\pi^*} - V_{\theta^*, R}^{\bar{\pi}} \leq \epsilon, \text{ and } \forall\pi, \text{ } \mathrm{D_{TV}}\left(\mathbb{P}_{\theta^\epsilon}^\pi(\tau_H), \mathbb{P}_{\theta^*}^\pi(\tau_H)\right) \leq \epsilon.$$

*In addition, PSR-UCB terminates with a sample complexity of*

$$\tilde{O}\left(\left(r + \frac{Q_A^2 H}{\gamma^2}\right) \frac{rdH^3|\mathcal{A}|^2 Q_A^4\beta}{\gamma^4\epsilon^2}\right).$$

**Proof:** The proof is under event $\mathcal{E}$, which occurs with probability at least $1 - 3\delta$.

Following from Corollary 2, if PSR-UCB terminates, we have

$$\forall\pi, \text{ } \mathrm{D_{TV}}\left(\mathbb{P}_{\theta^\epsilon}^\pi(\tau_H), \mathbb{P}_{\theta^*}^\pi(\tau_H)\right) = 2\max_R \left|V_{\theta^\epsilon, R}^\pi - V_{\theta^*, R}^\pi\right| \leq 2V_{\theta^\epsilon, \hat{b}^\epsilon}^\pi \leq \epsilon,$$

where the last inequality follows from the termination condition of PSR-UCB.

In addition,

$$V_{\theta^*, R}^{\pi^*} - V_{\theta^*, R}^{\bar{\pi}}$$

$$= V_{\theta^*, R}^{\pi^*} - V_{\theta^\epsilon, R}^{\pi^*} + V_{\theta^\epsilon, R}^{\pi^*} - V_{\theta^\epsilon, R}^{\bar{\pi}} + V_{\theta^\epsilon, R}^{\bar{\pi}} - V_{\theta^*, R}^{\bar{\pi}}$$

$$\overset{(a)}{\leq} 2\max_\pi V_{\theta^\epsilon, \hat{b}^\epsilon}^\pi \leq \epsilon,$$

where $(a)$ is due to the design of $\bar{\pi}$ and Corollary 2.

Finally, recall that Lemma 6 states that

$$\sum_{k=1}^{K} V_{\hat{\theta}^k, \hat{b}^k} \lesssim \left( \sqrt{r} + \frac{Q_A \sqrt{H}}{\gamma} \right) \frac{|\mathcal{A}| Q_A^2 H \sqrt{dr H \beta K \beta_0}}{\gamma^2}.$$

By the pigeon-hole principle and the termination condition of PSR-UCB, if

$$K = \tilde{O}\left( \left( r + \frac{Q_A^2 H}{\gamma^2} \right) \frac{r d H^2 |\mathcal{A}|^2 Q_A^4 \beta}{\gamma^4 \epsilon^2} \right),$$

PSR-UCB must terminate within $K$ episodes, implying that the sample complexity of PSR-UCB is at most

$$\tilde{O}\left( \left( r + \frac{Q_A^2 H}{\gamma^2} \right) \frac{r d H^3 |\mathcal{A}|^2 Q_A^4 \beta}{\gamma^4 \epsilon^2} \right).$$

∎

### C.5 Proof of Corollary 1 (for POMDPs)

When PSR is specialized to an $m$-step decodable POMDP, we follow the setup described in Section D in Liu et al. (2022b). In POMDPs, there exists a state space $\mathcal{S}$ such that the system dynamics proceeds as follows. When $h = 1$, the system is at the state $s_1$ and the agent observes $o_1$ that is determined by $s_1$. At any step $h \geq 1$, if the system is at a state $s_h \in \mathcal{S}$ and the agent takes an action $a_h \in \mathcal{A}$, the system transits to a state $s_{h+1}$ sampled from a *transition distribution* $\mathbf{T}_{h, a_h}(\cdot | s_h)$, and then the agent observes $o_{h+1}$ sampled from an *emission distribution* $\mathbf{O}_h(\cdot | s_{h+1})$. Notably, all states are unobservable to the agent.

To define $m$-step decodability, we introduce the $m$-step emission-action matrix $\{\mathbf{G}_h\}$ where

$$[\mathbf{G}_h]_{(\mathbf{a}, \mathbf{o}), s} = \mathbb{P}(o_h, \ldots, o_{\min\{h+m-1, H\}} = \mathbf{o} | s_h = s, a_h \ldots, a_{\min\{h+m-2, H\}} = \mathbf{a}).$$

Therefore, for $h \leq H - h$, we have $\mathbf{G}_h \in \mathbb{R}^{(|\mathcal{A}|^{m-1} |\mathcal{O}|^m) \times |\mathcal{S}|}$. To make a unified argument, we define $m$-step decodability as follows.

**Definition 3** ($m$-**step decodable POMDPs**) *A POMDP with $m$-step emmision-action matrix $\{\mathbf{G}_h\}$ is said to be $m$-step decodable if there exists an $\alpha > 0$ such that $\min_{h \leq H} \sigma_{|\mathcal{S}|}(\mathbf{G}_h) \geq \alpha$. Here $\sigma_n(\mathbf{A})$ is the $n$-th singular value of matrix $\mathbf{A}$.*

We remark that the key difference between our definition and the definition introduced in Liu et al. (2022a) is that we allow $h > H - m$. This indicates that after step $h > H - m$, the process is $(H - h)$-step decodable, which is reasonable in most real-world examples, since the closer we approach to the end, the clearer our understanding of the current state becomes. Moreover, the new definition admits more convenient notations with almost not impact on the logic of the proof.

In this section, a PSR $\theta = \{\phi_h, \mathbf{M}_h(o_h, a_h)\}$ is reparametrized by $\{\mathbf{T}_{h,a}, \mathbf{O}_h\}_{h \in [H], a \in \mathcal{A}}$. More importantly, we have the following equations and properties.

$$\begin{cases} \mathbf{Y}_h \in \arg\min_{\mathbf{Y}_h : \mathbf{Y}_h \mathbf{G}_h = 0} \|\mathbf{G}_h^\dagger + \mathbf{Y}_h\|_1 \\ \mathbf{M}_h(o_h, a_h) = \mathbf{G}_{h+1} \mathbf{T}_{h, a_h} \text{diag}(\mathbf{O}_h(o_h | \cdot))(\mathbf{G}_h^\dagger + \mathbf{Y}_h) \\ \phi_H^\top M_H(o_H, a_H) = \mathbf{e}_{(o_H, a_H)} \in \mathbb{R}^{\mathcal{O} \times \mathcal{A}} \\ \mathcal{Q}_h = (\mathcal{O} \times \mathcal{A})^{\min\{m-1, H-h\}} \\ \mathcal{Q}_h^A = \mathcal{A}^{\min\{m-1, H-h\}} \\ d \leq |\mathcal{O}|^m |\mathcal{A}|^m \\ r = |\mathcal{S}| \\ Q_A = |\mathcal{A}|^{m-1} \\ \gamma = \frac{|\mathcal{S}| + |\mathcal{A}|^{m-1}}{\alpha} = \frac{r + Q_A}{\alpha}. \end{cases} \tag{11}$$

**Corollary 3 (Restatement of Corollary 1)** *When PSR is specialized to $m$-step decodable POMDP, for each $k$, there exists a set of matrices $\{\hat{\mathbf{G}}_h^k\}_{h=1}^H \subset \mathbb{R}^{d \times r}$ such that if we replace $\bar{\hat{\psi}}^k(\tau_h)$ by $(\hat{\mathbf{G}}_h^k)^\dagger \bar{\hat{\psi}}^k(\tau_h)$ in Equation (3), then PSR-UCB terminates with a sample complexity of $poly(r, 1/\gamma, Q_A, H, |\mathcal{A}|, \beta)/\epsilon^2$.*

**Proof:** We use the notations defined in this section and particularly those in Equation (11).

The proof differentiates from the proof of Theorem 1 in two lemmas.

**First, in Lemma 2,** the meaning of $x_j$ changes from $\bar{\hat{\psi}}^k(\tau_{h-1})$ to $(\hat{\mathbf{G}}_h^k)^\dagger \bar{\hat{\psi}}^k(\tau_{h-1})$, and $w_i^\top$ changes from $\mathbf{m}^*(\omega_h)(\hat{\mathbf{M}}_h^k(o_h, a_h) - \mathbf{M}_h^*(o_h, a_h))$ to $\mathbf{m}^*(\omega_h)(\hat{\mathbf{M}}_h^k(o_h, a_h) - \mathbf{M}_h^*(o_h, a_h))\mathbf{G}_h$.

Then, the proof of the upper bound of the term $I_2$ will become $Q_A\sqrt{r\lambda}/\gamma^2$ (without dependency on $d$), simply because the dimension of bonus term $\hat{b}^k$ is now $r = |\mathcal{S}|$ instead of $d$.

**Second, in Lemma 4,** we need to show that the new bonus term $\|(\hat{\mathbf{G}}_{h+1}^k)^\dagger \bar{\hat{\psi}}(\tau_h)\|_{(\hat{U}_h^k)^{-1}}$ can still be bounded in a similar way, namely, $\|(\hat{\mathbf{G}}_{h+1}^k)^\dagger \bar{\hat{\psi}}(\tau_h)\|_{(\hat{U}_h^k)^{-1}} \leq \tilde{O}\left(\|(\mathbf{G}_{h+1}^*)^\dagger \bar{\psi}^*(\tau_h)\|_{(U_h^k)^{-1}}\right)$.

We note that the matrices $\hat{U}_h^k$ and $U_h^k$ now have new definitions:

$$\hat{U}_h^k = \lambda I + (\hat{\mathbf{G}}_{h+1}^k)^\dagger \sum_{\tau_h \in \mathcal{D}_h^k} \left[\bar{\hat{\psi}}^k(\tau_h)\bar{\hat{\psi}}^k(\tau_h)^\top\right](\hat{\mathbf{G}}_{h+1}^k)^{\dagger\top},$$

$$U_h^k = \lambda I + (\mathbf{G}_{h+1}^*)^\dagger \sum_{\tau_h \in \mathcal{D}_h^k} \left[\bar{\psi}^k(\tau_h)\bar{\psi}^*(\tau_h)^\top\right](\mathbf{G}_{h+1}^*)^{\dagger\top}.$$

To show such an upper bound, we next provide the key steps.

First, we apply Lemma 13 and obtain

$$\left\|(\hat{\mathbf{G}}_{h+1}^k)^\dagger \bar{\hat{\psi}}^k(\tau_h)\right\|_{(\hat{U}_h^k)^{-1}}$$

$$\leq \frac{1}{\sqrt{\lambda}}\left\|(\hat{\mathbf{G}}_{h+1}^k)^\dagger \bar{\hat{\psi}}^k(\tau_h) - (\mathbf{G}_{h-1}^*)^\dagger \bar{\psi}^*(\tau_h)\right\|_2$$

$$+ \left(1 + \frac{\sqrt{r}\sqrt{\sum_{\tau_h \in \mathcal{D}_h^k}\left\|(\hat{\mathbf{G}}_{h+1}^k)^\dagger \bar{\hat{\psi}}^k(\tau_h) - (\mathbf{G}_{h+1}^*)^\dagger \bar{\psi}^*(\tau_h)\right\|_2^2}}{\sqrt{\lambda}}\right)\left\|(\mathbf{G}_{h+1}^*)^\dagger \bar{\psi}^*(\tau_h)\right\|_{(U_h^k)^{-1}}.$$

For any $\tau_h$, we can derive

$$\left\|(\hat{\mathbf{G}}_{h+1}^k)^\dagger \bar{\hat{\psi}}^k(\tau_h) - (\mathbf{G}_{h+1}^*)^\dagger \bar{\psi}^*(\tau_h)\right\|_2$$

$$= \left\|\left((\hat{\mathbf{G}}_{h+1}^k)^\dagger + \hat{\mathbf{Y}}_{h+1}^k\right)\bar{\hat{\psi}}^k(\tau_h) - (\mathbf{G}_{h+1}^*)^\dagger \bar{\psi}^*(\tau_h)\right\|_2$$

$$\leq \left\|\left((\hat{\mathbf{G}}_{h+1}^k)^\dagger + \hat{\mathbf{Y}}_{h+1}^k\right)(\bar{\hat{\psi}}^k(\tau_h) - \bar{\psi}^*(\tau_h))\right\|_2 + \left\|\left((\hat{\mathbf{G}}_{h+1}^k)^\dagger + \hat{\mathbf{Y}}_{h+1}^k - (\mathbf{G}_{h+1}^*)^\dagger\right)\bar{\psi}^*(\tau_h)\right\|_2$$

$$\leq \left\|(\hat{\mathbf{G}}_{h+1}^k)^\dagger + \hat{\mathbf{Y}}_{h+1}^k\right\|_1 \left\|\bar{\hat{\psi}}^k(\tau_h) - \bar{\psi}^*(\tau_h)\right\|_1 + \left\|\left((\hat{\mathbf{G}}_{h+1}^k)^\dagger + \hat{\mathbf{Y}}_{h+1}^k\right)\left(I - \hat{\mathbf{G}}_{h+1}^k(\mathbf{G}_{h+1}^*)^\dagger\right)\bar{\psi}^*(\tau_h)\right\|_2$$

$$\leq \frac{1}{\gamma}\left\|\bar{\hat{\psi}}^k(\tau_h) - \bar{\psi}^*(\tau_h)\right\|_1 + \left\|\left((\hat{\mathbf{G}}_{h+1}^k)^\dagger + \hat{\mathbf{Y}}_{h+1}^k\right)\left(\mathbf{G}_{h+1}^* - \hat{\mathbf{G}}_{h+1}^k\right)(\mathbf{G}_{h+1}^*)^\dagger \bar{\psi}^*(\tau_h)\right\|_2,$$

where the last inequality is due to the $m$-step decodability.

The first term above has been bounded in Lemma 4. Then, we bound the second term by the Cauchy's inequality as follows:

$$\left\|\left((\hat{\mathbf{G}}_{h+1}^k)^\dagger + \hat{\mathbf{Y}}_{h+1}^k\right)\left(\mathbf{G}_{h+1}^* - \hat{\mathbf{G}}_{h+1}^k\right)(\mathbf{G}_{h+1}^*)^\dagger \bar{\psi}^*(\tau_h)\right\|_2$$

$$\leq \left\| (\mathbf{G}_{h+1}^*)^\dagger \bar{\psi}^*(\tau_h) \right\|_{(\bar{U}_h^k)^{-1}}$$

$$\times \sqrt{rQ_A\lambda/\gamma + \sum_{t<k} \mathbb{E}_{\tau_h \sim \pi^t} \left\| \left( (\hat{\mathbf{G}}_{h+1}^k)^\dagger + \hat{\mathbf{Y}}_{h+1}^k \right) \left( \mathbf{G}_{h+1}^* - \hat{\mathbf{G}}_{h+1}^k \right) (\mathbf{G}_{h+1}^*)^\dagger \bar{\psi}^*(\tau_h) \right\|_2^2}$$

$$\leq 5 \left\| (\mathbf{G}_{h+1}^*)^\dagger \bar{\psi}^*(\tau_h) \right\|_{(U_h^k)^{-1}} \sqrt{rQ_A\lambda/\gamma + |\mathcal{A}|Q_A^2\beta/\gamma^2},$$

where $\bar{U}_h^k$ denotes the expected matrix of $U_h^k$, and has the following form

$$\bar{U}_h^k = \mathbb{E}[U_h^k] = \lambda I + (\mathbf{G}_{h+1}^*)^\dagger \sum_{t<k} \mathbb{E}_{\tau_h \sim \pi^t} \left[ \bar{\psi}^k(\tau_h)\bar{\psi}^*(\tau_h)^\top \right] (\mathbf{G}_{h+1}^*)^{\dagger\top}.$$

In particular, the last inequality above follows from two facts. First, we use Lemma 39 in Zanette et al. (2021) to transform the expected matrix to the empirical matrix $U_h^k$. Second, we use Lemma Lemma 12 to upper-bound $\mathbb{E}_{\tau_h \sim \pi^t} \left\| \left( (\hat{\mathbf{G}}_{h+1}^k)^\dagger + \hat{\mathbf{Y}}_{h+1}^k \right) \left( \mathbf{G}_{h+1}^* - \hat{\mathbf{G}}_{h+1}^k \right) (\mathbf{G}_{h+1}^*)^\dagger \bar{\psi}^*(\tau_h) \right\|_2^2$ by the Hellinger squared distance between the estimated model $\hat{\theta}^k$ and the true model $\theta^*$. Then, due to the estimation guarantee (Lemma 1), we have

$$\sum_{t<k} \mathbb{E}_{\tau_h \sim \mathbb{P}_{\theta^*}^{\pi^t}} \left\| \left( (\hat{\mathbf{G}}_{h+1}^k)^\dagger + \hat{\mathbf{Y}}_{h+1}^k \right) \left( \mathbf{G}_{h+1}^* - \hat{\mathbf{G}}_{h+1}^k \right) (\mathbf{G}_{h+1}^*)^\dagger \bar{\psi}^*(\tau_h) \right\|_2^2$$

$$\leq \frac{1}{\gamma^2} \sum_{t<k} \mathbb{E}_{\tau_h \sim \mathbb{P}_{\theta^*}^{\pi^t}} \left\| \left( \mathbf{G}_{h+1}^* - \hat{\mathbf{G}}_{h+1}^k \right) (\mathbf{G}_{h+1}^*)^\dagger \bar{\psi}^*(\tau_h) \right\|_1^2$$

$$= \frac{1}{\gamma^2} \sum_{t<k} \mathbb{E}_{\tau_h \sim \mathbb{P}_{\theta^*}^{\pi^t}} \left( \sum_{\mathbf{o} \in \mathcal{Q}_{h+1}, \mathbf{a} \in \mathcal{Q}_{h+1}^A} \sum_{s \in \mathcal{S}} \left( \mathbb{P}_{\hat{\theta}^k}(\mathbf{o}|\mathbf{a}, s_{h+1} = s) - \mathbb{P}_{\theta^*}(\mathbf{o}|\mathbf{a}, s_{h+1} = s) \right) \mathbb{P}_{\theta^*}(s|\tau_h) \right)^2$$

$$\leq \frac{Q_A^2}{\gamma^2} \sum_{t<k} \mathbb{E}_{\tau_h \sim \mathbb{P}_{\theta^*}^{\pi^t}} \left( \mathbb{E}_{s \sim \mathbb{P}_{\theta^*}(\cdot|\tau_h)} \mathsf{D}_{\mathrm{TV}} \left( \mathbb{P}_{\hat{\theta}^k}^{\mathbf{u}_{\mathcal{Q}_{h+1}^A}}(\cdot|s_{h+1} = s), \mathbb{P}_{\theta^*}^{\mathbf{u}_{\mathcal{Q}_{h+1}^A}}(\cdot|s_{h+1} = s) \right) \right)^2$$

$$\leq \frac{Q_A^2}{\gamma^2} \sum_{t<k} \mathbb{E}_{\tau_h \sim \mathbb{P}_{\theta^*}^{\pi^t}} \mathbb{E}_{s \sim \mathbb{P}_{\theta^*}(\cdot|\tau_h)} \mathsf{D}_{\mathrm{TV}}^2 \left( \mathbb{P}_{\hat{\theta}^k}^{\mathbf{u}_{\mathcal{Q}_{h+1}^A}}(\cdot|s_{h+1} = s), \mathbb{P}_{\theta^*}^{\mathbf{u}_{\mathcal{Q}_{h+1}^A}}(\cdot|s_{h+1} = s) \right)$$

$$\lesssim \frac{|\mathcal{A}|Q_A^2}{\gamma^2} \sum_{\pi \in \mathcal{D}^\|} \mathsf{D}_{\mathrm{H}}^2 \left( \mathbb{P}_{\hat{\theta}^k}^\pi, \mathbb{P}_{\theta^*}^\pi \right) = \tilde{O}(\beta).$$

Combining above key steps, we obtain that

$$\mathbb{E}_{\tau_h \sim \pi} \left\| (\hat{\mathbf{G}}_{h+1}^k)^\dagger \bar{\hat{\psi}}^k(\tau_h) \right\|_{(\hat{U}_h^k)^{-1}}$$

$$\leq \mathrm{poly}(r, |\mathcal{A}|, Q_A, 1/\gamma) \left( \mathbb{E}_{\tau_h \sim \pi} \| (\mathbf{G}_{h+1}^*)^\dagger \bar{\psi}^*(\tau_h) \|_{(U_h^k)^{-1}} + \mathsf{D}_{\mathrm{TV}} \left( \mathbb{P}_{\theta^*}^\pi(\tau_H), \mathbb{P}_{\hat{\theta}^k}^\pi(\tau_H) \right) \right).$$

The rest of the proof follow the same argument as that for Theorem 1.

∎

## D    PROOF OF THEOREM 2 (FOR OFFLINE PSR-LCB)

In this section, we present the full analysis for the offline algorithm PSR-LCB to show Theorem 2. In particular, the proof of Theorem 2 consists of three main steps. **Step 1:** We provide the offline estimation guarantee for the estimated model $\hat{\theta}$. **Step 2:** Building up on the first step, we are able to show that $V_{\hat{\theta},\hat{b}}^\pi$ is a valid upper bound of the total variation distance between $\hat{\theta}$ and $\theta^*$. Hence, $V_{\hat{\theta},R}^\pi - V_{\hat{\theta},\hat{b}}^\pi$ is a valid LCB for the true value $V_{\theta^*,R}^\pi$. **Step 3:** We translate $V_{\hat{\theta},\hat{b}}^\pi$ to the ground-truth prediction feature $\bar{\psi}^*(\tau_h)$. Finally, we show that the $V_{\hat{\theta},\hat{b}}^\pi$ scales in the order of $1/\sqrt{K}$, which characterizes the performance of PSR-LCB and completes the proof.

We first introduce the following definitions for good events.

**Good Events.** Recall that $\Theta_{\min} = \left\{ \theta : \forall h, \tau_h \in \mathcal{D}_h, \; \mathbb{P}_\theta^{\pi^b}(\tau_h) \geq p_{\min} \right\}$, where $p_{\min} \leq \frac{\delta}{KH(|\mathcal{O}||\mathcal{A}|)^H}$. Let $\varepsilon \leq \frac{p_{\min}}{KH}$. Analogous to the proof for online learning, we introduce three events defined as follows.

$$
\mathcal{E}_\omega^o = \left\{ \forall \theta \in \Theta_{\min}, \; \sum_h \sum_{\tau_h \in \mathcal{D}_h} \mathrm{D}_{\mathrm{TV}}^2 \left( \mathbb{P}_\theta^{\pi^b}(\omega_h | \tau_h), \mathbb{P}_{\theta^*}^{\pi^b}(\omega_h | \tau_h) \right) \right.
$$
$$
\left. \leq 6 \sum_h \sum_{\tau_H \in \mathcal{D}_h} \log \frac{\mathbb{P}_{\theta^*}^{\pi^b}(\tau_H)}{\mathbb{P}_\theta^{\pi^b}(\tau_H)} + 31 \log \frac{3K\,|\bar{\Theta}_\varepsilon|}{\delta} \right\},
$$
$$
\mathcal{E}_\pi^o = \left\{ \forall \theta \in \Theta, K, \; \mathrm{D}_{\mathrm{H}}^2(\mathbb{P}_\theta^{\pi^b}(\tau_H), \mathbb{P}_{\theta^*}^\pi(\tau_H)) \leq \sum_{\tau_H \in \mathcal{D}} \log \frac{\mathbb{P}_{\theta^*}^{\pi^b}(\tau_H)}{\mathbb{P}_\theta^{\pi^b}(\tau_H)} + 2 \log \frac{3K|\bar{\Theta}_\varepsilon|}{\delta} \right\},
$$
$$
\mathcal{E}_{\min}^o = \left\{ \forall h, \tau_h \in \mathcal{D}_h, \; \mathbb{P}_{\theta^*}^{\pi^b}(\tau_h) \geq p_{\min} \right\}.
$$

Following the proof steps similar to those in Appendix B, we can conclude that $\mathcal{E}^o := \mathcal{E}_\omega^o \cap \mathcal{E}_\pi^o \cap \mathcal{E}_{\min}^o$ occurs with probability at least $1 - \delta$.

## D.1 STEP 1: ESTIMATION GUARANTEE

**Lemma 7 (MLE guarantee)** *Under event $\mathcal{E}^o$, the estimated model $\hat{\theta}$ by PSR-LCB satisfies*

$$
\sum_h \sum_{\tau_h \in \mathcal{D}_h} \mathrm{D}_{\mathrm{TV}}^2 \left( \mathbb{P}_{\hat{\theta}}^{\pi^b}(\omega_h | \tau_h), \mathbb{P}_{\theta^*}^{\pi^b}(\omega_h | \tau_h) \right) \leq 7\hat{\beta},
$$
$$
\mathrm{D}_{\mathrm{H}}^2 \left( \mathbb{P}_{\hat{\theta}}^{\pi^b}(\tau_H), \mathbb{P}_{\theta^*}^{\pi^b}(\tau_H) \right) \leq 7\hat{\beta}/K,
$$

*where $\hat{\beta} = 31 \log \frac{3K|\bar{\Theta}_\varepsilon|}{\delta}$.*

**Proof:** The proof follows the steps similar to those in Lemma 1, except that we replace the exploration policies $\nu_h(\pi^k, \mathrm{u}_{\mathcal{Q}_{h-1}^{\mathrm{exp}}})$ by the behavior policy. ∎

## D.2 STEP 2: UCB FOR TOTAL VARIATION DISTANCE AND LCB FOR VALUE FUNCTION

The following lemma provides an explicit upper bound on the total variation distance between the estimated model and the true model.

**Lemma 8** *Under event $\mathcal{E}^o$, for any policy $\pi$, we have*

$$
\mathrm{D}_{\mathrm{TV}} \left( \mathbb{P}_{\theta^*}^\pi(\tau_H), \mathbb{P}_{\hat{\theta}}^\pi(\tau_H) \right) \lesssim \sqrt{\frac{\hat{\beta}}{H\iota^2\gamma^2}} \sum_{h=1}^H \mathbb{E}_{\tau_{h-1} \sim \mathbb{P}_{\theta^*}^\pi} \left[ \left\| \bar{\psi}^*(\tau_{h-1}) \right\|_{\Lambda_{h-1}^{-1}} \right], \tag{12}
$$

*where $\Lambda_{h-1} = \lambda_0 I + \frac{K}{H} \mathbb{E}_{\tau_{h-1} \sim \mathbb{P}_{\theta^*}^b} \left[ \bar{\psi}^*(\tau_{h-1}) \bar{\psi}^*(\tau_{h-1})^\top \right]$, and $\lambda_0 = \frac{\gamma^4}{4Q_A^2 d}$.*

**Proof:** Similarly to the analysis in that for Lemma 9, we index $\omega_{h-1} = (o_h, a_h, \ldots, o_H, a_H)$ by $i$, and $\tau_{h-1}$ by $j$. In addition, we denote $\hat{\mathrm{m}}(\omega_h)^\top \left( \hat{\mathbf{M}}_h(o_h, a_h) - \mathbf{M}_h^*(o_h, a_h) \right)$ by $w_i^\top$, denote $\bar{\psi}^*(\tau_{h-1})$ by $x_j$, and denote $\pi(\omega_{h-1} | \tau_{h-1})$ by $\pi_{i|j}$. Then, we have

$$
\sum_{\tau_H} \left| \hat{\mathrm{m}}(\omega_h)^\top \left( \hat{\mathbf{M}}_h(o_h, a_h) - \mathbf{M}_h^*(o_h, a_h) \right) \psi^*(\tau_{h-1}) \right| \pi(\tau_H)
$$
$$
= \sum_i \sum_j |w_i^\top x_j| \pi_{i|j} \mathbb{P}_{\theta^*}^\pi(j)
$$

$$= \sum_j \sum_i (\pi_{i|j} \cdot \mathtt{sgn}(w_i^\top x_j) \cdot w_i)^\top x_j \cdot \mathbb{P}_{\theta^*}^\pi(j)$$

$$= \sum_j \left( \sum_i \pi_{i|j} \cdot \mathtt{sgn}(w_i^\top x_j) \cdot w_i \right)^\top x_j \cdot \mathbb{P}_{\theta^*}^\pi(j)$$

$$\leq \mathop{\mathbb{E}}_{j \sim \mathbb{P}_{\theta^*}^\pi} \left[ \|x_j\|_{\Lambda_{h-1}^{-1}} \sqrt{\left\| \sum_i \pi_{i|j} \cdot \mathtt{sgn}(w_i^\top x_j) \cdot w_i \right\|_{\Lambda_{h-1}}^2} \right],$$

where $\Lambda_{h-1} = \lambda_0 I + \frac{K}{H} \mathbb{E}_{\tau_{h-1} \sim \mathbb{P}_{\theta^*}^{\pi^b}} [\bar{\psi}^*(\tau_{h-1}) \bar{\psi}^*(\tau_{h-1})^\top]$ and $\lambda_0$ will be determined later. We fix $\tau_{h-1} = j_0$ and aim to analyze the coefficient of $\|x_{j_0}\|_{\Lambda_{h-1}^{-1}}$. We have

$$\left\| \sum_i \pi_{i|j_0} \cdot \mathtt{sgn}(w_i^\top x_{j_0}) \cdot w_i \right\|_{\Lambda_{h-1}}^2$$

$$= \lambda_0 \underbrace{\left\| \sum_i \pi_{i|j_0} \cdot \mathtt{sgn}(w_i^\top x_{j_0}) \cdot w_i \right\|_2^2}_{I_1} + \underbrace{\frac{K}{H} \mathop{\mathbb{E}}_{j \sim \mathbb{P}_{\theta^*}^{\pi^b}} \left[ \left( \left( \sum_i \pi_{i|j_0} \cdot \mathtt{sgn}(w_i^\top x_{j_0}) \cdot w_i \right)^\top x_j \right)^2 \right]}_{I_2}.$$

For the first term $I_1$, we have

$$\sqrt{I_1} = \sqrt{\lambda_0} \max_{x \in \mathbb{R}^{d_{h-1}}: \|x\|_2 = 1} \left| \sum_i \pi_{i|j_0} \mathtt{sgn}(w_i^\top x_{j_0}) w_i^\top x \right|$$

$$\leq \sqrt{\lambda_0} \max_{x \in \mathbb{R}^{d_{h-1}}: \|x\|_2 = 1} \sum_{\omega_{h-1}} \left| \mathbf{m}^*(\omega_h)^\top \left( \hat{\mathbf{M}}_h(o_h, a_h) - \mathbf{M}_h^*(o_h, a_h) \right) x \right| \pi(\omega_{h-1} | j_0)$$

$$\leq \sqrt{\lambda_0} \max_{x \in \mathbb{R}^{d_{h-1}}: \|x\|_2 = 1} \sum_{\omega_{h-1}} \left| \hat{\mathbf{m}}(\omega_h)^\top \hat{\mathbf{M}}_h(o_h, a_h) x \right| \pi(\omega_{h-1} | j_0)$$

$$+ \sqrt{\lambda_0} \max_{x \in \mathbb{R}^{d_{h-1}}: \|x\|_2 = 1} \sum_{\omega_{h-1}} \left| \hat{\mathbf{m}}(\omega_h)^\top \mathbf{M}_h^*(o_h, a_h) x \right| \pi(\omega_{h-1} | j_0)$$

$$\overset{(a)}{\leq} \frac{\sqrt{d \lambda_0}}{\gamma} + \frac{\sqrt{\lambda_0}}{\gamma} \max_{x \in \mathbb{R}^{d_{h-1}}: \|x\|_2 = 1} \sum_{o_h, a_h} \left\| \hat{\mathbf{M}}_h(o_h, a_h) x \right\|_1 \pi(a_h | o_h, j_0)$$

$$\overset{(b)}{\leq} \frac{2 Q_A \sqrt{d \lambda_0}}{\gamma^2},$$

where $(a)$ follows from Assumption 1, and $(b)$ follows from Proposition 2.

For the second term $I_2$, we have

$$I_2 \leq \frac{K}{H} \mathop{\mathbb{E}}_{\tau_{h-1} \sim \mathbb{P}_{\theta^*}^{\pi^b}} \left[ \left( \sum_{\omega_{h-1}} \left| \hat{\mathbf{m}}(\omega_h)^\top \left( \hat{\mathbf{M}}_h(o_h, a_h) - \mathbf{M}_h^*(o_h, a_h) \right) \bar{\psi}^*(\tau_{h-1}) \right| \pi(\omega_{h-1} | j_0) \right)^2 \right]$$

$$\leq \frac{K}{H} \mathop{\mathbb{E}}_{\tau_{h-1} \sim \mathbb{P}_{\theta^*}^{\pi^b}} \left[ \left( \sum_{\omega_{h-1}} \left| \hat{\mathbf{m}}(\omega_h)^\top \left( \hat{\mathbf{M}}_h(o_h, a_h) \hat{\bar{\psi}}(\tau_{h-1}) - \mathbf{M}_h^*(o_h, a_h) \bar{\psi}^*(\tau_{h-1}) \right) \right| \pi(\omega_{h-1} | j_0) \right. \right.$$

$$\left. \left. + \sum_{\omega_{h-1}} \left| \hat{\mathbf{m}}(\omega_h)^\top \hat{\mathbf{M}}_h(o_h, a_h) \left( \hat{\bar{\psi}}(\tau_{h-1}) - \bar{\psi}^*(\tau_{h-1}) \right) \right| \pi(\omega_{h-1} | j_0) \right)^2 \right]$$

$$\overset{(a)}{\leq} \frac{K}{H} \mathop{\mathbb{E}}_{\tau_{h-1} \sim \mathbb{P}_{\theta^*}^{\pi^b}} \left[ \left( \frac{1}{\gamma} \sum_{o_h, a_h} \left\| \mathbb{P}_{\hat{\theta}}(o_h | \tau_{h-1}) \hat{\bar{\psi}}_h(\tau_h) - \mathbb{P}_\theta(o_h | \tau_{h-1}) \bar{\psi}_h^*(\tau_h) \right\|_1 \pi(a_h | o_h, j_0) \right. \right.$$

$$+ \frac{1}{\gamma} \left\| \bar{\hat{\psi}}(\tau_{h-1}) - \bar{\psi}^*(\tau_{h-1}) \right\|_1 \bigg)^2 \bigg]$$

$$= \frac{K}{H\gamma^2} \mathbb{E}_{\tau_{h-1} \sim \mathbb{P}_{\theta^*}^{\pi^b}} \left[ \left( \sum_{o_h, a_h} \sum_{\ell=1}^{|\mathcal{Q}_h|} \left| \mathbb{P}_{\hat{\theta}}(\mathbf{o}_h^\ell, o_h | \tau_{h-1}, a_h, \mathbf{a}_h^\ell) - \mathbb{P}_{\theta^*}(\mathbf{o}_h^\ell, o_h | \tau_{h-1}, a_h, \mathbf{a}_h^\ell) \right| \pi(a_h | o_h, j_0) \right. \right.$$

$$\left. \left. + \sum_{\ell=1}^{|\mathcal{Q}_{h-1}|} \left| \mathbb{P}_{\hat{\theta}}(\mathbf{o}_{h-1}^\ell | \tau_{h-1}, \mathbf{a}_{h-1}^\ell) - \mathbb{P}_{\theta^*}(\mathbf{o}_{h-1}^\ell | \tau_{h-1}, \mathbf{a}_{h-1}^\ell) \right| \right)^2 \right]$$

$$\leq \frac{K}{H\gamma^2} \mathbb{E}_{\tau_{h-1} \sim \mathbb{P}_{\theta^*}^{\pi^b}} \left[ \left( \sum_{\mathbf{a}_{h-1} \in \mathcal{Q}_h^{\exp}} \sum_{\omega_{h-1}^o} \left| \mathbb{P}_{\hat{\theta}}(\omega_{h-1}^o | \tau_{h-1}, \mathbf{a}_{h-1}) - \mathbb{P}_{\theta^*}(\omega_{h-1}^o | \tau_{h-1}, \mathbf{a}_{h-1}) \right| \right)^2 \right]$$

$$\overset{(b)}{\leq} \frac{K}{H\iota^2\gamma^2} \mathbb{E}_{\tau_{h-1} \sim \mathbb{P}_{\theta^*}^{\pi^b}} \left[ \mathsf{D}_{\mathrm{TV}}^2 \left( \mathbb{P}_{\hat{\theta}}^{\pi^b}(\omega_{h-1} | \tau_{h-1}), \mathbb{P}_{\theta^*}^{\pi^b}(\omega_{h-1} | \tau_{h-1}) \right) \right]$$

$$\overset{(c)}{\leq} \frac{64K}{H\iota^2\gamma^2} \mathsf{D}_{\mathrm{H}}^2 \left( \mathbb{P}_{\hat{\theta}}^{\pi^b}(\tau_H), \mathbb{P}_{\theta^*}^{\pi^b}(\tau_H) \right)$$

$$\overset{(d)}{\lesssim} \frac{\hat{\beta}}{H\iota^2\gamma^2},$$

where $(a)$ follows from Assumption 1 and Equation (8), $(b)$ follows because $\pi^b(\mathbf{a}_{h-1}) \geq \iota$ for all $\mathbf{a}_{h-1} \in \mathcal{Q}_{h-1}^{\exp}$, $(c)$ follows from Lemma 12, and $(d)$ follows from Lemma 7.

Thus, by choosing $\lambda_0 = \frac{\gamma^4}{4Q_A^2 d}$, we have

$$\sum_{\tau_H} \left| \hat{\mathbf{m}}(\omega_h)^\top \left( \hat{\mathbf{M}}_h(o_h, a_h) - \mathbf{M}_h^*(o_h, a_h) \right) \psi^*(\tau_{h-1}) \right| \pi(\tau_H)$$

$$\lesssim \sqrt{\frac{\hat{\beta}}{H\iota^2\gamma^2}} \mathbb{E}_{\tau_{h-1} \sim \mathbb{P}_{\theta^*}^\pi} \left[ \left\| \bar{\psi}^*(\tau_{h-1}) \right\|_{\Lambda_{h-1}^{-1}} \right].$$

Following from Proposition 1, we conclude that

$$\mathsf{D}_{\mathrm{TV}} \left( \mathbb{P}_{\theta^*}^\pi(\tau_H), \mathbb{P}_{\hat{\theta}}^\pi(\tau_H) \right) \lesssim \min \left\{ \sqrt{\frac{\hat{\beta}}{H\iota^2\gamma^2}} \sum_{h=1}^H \mathbb{E}_{\tau_{h-1} \sim \mathbb{P}_{\theta^*}^\pi} \left[ \left\| \bar{\psi}^*(\tau_{h-1}) \right\|_{\Lambda_{h-1}^{-1}} \right], 2 \right\}.$$

∎

Following from Proposition 1, we provide the following lemma that upper bounds the estimation error by the bonus function.

**Lemma 9** *Under event $\mathcal{E}^o$, for any policy $\pi$, we have*

$$\sum_{\tau_H} \left| \mathbf{m}^*(\omega_h)^\top \left( \hat{\mathbf{M}}_h(o_h, a_h) - \mathbf{M}_h^*(o_h, a_h) \right) \hat{\psi}(\tau_{h-1}) \right| \pi(\tau_H) \leq \mathbb{E}_{\tau_{h-1} \sim \mathbb{P}_{\hat{\theta}}^\pi} \left[ \hat{\alpha}_{h-1} \left\| \bar{\hat{\psi}}(\tau_{h-1}) \right\|_{(\hat{U}_{h-1})^{-1}} \right],$$

*where*

$$\hat{U}_{h-1} = \hat{\lambda} I + \sum_{\tau_{h-1} \in \mathcal{D}_{h-1}} \bar{\hat{\psi}}(\tau_{h-1}) \bar{\hat{\psi}}(\tau_{h-1})^\top,$$

$$\hat{\alpha}_{h-1} = \frac{\hat{\lambda} Q_A^2 d}{\gamma^4} + \frac{4}{\iota^2\gamma^2} \sum_{\tau_{h-1} \in \mathcal{D}_{h-1}} \mathsf{D}_{\mathrm{TV}}^2 \left( \mathbb{P}_{\hat{\theta}}^{\pi^b}(\omega_{h-1} | \tau_{h-1}), \mathbb{P}_{\theta^*}^{\pi^b}(\omega_{h-1} | \tau_{h-1}) \right).$$

**Proof:** We index $\omega_{h-1} = (o_h, a_h, \ldots, o_H, a_H)$ by $i$, and $\tau_{h-1}$ by $j$. In addition, we denote $\mathbf{m}^*(\omega_h)^\top \left( \hat{\mathbf{M}}_h(o_h, a_h) - \mathbf{M}_h^*(o_h, a_h) \right)$ by $w_i^\top$, $\bar{\hat{\psi}}_{h-1}(\tau_{h-1}) = \frac{\hat{\psi}_{h-1}(\tau_{h-1})}{\hat{\phi}_{h-1}^\top \hat{\psi}_{h-1}(\tau_{h-1})}$ by $x_j$, and

$\pi(\omega_{h-1}|\tau_{h-1})$ by $\pi_{i|j}$. Then, we have

$$\sum_{\tau_H} \left| \mathbf{m}^*(\omega_h)^\top \left( \hat{\mathbf{M}}_h(o_h, a_h) - \mathbf{M}_h^*(o_h, a_h) \right) \hat{\psi}_{h-1}(\tau_{h-1}) \right| \pi(\tau_H)$$

$$= \sum_i \sum_j |w_i^\top x_j| \pi_{i|j} \mathbb{P}_{\hat{\theta}}^\pi(j)$$

$$= \sum_j \sum_i (\pi_{i|j} \cdot \mathtt{sgn}(w_i^\top x_j) \cdot w_i)^\top x_j \cdot \mathbb{P}_{\hat{\theta}}^\pi(j)$$

$$= \sum_j \left( \sum_i \pi_{i|j} \cdot \mathtt{sgn}(w_i^\top x_j) \cdot w_i \right)^\top x_j \cdot \mathbb{P}_{\hat{\theta}}^\pi(j)$$

$$\leq \mathbb{E}_{j \sim \mathbb{P}_{\hat{\theta}}^\pi} \left[ \|x_j\|_{(\hat{U}_{h-1}^k)^{-1}} \sqrt{\left\| \sum_i \pi_{i|j} \cdot \mathtt{sgn}(w_i^\top x_j) \cdot w_i \right\|_{\hat{U}_{h-1}}^2} \right].$$

We fix an index $j = j_0$, and aim to analyze $\left\| \sum_i \pi_{i|j_0} \cdot \mathtt{sgn}(w_i^\top x_{j_0}) \cdot w_i \right\|_{\hat{U}_{h-1}^k}^2$, which can be written as

$$\left\| \sum_i \pi_{i|j_0} \cdot \mathtt{sgn}(w_i^\top x_{j_0}) \cdot w_i \right\|_{\hat{U}_{h-1}^k}^2$$

$$= \underbrace{\hat{\lambda} \left\| \sum_i \pi_{i|j_0} \cdot \mathtt{sgn}(w_i^\top x_{j_0}) \cdot w_i \right\|_2^2}_{I_1} + \underbrace{\sum_{j \in \mathcal{D}_{h-1}} \left[ \left( \sum_i \pi_{i|j_0} \cdot \mathtt{sgn}(w_i^\top x_{j_0}) \cdot w_i \right)^\top x_j \right]^2}_{I_2}.$$

For the first term $I_1$, we have

$$\sqrt{I_1} = \sqrt{\hat{\lambda}} \max_{x \in \mathbb{R}^{d_{h-1}}: \|x\|_2 = 1} \left| \sum_i \pi_{i|j_0} \mathtt{sgn}(w_i^\top x_{j_0}) w_i^\top x \right|$$

$$\leq \sqrt{\hat{\lambda}} \max_{x \in \mathbb{R}^{d_{h-1}}: \|x\|_2 = 1} \sum_{\omega_{h-1}} \left| \mathbf{m}^*(\omega_h)^\top \left( \hat{\mathbf{M}}_h(o_h, a_h) - \mathbf{M}_h^*(o_h, a_h) \right) x \right| \pi(\omega_{h-1}|j_0)$$

$$\leq \sqrt{\hat{\lambda}} \max_{x: \|x\|_2 = 1} \sum_{\omega_{h-1}} \left| \mathbf{m}^*(\omega_h)^\top \hat{\mathbf{M}}_h(o_h, a_h) x \right| \pi(\omega_{h-1}|j_0)$$

$$+ \sqrt{\hat{\lambda}} \max_{x: \|x\|_2 = 1} \sum_{\omega_{h-1}} \left| \mathbf{m}^*(\omega_h)^\top \mathbf{M}_h^*(o_h, a_h) x \right| \pi(\omega_{h-1}|j_0)$$

$$\leq \frac{\sqrt{\hat{\lambda}}}{\gamma} \max_{x: \|x\|_2 = 1} \sum_{o_h, a_h} \left\| \hat{\mathbf{M}}_h(o_h, a_h) x \right\|_1 \pi(a_h|o_h, j_0) + \frac{\sqrt{\hat{\lambda}}}{\gamma} \|x\|_1$$

$$\leq \frac{2 Q_A \sqrt{d \hat{\lambda}}}{\gamma^2}.$$

For the second term $I_2$, we have

$$I_2 \leq \sum_{\tau_{h-1} \in \mathcal{D}_{h-1}} \left( \sum_{\omega_{h-1}} \left| \mathbf{m}^*(\omega_h)^\top \left( \hat{\mathbf{M}}_h(o_h, a_h) - \mathbf{M}_h^*(o_h, a_h) \right) \bar{\hat{\psi}}(\tau_{h-1}) \right| \pi(\omega_{h-1}|j_0) \right)^2$$

$$\leq \sum_{\tau_{h-1} \in \mathcal{D}_{h-1}} \left( \sum_{\omega_{h-1}} \left| \mathbf{m}^*(\omega_h)^\top \left( \hat{\mathbf{M}}_h(o_h, a_h) \bar{\hat{\psi}}(\tau_{h-1}) - \mathbf{M}_h^*(o_h, a_h) \bar{\psi}^*(\tau_{h-1}) \right) \right| \pi(\omega_{h-1}|j_0) \right.$$

$$
+ \sum_{\omega_{h-1}} \left| \mathbf{m}^*(\omega_h)^\top \mathbf{M}_h^*(o_h, a_h) \left( \bar{\hat{\psi}}(\tau_{h-1}) - \bar{\psi}^*(\tau_{h-1}) \right) \right| \pi(\omega_{h-1}|j_0) \Bigg)^2
$$

$$
\overset{(a)}{\leq} \sum_{\tau_{h-1} \in \mathcal{D}_{h-1}} \left( \frac{1}{\gamma} \sum_{o_h, a_h} \left\| \mathbb{P}_{\hat{\theta}}(o_h|\tau_{h-1}) \bar{\hat{\psi}}(\tau_h) - \mathbb{P}_\theta(o_h|\tau_{h-1}) \bar{\psi}^*(\tau_h) \right\|_1 \pi(a_h|o_h, j_0) \right.
$$

$$
\left. + \frac{1}{\gamma} \left\| \bar{\hat{\psi}}(\tau_{h-1}) - \bar{\psi}^*(\tau_{h-1}) \right\|_1 \right)^2
$$

$$
= \frac{1}{\gamma^2} \sum_{\tau_{h-1} \in \mathcal{D}_{h-1}} \left( \sum_{o_h, a_h} \sum_{\ell=1}^{|\mathcal{Q}_h|} \left| \mathbb{P}_{\hat{\theta}}(\mathbf{o}_h^\ell, o_h|\tau_{h-1}, a_h, \mathbf{a}_h^\ell) - \mathbb{P}_{\theta^*}(\mathbf{o}_h^\ell, o_h|\tau_{h-1}, a_h, \mathbf{a}_h^\ell) \right| \pi(a_h|o_h, j_0) \right.
$$

$$
\left. + \sum_{\ell=1}^{|\mathcal{Q}_{h-1}|} \left| \mathbb{P}_{\hat{\theta}}(\mathbf{o}_{h-1}^\ell|\tau_{h-1}, \mathbf{a}_{h-1}^\ell) - \mathbb{P}_{\theta^*}(\mathbf{o}_{h-1}^\ell|\tau_{h-1}, \mathbf{a}_{h-1}^\ell) \right| \right)^2
$$

$$
\leq \frac{1}{\gamma^2} \sum_{\tau_{h-1} \in \mathcal{D}_{h-1}} \left( \sum_{\mathbf{a}_{h-1} \in \mathcal{Q}_{h-1}^{\exp}} \sum_{\omega_{h-1}^o} \left| \mathbb{P}_{\hat{\theta}}(\omega_{h-1}^o|\tau_{h-1}, \mathbf{a}_{h-1}) - \mathbb{P}_{\theta^*}(\omega_{h-1}^o|\tau_{h-1}, \mathbf{a}_{h-1}) \right| \right)^2
$$

$$
\overset{(b)}{\leq} \frac{1}{\iota^2 \gamma^2} \sum_{\tau_{h-1} \in \mathcal{D}_{h-1}} \mathtt{D}_{\mathrm{TV}}^2 \left( \mathbb{P}_{\hat{\theta}}^{\pi^b}(\omega_{h-1}|\tau_{h-1}), \mathbb{P}_{\theta^*}^{\pi^b}(\omega_{h-1}|\tau_{h-1}) \right),
$$

where $(a)$ follows from Assumption 1, and $(b)$ follows from the condition $\pi^b(\mathbf{a}_h) \geq \iota$ for any $\mathbf{a}_h \in \mathcal{Q}_{h-1}^{\exp}$.

Thus, we conclude that

$$
\sum_{\tau_H} \left| \mathbf{m}^*(\omega_h)^\top \left( \hat{\mathbf{M}}_h(o_h, a_h) - \mathbf{M}_h^*(o_h, a_h) \right) \hat{\psi}(\tau_{h-1}) \right| \pi(\tau_H)
$$

$$
\leq \mathbb{E}_{\tau_{h-1} \sim \mathbb{P}_{\hat{\theta}}^\pi} \left[ \hat{\alpha}_{h-1} \left\| \bar{\hat{\psi}}(\tau_{h-1}) \right\|_{(\hat{U}_{h-1})^{-1}} \right],
$$

where

$$
(\hat{\alpha}_{h-1})^2 = \frac{4\hat{\lambda} Q_A^2 d}{\gamma^4} + \frac{1}{\iota^2 \gamma^2} \sum_{\tau_{h-1} \in \mathcal{D}_{h-1}} \mathtt{D}_{\mathrm{TV}}^2 \left( \mathbb{P}_{\hat{\theta}}^{\pi^b}(\omega_{h-1}|\tau_{h-1}), \mathbb{P}_{\theta^*}^{\pi^b}(\omega_{h-1}|\tau_{h-1}) \right).
$$

∎

**Corollary 4** *Under event $\mathcal{E}^o$, for any reward R, we have,*

$$
\left| V_{\hat{\theta}, R}^\pi - V_{\theta^*, R}^\pi \right| \leq V_{\hat{\theta}, \hat{b}}^\pi,
$$

*where* $\hat{b}^k(\tau_H) = \min \left\{ \hat{\alpha} \sqrt{\sum_h \left\| \bar{\hat{\psi}}_h(\tau_h) \right\|_{(\hat{U}_h)^{-1}}^2}, 1 \right\}$, *and* $\hat{\alpha} = \sqrt{\frac{4\lambda H Q_A^2 d}{\gamma^4} + \frac{7\hat{\beta}}{\iota^2 \gamma^2}}$.

**Proof:** By the definition of the total variation distance, we have

$$
\left| V_{\hat{\theta}, R}^\pi - V_{\theta^*, R}^\pi \right| \leq \mathtt{D}_{\mathrm{TV}} \left( \mathbb{P}_{\hat{\theta}}^\pi, \mathbb{P}_{\theta^*}^\pi \right)
$$

$$
\overset{(a)}{\leq} \sum_{h=1}^{H} \sum_{\tau_H} \left| \mathbf{m}^*(\omega_h)^\top \left( \hat{\mathbf{M}}_h(o_h, a_h) - \mathbf{M}_h^*(o_h, a_h) \right) \hat{\psi}(\tau_{h-1}) \right| \pi(\tau_H)
$$

$$
\overset{(b)}{\leq} \min \left\{ \sum_{h=1}^{H} \mathbb{E}_{\tau_{h-1} \sim \mathbb{P}_{\hat{\theta}}^\pi} \left[ \hat{\alpha}_{h-1} \left\| \bar{\hat{\psi}}(\tau_{h-1}) \right\|_{(\hat{U}_{h-1})^{-1}} \right], 1 \right\}
$$

$$\overset{(c)}{\leq} \min\left\{\sqrt{\sum_{h=1}^{H}\hat{\alpha}_{h-1}^2}\sqrt{\sum_{h=0}^{H-1}\left\|\bar{\hat{\psi}}_h(\tau_h)\right\|_{(\hat{U}_h)^{-1}}^2}, 1\right\},$$

where $(a)$ follows from Proposition 1, $(b)$ follows from Lemma 9 and because $R(\tau_H) \in [0,1]$, and $(c)$ follows from the Cauchy's inequality.

By Lemma 7, we further have

$$\sum_{h=1}^{H}\hat{\alpha}_{h-1}^2 \leq \frac{4\hat{\lambda}HQ_A^2 d}{\gamma^4} + \frac{1}{\iota^2\gamma^2}\sum_{h}\sum_{\tau_{h-1}\in\mathcal{D}_{h-1}}\mathrm{D}_{\mathrm{TV}}^2\left(\mathbb{P}_{\hat{\theta}^k}^{\pi^b}(\omega_{h-1}|\tau_{h-1}), \mathbb{P}_{\theta^*}^{\pi^b}(\omega_{h-1}|\tau_{h-1})\right)$$

$$\leq \frac{4\lambda HQ_A^2 d}{\gamma^4} + \frac{7\hat{\beta}}{\iota^2\gamma^2},$$

which concludes the proof. ∎

### D.3 Step 3: Relationship between Empirical Bonus and Ground-Truth Bonus

**Lemma 10 (Offline empirical bonus and true bonus term)** *Under event $\mathcal{E}^o$, for any $\pi$, we have*

$$\mathbb{E}_{\tau_H\sim\mathbb{P}_{\theta^*}^{\pi}}\left[\sqrt{\sum_{h=0}^{H-1}\left\|\bar{\hat{\psi}}_h(\tau_h)\right\|_{(\hat{U}_h)^{-1}}^2}\right]$$

$$\leq \left(1 + \frac{2\sqrt{7r\hat{\beta}}}{\iota\sqrt{\hat{\lambda}}}\right)\sum_{h=0}^{H-1}\mathbb{E}_{\tau_h\sim\mathbb{P}_{\theta^*}^{\pi}}\left[\left\|\bar{\psi}_h^*(\tau_h)\right\|_{(U_h)^{-1}}\right] + \frac{2HQ_A}{\sqrt{\hat{\lambda}}}\mathrm{D}_{\mathrm{TV}}\left(\mathbb{P}_{\theta^*}^{\pi}(\tau_H), \mathbb{P}_{\hat{\theta}}^{\pi}(\tau_H)\right).$$

**Proof:** Recall that

$$\hat{U}_h = \hat{\lambda}I + \sum_{\tau_h\in\mathcal{D}_h}\bar{\hat{\psi}}(\tau_h)\bar{\hat{\psi}}(\tau_h)^\top.$$

We define the ground-truth counterpart of $\hat{U}_h$ as follows:

$$U_h = \hat{\lambda}I + \sum_{\tau_h\in\mathcal{D}_h}\bar{\psi}(\tau_h)\bar{\psi}(\tau_h)^\top.$$

Then, by Lemma 13, we have

$$\sqrt{\sum_{h}\left\|\bar{\hat{\psi}}(\tau_h)\right\|_{(\hat{U}_h)^{-1}}^2}$$

$$\leq \sum_{h}\left\|\bar{\hat{\psi}}_h(\tau_h)\right\|_{(\hat{U}_h)^{-1}}$$

$$\leq \frac{1}{\sqrt{\hat{\lambda}}}\sum_{h}\left\|\bar{\hat{\psi}}(\tau_h) - \bar{\psi}_h^*(\tau_h)\right\|_2 + \sum_{h}\left(1 + \frac{\sqrt{r}\sqrt{\sum_{\tau_h\in\mathcal{D}_h}\left\|\bar{\hat{\psi}}(\tau_h) - \bar{\psi}^*(\tau_h)\right\|_2^2}}{\sqrt{\hat{\lambda}}}\right)\left\|\bar{\psi}^*(\tau_h)\right\|_{(U_h)^{-1}}.$$

Furthermore, note that

$$\left\|\bar{\hat{\psi}}(\tau_h) - \bar{\psi}^*(\tau_h)\right\|_2 \leq \left\|\bar{\hat{\psi}}_h(\tau_h) - \bar{\psi}_h^*(\tau_h)\right\|_1$$

$$\leq \frac{1}{\iota}\mathrm{D}_{\mathrm{TV}}\left(\mathbb{P}_{\hat{\theta}}^{\pi^b}(\omega_h|\tau_h), \mathbb{P}_{\theta^*}^{\pi^b}(\omega_h|\tau_h)\right),$$

where the second inequality follows because $\pi^b(\mathbf{a}_h) \geq \iota$ for all $\mathbf{a}_h \in \mathcal{Q}_h^{\exp}$. By Lemma 7, we conclude that

$$\sqrt{\sum_{h}\left\|\bar{\hat{\psi}}_h(\tau_h)\right\|_{(\hat{U}_h)^{-1}}^2}$$

$$\leq \frac{1}{\sqrt{\hat{\lambda}}} \sum_h \left\| \bar{\hat{\psi}}_h(\tau_h) - \bar{\psi}_h^*(\tau_h) \right\|_2 + \left( 1 + \frac{\sqrt{7r\hat{\beta}}}{\iota\sqrt{\hat{\lambda}}} \right) \sum_h \left\| \bar{\psi}_h^*(\tau_h) \right\|_{(U_h^k)^{-1}}.$$

For the first term, following an argument similar to those in Lemma 4 and taking the expectation, we have

$$\sum_h \mathbb{E}_{\tau_h \sim \mathbb{P}_{\theta^*}^\pi} \left[ \left\| \bar{\hat{\psi}}(\tau_h) - \bar{\psi}(\tau_h) \right\|_1 \right]$$

$$\leq \sum_h \sum_{\tau_h} \left( \left\| \bar{\hat{\psi}}(\tau_h) \left( \mathbb{P}_\theta^\pi(\tau_h) - \mathbb{P}_{\hat{\theta}}^\pi(\tau_h) \right) + \bar{\hat{\psi}}(\tau_h) \mathbb{P}_{\hat{\theta}}^\pi(\tau_h) - \bar{\psi}(\tau_h) \mathbb{P}_{\theta^*}^\pi(\tau_h) \right\|_1 \right)$$

$$\leq \sum_h \sum_{\tau_h} \left( \left\| \bar{\hat{\psi}}(\tau_h) \right\|_1 \left| \mathbb{P}_\theta^\pi(\tau_h) - \mathbb{P}_{\hat{\theta}}^\pi(\tau_h) \right| + \left\| \hat{\psi}(\tau_h) - \bar{\psi}^*(\tau_h) \right\|_1 \pi(\tau_h) \right)$$

$$\leq 2 \sum_h |\mathcal{Q}_h^A| \mathsf{D}_{\mathsf{TV}} \left( \mathbb{P}_{\theta^*}^\pi(\tau_h), \mathbb{P}_{\hat{\theta}}^\pi(\tau_h) \right)$$

$$\leq 2 H Q_A \mathsf{D}_{\mathsf{TV}} \left( \mathbb{P}_{\theta^*}^\pi(\tau_H), \mathbb{P}_{\hat{\theta}}^\pi(\tau_H) \right),$$

which completes the proof. ∎

### D.4 PROOF OF THEOREM 2

Now, we are ready to prove Theorem 2.

**Theorem 4 (Restatement of Theorem 2)** *Suppose Assumption 1 holds. Let* $\iota = \min_{\mathbf{a}_h \in \mathcal{Q}_h^{\exp}} \pi^b(\mathbf{a}_h)$, $p_{\min} = O(\frac{\delta}{KH(|\mathcal{O}||\mathcal{A}|)^H})$, $\varepsilon = O(\frac{p_{\min}}{KH})$, $\hat{\beta} = O(\log |\bar{\Theta}_\varepsilon|)$, $\hat{\lambda} = \frac{\gamma C_{\pi^b,\infty}^\pi \hat{\beta} \max\{\sqrt{r}, Q_A\sqrt{H}/\gamma\}}{\iota^2 Q_A\sqrt{dH}}$, *and* $\hat{\alpha} = O\left( \frac{Q_A\sqrt{dH}}{\gamma^2}\sqrt{\hat{\lambda}} + \frac{\sqrt{\hat{\beta}}}{\iota\gamma} \right)$. *Then, with probability at least* $1 - \delta$, *the output* $\bar{\pi}$ *of Algorithm 2 satisfies that*

$$\forall \pi, \ V_{\theta^*,R}^\pi - V_{\theta^*,R}^{\bar{\pi}} \leq \tilde{O}\left( \left( \sqrt{r} + \frac{Q_A\sqrt{H}}{\gamma} \right) \frac{C_{\pi^b,\infty}^\pi Q_A H^2}{\iota\gamma^2} \sqrt{\frac{rd\hat{\beta}}{K}} \right).$$

**Proof:** Under event $\mathcal{E}^o$, the performance difference can be upper bounded as follows.

$$V_{\theta^*,r}^\pi - V_{\theta^*,r}^{\hat{\pi}}$$

$$= V_{\theta^*,r}^\pi - (V_{\hat{\theta},r}^\pi - V_{\hat{\theta},\hat{b}}^\pi) + \underbrace{(V_{\hat{\theta},r}^\pi - V_{\hat{\theta},\hat{b}}^\pi) - (V_{\hat{\theta},r}^{\hat{\pi}} - V_{\hat{\theta},\hat{b}}^{\hat{\pi}})}_{I_1} + \underbrace{V_{\hat{\theta},r}^{\hat{\pi}} - V_{\hat{\theta},\hat{b}}^{\hat{\pi}} - V_{\theta^*,r}^{\hat{\pi}}}_{I_2}$$

$$\overset{(a)}{\leq} \frac{1}{2}\mathsf{D}_{\mathsf{TV}}\left( \mathbb{P}_{\hat{\theta}}^\pi, \mathbb{P}_{\theta^*}^\pi \right) + V_{\hat{\theta},\hat{b}}^\pi$$

$$\overset{(b)}{\leq} \mathsf{D}_{\mathsf{TV}}\left( \mathbb{P}_{\hat{\theta}}^\pi, \mathbb{P}_{\theta^*}^\pi \right) + V_{\theta^*,\hat{b}}^\pi,$$

where $(a)$ follows from the design of $\bar{\pi}$ that results in $I_1 \leq 0$, and from Corollary 4 which implies $I_2 \leq 0$, and $(b)$ follows from Corollary 4.

By Lemma 8, we have

$$\mathsf{D}_{\mathsf{TV}}\left( \mathbb{P}_{\hat{\theta}}^\pi, \mathbb{P}_{\theta^*}^\pi \right) \lesssim \sqrt{\frac{\hat{\beta}}{H\iota^2\gamma^2}} \sum_{h=1}^H \mathbb{E}_{\tau_{h-1} \sim \mathbb{P}_{\theta^*}^\pi} \left[ \left\| \bar{\psi}^*(\tau_{h-1}) \right\|_{\Lambda_{h-1}^{-1}} \right]$$

$$\overset{(a)}{\leq} C_{\pi^b,\infty}^\pi \sqrt{\frac{\hat{\beta}}{H\iota^2\gamma^2}} \sum_{h=1}^H \mathbb{E}_{\tau_{h-1} \sim \mathbb{P}_{\theta^*}^{\pi^b}} \left[ \left\| \bar{\psi}^*(\tau_{h-1}) \right\|_{\Lambda_{h-1}^{-1}} \right]$$

$$\leq C^{\pi}_{\pi^b,\infty} H \sqrt{\frac{\hat{\beta}}{H\iota^2\gamma^2}} \sqrt{\frac{rH}{K}} = C^{\pi}_{\pi^b,\infty} H \sqrt{\frac{r\hat{\beta}}{K\iota^2\gamma^2}},$$

where $(a)$ follows from the definition of the coverage coefficient.

By Lemma 10, we have

$$V^{\pi}_{\theta^*,\hat{b}}$$

$$\leq \min\left\{\hat{\alpha}\left(1+\frac{2\sqrt{7r\hat{\beta}}}{\iota\sqrt{\hat{\lambda}}}\right)\sum_{h=0}^{H-1} \mathbb{E}_{\tau_h \sim \mathbb{P}^{\pi}_{\theta^*}}\left[\left\|\bar{\psi}^*(\tau_h)\right\|_{(U_h)^{-1}}\right] + \frac{2\hat{\alpha}HQ_A}{\sqrt{\hat{\lambda}}}\mathsf{D}_{\mathsf{TV}}\left(\mathbb{P}^{\pi}_{\theta^*}(\tau_H),\mathbb{P}^{\pi}_{\hat{\theta}}(\tau_H)\right),1\right\}$$

$$\leq \min\left\{\hat{\alpha}\left(1+\frac{2\sqrt{7r\hat{\beta}}}{\iota\sqrt{\hat{\lambda}}}\right)\sum_{h=0}^{H-1} \mathbb{E}_{\tau_h \sim \mathbb{P}^{\pi}_{\theta^*}}\left[\left\|\bar{\psi}^*(\tau_h)\right\|_{(U_h)^{-1}}\right],1\right\} + \frac{2\hat{\alpha}HQ_A}{\sqrt{\hat{\lambda}}}\mathsf{D}_{\mathsf{TV}}\left(\mathbb{P}^{\pi}_{\theta^*}(\tau_H),\mathbb{P}^{\pi}_{\hat{\theta}}(\tau_H)\right)$$

$$\leq \underbrace{\min\left\{\hat{\alpha}C^{\pi}_{\pi^b,\infty}\left(1+\frac{2\sqrt{7r\hat{\beta}}}{\iota\sqrt{\hat{\lambda}}}\right)\sum_{h=0}^{H-1} \mathbb{E}_{\tau_h \sim \mathbb{P}^{\pi^b}_{\theta^*}}\left[\left\|\bar{\psi}^*(\tau_h)\right\|_{(U_h)^{-1}}\right],1\right\}}_{I_3} + \frac{2\hat{\alpha}HQ_A}{\sqrt{\hat{\lambda}}}\mathsf{D}_{\mathsf{TV}}\left(\mathbb{P}^{\pi}_{\theta^*}(\tau_H),\mathbb{P}^{\pi}_{\hat{\theta}}(\tau_H)\right).$$

Recall that $U_h = \hat{\lambda}I + \sum_{\tau_h \in \mathcal{D}_h}\bar{\psi}^*(\tau_h)\bar{\psi}^*(\tau_h)^\top$ and the distribution of $\tau_h \in \mathcal{D}_h$ follows $\mathbb{P}^{\pi^b}_{\theta^*}$. Therefore, by the Azuma-Hoeffding's inequality (Lemma 11), with probability at least $1-\delta$, we have

$$KI_3 \leq \sqrt{2K\log(2/\delta)} + \min\left\{\hat{\alpha}C^{\pi}_{\pi^b,\infty}\left(1+\frac{2\sqrt{7r\hat{\beta}}}{\iota\sqrt{\hat{\lambda}}}\right)\sum_{h=0}^{H-1}\sum_{\tau_h \in \mathcal{D}_h}\left\|\bar{\psi}^*(\tau_h)\right\|_{(U_h)^{-1}},1\right\}$$

$$\overset{(a)}{\lesssim} \sqrt{K\log(2/\delta)} + \hat{\alpha}C^{\pi}_{\pi^b,\infty}\left(1+\frac{\sqrt{r\hat{\beta}}}{\iota\sqrt{\hat{\lambda}}}\right)H\sqrt{\frac{rK}{H}}$$

$$\lesssim \hat{\alpha}C^{\pi}_{\pi^b,\infty}\left(1+\frac{\sqrt{r\hat{\beta}}}{\iota\sqrt{\hat{\lambda}}}\right)\sqrt{rHK\log(2/\delta)},$$

where $(a)$ follows from the Cauchy's inequality.

Combining the above results, we have

$$V^{\pi}_{\theta^*,r} - V^{\hat{\pi}}_{\theta^*,r}$$

$$\lesssim \hat{\alpha}C^{\pi}_{\pi^b,\infty}\left(1+\frac{\sqrt{r\hat{\beta}}}{\iota\sqrt{\hat{\lambda}}}\right)\sqrt{\frac{rH\log(2/\delta)}{K}} + \frac{\hat{\alpha}HQ_A}{\sqrt{\hat{\lambda}}}\frac{C^{\pi}_{\pi^b,\infty}H}{\iota\gamma}\sqrt{\frac{r\hat{\beta}}{K}}$$

$$\lesssim C^{\pi}_{\pi^b,\infty}\left(\frac{Q_A\sqrt{dH}}{\gamma^2}\sqrt{\hat{\lambda}}+\frac{\sqrt{\hat{\beta}}}{\iota\gamma}\right)\left(1+\frac{\max\{\sqrt{r},\sqrt{H}Q_A/\gamma\}}{\iota\sqrt{\hat{\lambda}}}\right)H\sqrt{\frac{rH\hat{\beta}\log(2/\delta)}{K}}$$

$$= C^{\pi}_{\pi^b,\infty}\left(\frac{\iota Q_A\sqrt{dH}}{\gamma\sqrt{\hat{\beta}}}\sqrt{\hat{\lambda}}+1\right)\left(1+\frac{\max\{\sqrt{r},\sqrt{H}Q_A/\gamma\}}{\iota\sqrt{\hat{\lambda}}}\right)\frac{H\sqrt{\hat{\beta}}}{\iota\gamma}\sqrt{\frac{rH\hat{\beta}\log(2/\delta)}{K}}$$

$$\overset{(a)}{\lesssim} \max\left\{\sqrt{r},\frac{\sqrt{H}Q_A}{\gamma}\right\}C^{\pi}_{\pi^b,\infty}\frac{Q_AH^2\sqrt{d}}{\iota\gamma^2}\sqrt{\frac{r\hat{\beta}\log(2/\delta)}{K}},$$

where $(a)$ follows because

$$\lambda = \frac{\gamma\sqrt{\hat{\beta}}\max\{\sqrt{r},Q_A\sqrt{H}/\gamma\}}{\iota^2 Q_A\sqrt{dH}}.$$

This completes the proof. ∎

## E   Auxiliary Lemmas

**Lemma 11 (Azuma–Hoeffding inequality)** *Consider a domain $\mathcal{X}$, and a filtration $\mathcal{F}_1 \subset \ldots \subset \mathcal{F}_k \subset \ldots$ on the domain $\mathcal{X}$. Suppose that $\{X_1, \ldots, X_k, \ldots\} \subset [-B, B]$ is adapted to the filtration $(\mathcal{F}_t)_{t=1}^{\infty}$, i.e., $X_k$ is $\mathcal{F}_k$-measurable. Then*

$$\mathbb{P}\left( \left| \sum_{t=1}^{k-1} X_t - \mathbb{E}[X_t] \right| \geq \sqrt{2kB^2 \log(2/\delta)} \right) \leq \delta. \tag{13}$$

The following lemma characterize the relationship between the total variation distance and the Hellinger-squared distance. Note that the result for probability measures has been proved in Lemma H.1 in Zhong et al. (2022). Since we consider more general bounded measures, we provide the full proof for completeness.

**Lemma 12** *Given two bounded measures $P$ and $Q$ defined on the set $\mathcal{X}$. Let $|P| = \sum_{x \in \mathcal{X}} P(x)$ and $|Q| = \sum_{x \in \mathcal{X}} Q(x)$. We have*

$$\mathtt{D}_{\mathtt{TV}}^2(P, Q) \leq 4(|P| + |Q|)\mathtt{D}_{\mathtt{H}}^2(P, Q)$$

*In addition, if $P_{Y|X}, Q_{Y|X}$ are two conditional distributions over a random variable $Y$, and $P_{X,Y} = P_{Y|X}P$, $Q_{X,Y} = Q_{Y|X}Q$ are the joint distributions when $X$ follows the distributions $P$ and $Q$, respectively, we have*

$$\mathop{\mathbb{E}}_{X \sim P} \left[ \mathtt{D}_{\mathtt{H}}^2(P_{Y|X}, Q_{Y|X}) \right] \leq 8\mathtt{D}_{\mathtt{H}}^2(P_{X,Y}, Q_{X,Y}).$$

**Proof:** We first prove the first inequality. By the definition of total variation distance, we have

$$\begin{aligned}
\mathtt{D}_{\mathtt{TV}}^2(P, Q) &= \left( \sum_x |P(x) - Q(x)| \right)^2 \\
&= \left( \sum_x \left( \sqrt{P(x)} - \sqrt{Q(x)} \right) \left( \sqrt{P(x)} + \sqrt{Q(x)} \right) \right)^2 \\
&\overset{(a)}{\leq} \left( \sum_x \left( \sqrt{P(x)} - \sqrt{Q(x)} \right)^2 \right) \left( 2\sum_x (P(x) + Q(x)) \right) \\
&\leq 4(|P| + |Q|)\mathtt{D}_{\mathtt{H}}^2(P, Q),
\end{aligned}$$

where $(a)$ follows from the Cauchy's inequality and because $(a + b)^2 \leq 2a^2 + 2b^2$.

For the second inequality, we have,

$$\begin{aligned}
\mathop{\mathbb{E}}_{X \sim P} &\left[ \mathtt{D}_{\mathtt{H}}^2(P_{Y|X}, Q_{Y|X}) \right] \\
&= \sum_x P(x) \left( \sum_y \left( \sqrt{P_{Y|X}(y)} - \sqrt{Q_{Y|X}(y)} \right)^2 \right) \\
&= \sum_{x,y} \left( \sqrt{P_{X,Y}(x,y)} - \sqrt{Q_{X,Y}(x,y)} + \sqrt{Q_{Y|X}(y)Q(x)} - \sqrt{Q_{Y|X}(y)P(x)} \right)^2 \\
&\leq 2\sum_{x,y} \left( \sqrt{P_{X,Y}(x,y)} - \sqrt{Q_{X,Y}(x,y)} \right)^2 + 2\sum_{x,y} Q_{Y|X}(y) \left( \sqrt{Q(x)} - \sqrt{P(x)} \right)^2 \\
&= 4\mathtt{D}_{\mathtt{H}}^2(P_{X,Y}, Q_{X,Y}) + 2(|P| + |Q| - 2\sum_x \sqrt{P(x)Q(x)}) \\
&\overset{(a)}{\leq} 4\mathtt{D}_{\mathtt{H}}^2(P_{X,Y}, Q_{X,Y}) + 2(|P| + |Q| - 2\sum_x \sum_y \sqrt{P_{Y|X}(y)P(x)Q_{Y|X}(y)Q(x)})
\end{aligned}$$

$$= 8\mathrm{D}_{\mathrm{H}}^2(P_{X,Y}, Q_{X,Y}),$$

where $(a)$ follows from the Cauchy's inequality that applies on $\sum_y \sqrt{P_{Y|X}(y)Q_{Y|X}(y)}$. ∎

**Lemma 13** *Consider two vector sequences $\{x_i\}_{i \in \mathcal{I}}$ and $\{y_i\}_{i \in \mathcal{I}}$ and an index subset $\mathcal{J} \subset \mathcal{I}$. Suppose $A = \lambda I + \sum_{j \in \mathcal{J}} x_j x_j^\top$ and $B = \sum_{j \in \mathcal{J}} y_j y_j^\top$, and $\mathtt{rank}\left(\{x_i\}_{i \in \mathcal{I}}\right) = \mathtt{rank}\left(\{y_i\}_{i \in \mathcal{I}}\right) = r$. Then*

$$\forall i \in \mathcal{I}, \ \|x_i\|_{A^{-1}} \leq \frac{1}{\sqrt{\lambda}}\|x_i - y_i\|_2 + \left(1 + \frac{2\sqrt{r}\sqrt{\sum_{j \in \mathcal{J}} \|x_j - y_j\|_2^2}}{\sqrt{\lambda}}\right)\|y_i\|_{B^{-1}}.$$

**Proof:** We first write

$$\begin{aligned}
&\|x_i\|_{A^{-1}} - \|y_i\|_{B^{-1}} \\
&= \|x_i\|_{A^{-1}} - \|y_i\|_{A^{-1}} + \|y_i\|_{A^{-1}} - \|y_i\|_{B^{-1}} \\
&= \underbrace{\frac{\|x_i\|_{A^{-1}}^2 - \|y_i\|_{A^{-1}}^2}{\|x_i\|_{A^{-1}} + \|y_i\|_{A^{-1}}}}_{I_1} + \underbrace{\frac{\|y_i\|_{A^{-1}}^2 - \|y_i\|_{B^{-1}}^2}{\|y_i\|_{A^{-1}} + \|y_i\|_{B^{-1}}}}_{I_2}.
\end{aligned}$$

For the first term $I_1$, we repeatedly apply the Cauchy's inequality, and have

$$\begin{aligned}
&\|x_i\|_{A^{-1}}^2 - \|y_i\|_{A^{-1}}^2 \\
&= x_i^\top A^{-1}(x_i - y_i) + y_i^\top A^{-1}(x_i - y_i) \\
&\leq \|x_i\|_{A^{-1}}\|x_i - y_i\|_{A^{-1}} + \|y_i\|_{A^{-1}}\|x_i - y_i\|_{A^{-1}} \\
&\leq \frac{1}{\sqrt{\lambda}}\left(\|x_i\|_{A^{-1}} + \|y_i\|_{A^{-1}}\right)\|x_i - y_i\|_2.
\end{aligned}$$

For the second term $I_2$, we repeatedly apply the Cauchy's inequality, and have

$$\begin{aligned}
&\|y_i\|_{A^{-1}}^2 - \|y_i\|_{B^{-1}}^2 \\
&= y_i^\top A^{-1}(B - A)B^{-1}y_i \\
&= \sum_{j \in \mathcal{J}}\left(y_i^\top A^{-1}x_j(y_j - x_j)^\top B^{-1}y_i + y_i^\top A^{-1}(y_j - x_j)y_j^\top B^{-1}y_i\right) \\
&\leq \sum_{j \in \mathcal{J}}\|x_j\|_{A^{-1}}\|y_j - x_j\|_{B^{-1}}\|y_i\|_{A^{-1}}\|y_i\|_{B^{-1}} \\
&\quad + \sum_{j \in \mathcal{J}}\|y_j - x_j\|_{A^{-1}}\|y_j\|_{B^{-1}}\|y_i\|_{A^{-1}}\|y_i\|_{B^{-1}} \\
&\leq \frac{1}{\sqrt{\lambda}}\|y_i\|_{A^{-1}}\|y_i\|_{B^{-1}}\sqrt{\sum_{j \in \mathcal{J}}\|x_j - y_j\|_2^2}\sqrt{\sum_{j \in \mathcal{J}}\|x_j\|_{A^{-1}}^2} \\
&\quad + \frac{1}{\sqrt{\lambda}}\|y_i\|_{A^{-1}}\|y_i\|_{B^{-1}}\sqrt{\sum_{j \in \mathcal{J}}\|x_j - y_j\|_2^2}\sqrt{\sum_{j \in \mathcal{J}}\|y_j\|_{B^{-1}}^2} \\
&\leq \frac{2\sqrt{r}}{\sqrt{\lambda}}\|y_i\|_{A^{-1}}\|y_i\|_{B^{-1}}\sqrt{\sum_{j \in \mathcal{J}}\|x_{j_h} - y_{j_h}\|_2^2}.
\end{aligned}$$

Therefore, the lemma follows from the fact that $\|y_i\|_{A^{-1}} \leq \|y_i\|_{A^{-1}} + \|y_i\|_{B^{-1}}$. ∎

The following lemma and its variants has been developed in Dani et al. (2008); Abbasi-Yadkori et al. (2011); Carpentier et al. (2020). We slightly generalize the padding term from 1 to an arbitrary positive number $B$ and provide the full proof for completeness.

**Lemma 14 (Elliptical potential lemma )** *For any sequence of vectors $\mathcal{X} = \{x_1, \ldots, x_n, \ldots\} \subset \mathbb{R}^d$, let $U_k = \lambda I + \sum_{t<k} x_k x_k^\top$, where $\lambda$ is a positive constant, and $B > 0$ is a real number. If the rank of $\mathcal{X}$ is at most $r$, then, we have*

$$\sum_{k=1}^{K} \min\left\{\|x_k\|_{U_k^{-1}}^2, B\right\} \leq (1+B)r\log(1+K/\lambda),$$

$$\sum_{k=1}^{K} \min\left\{\|x_k\|_{U_k^{-1}}, \sqrt{B}\right\} \leq \sqrt{(1+B)rK\log(1+K/\lambda)}.$$

**Proof:** Note that the second inequality is an immediate result from the first inequality by the Cauchy's inequality. Hence, it suffices to prove the first inequality. To this end, we have

$$\sum_{k=1}^{K} \min\left\{\|x_k\|_{U_k^{-1}}^2, B\right\} \overset{(a)}{\leq} (1+B) \sum_{k=1}^{K} \log\left(1 + \|x_k\|_{U_k^{-1}}^2\right)$$

$$= (1+B) \sum_{k=1}^{K} \log\left(1 + \texttt{trace}\left((U_{k+1} - U_k)\, U_k^{-1}\right)\right)$$

$$= (1+B) \sum_{k=1}^{K} \log\left(1 + \texttt{trace}\left(U_k^{-1/2}\left(U_{k+1} - U_k\right) U_k^{-1/2}\right)\right)$$

$$\leq (1+B) \sum_{k=1}^{K} \log \texttt{det}\left(I_d + U_k^{-1/2}\left(U_{k+1} - U_k\right) U_k^{-1/2}\right)$$

$$= (1+B) \sum_{k=1}^{K} \log \frac{\texttt{det}\left(U_{k+1}\right)}{\texttt{det}(U_k)}$$

$$= (1+B) \log \frac{\texttt{det}(U_{K+1})}{\texttt{det}(U_1)}$$

$$= (1+B) \log \texttt{det}\left(I + \frac{1}{\lambda} \sum_{k=1}^{K} x_k x_k^\top\right)$$

$$\overset{(b)}{\leq} (1+B)r\log(1+K/\lambda),$$

where $(a)$ follows because $x \leq (1+B)\log(1+x)$ if $0 < x \leq B$, and $(b)$ follows because $\texttt{rank}(\mathcal{X}) \leq r$. ∎

