# OpenReview forum: "Provably Efficient UCB-type Algorithms For Learning Predictive State Representations"
_ICLR.cc/2024/Conference — ICLR 2024 poster_

### Official Review · Reviewer_eKGX · 2023-10-29

**Soundness:** 3 good
**Presentation:** 3 good
**Contribution:** 3 good
**Rating:** 6
**Confidence:** 3

**Summary:**

This work establishes the first UCB-type algorithms for PSR problems. The proposed algorithm is computationally tractable compared with previous methods and obtains sample complexity which matches the previous best upper bound w.r.t. the rank of PSR and accuracy level. The counterpart of the algorithm and analysis in the offline case has also been established.

**Strengths:**

1. Motivation: Given previous methods studying PSR problems that are not computationally tractable or sample from posteriors maintained over a large model set, studying a computationally tractable method for PSR is well-motivated.
2. Novelty: This work proposes the first UCB bonus to enable efficient exploration of PSR problems, the core design of which is a new MLE guarantee. The newly proposed MLE guarantee and UCB bonus might be of independent interest.

**Weaknesses:**

1. The proof of Theorem 1 relies on some sort of MLE guarantee, a confidence bound constructed by the empirical features and relating the empirical bonus to the ground-truth bonus. Such an approach shares similar spirits as previous works studying MDP with low-rank structures (say [1]). I would like to suggest the authors give more comparisons on this.
2. The result in Theorem 1 seems to critically depend on the newly proposed MLE guarantee in this work. It might be better if discussions about how this MLE guarantee is established and to what extent it differs from previous ones are given.
3. By “computationally efficient”, it in most cases means that the algorithm enjoys polynomial computation efficiency. In some parts of this work, the authors claim that the proposed algorithms are computationally efficient. In my opinion, the proposed algorithm is indeed computationally tractable or oracle-efficient but might not be “computation efficient”, since the polynomial efficiency of the proposed algorithm is not guaranteed (correct me if I am wrong).

[1] Uehara et al. Representation Learning for Online and Offline RL in Low-rank MDPs. ICLR, 2022.

**Questions:**

Please see the weakness part above.

---

> ### Author Response · Authors · 2023-11-19
> **Response to Reviewer eKGX (Part 1)**
>
> We thank the reviewer for the helpful comments, and would like to provide the following responses. The paper has been revised accordingly to reflect these responses, where the changes are highlighted in the blue color. We hope that these discussions can clarify the reviewer's concerns, and we will be more than happy to answer any further questions you may have.
>
>
> **Q1:** The proof of Theorem 1 relies on some sort of MLE guarantee, a confidence bound constructed by the empirical features and relating the empirical bonus to the ground-truth bonus. Such an approach shares similar spirits as previous works studying MDP with low-rank structures (say [1]). I would like to suggest the authors give more comparisons on this.
>
> [1] Uehara et al. Representation Learning for Online and Offline RL in Low-rank MDPs. ICLR, 2022.
>
> **A1:** While both [1] and this work utilize MLE oracle and UCB-based design, our method and analysis differs significantly from [1] as follows.
>
> First, we adopt a constrained MLE oracle, which provides a more **stable estimation** (page 6) compared with vanilla MLE used in [1]. Here, the stable estimation implies that the estimated model is guaranteed to be close to the true model not only in expectation, but also on empirical samples.
> Specifically, we establish the estimated model guarantee in terms of the empirical conditional probability distributions, i.e. $$\sum\_{k,h} \mathtt{D}\_{\mathtt{TV}} (\mathbb{P}\_{\hat{\theta}}(\cdot|\tau_h^{k,h+1}), \mathbb{P}\_{\theta^*}(\cdot|\tau_h^{k,h+1}) ) \leq \beta.$$
> This stability indicates that $\bar{\hat{\psi}}^k(\tau_h^{k,h+1})$ is close to the true feature $\bar{\psi}^*(\tau_h^{k,h+1})$, which allows us to use empirical features $\bar{\hat{\psi}}^k(\tau_h)$ to design bonus function $\hat{b}^k$. This cannot be achieved by vanilla MLE estimation in PSRs.
>
> Second, due to **non-Markovian property of PSR**, the step-back technique, which is frequently used to relate the empirical bonus to the ground-truth bonus in low-rank MDPs, cannot be employed in our setting. Instead, we develop a new technique in Lemma 13, which controls the difference between two bonus functions using different features.
>
> Third, we exploit the **physical meaning of the features in PSR** described in Sec 4.1 for efficient exploration. Specifically, each coordinate of a feature represents the transition probability from a history to a core test. This unique property of PSR further allows us to successfully design a UCB-based approach and addresses the challenge brought by PSR regarding how to relate the empirical bonus to the true bonus (see Lemma 4).
>
>
> **Q2:** It might be better if discussions about how this MLE guarantee is established and to what extent it differs from previous ones are given.
>
>
> **A2:** Our MLE guarantee is established by analyzing the conditional probability $\mathbb{P}\_{\theta}(\omega_h|\tau_h^{k,h+1})$ for those sampled history $\tau_h^{k,h+1}$. Due to the constraint, we can find a good cover of the parameter space $\Theta$ in terms of the conditional probability distribution distance (see Proposition 5), i.e. $\mathtt{D}\_{\mathtt{TV}}(\mathbb{P}_{\theta}(\cdot|\tau_h^{k,h+1}), \mathbb{P}\_{\theta^*}(\cdot|\tau_h^{k,h+1}))$. Equipped with the optimistic cover, we then apply Chernoff bound and union bound to obtain the final guarantee.
>
> The key difference of our proposed constrained MLE oracle from previous ones is that the constraint provides a more stable MLE solution. Specifically, we establish the estimated model guarantee in terms of the empirical conditional probability distributions, i.e. $$\sum\_{k,h} \mathtt{D}\_{\mathtt{TV}} (\mathbb{P}\_{\hat{\theta}}(\cdot|\tau_h^{k,h+1}), \mathbb{P}\_{\theta^*}(\cdot|\tau_h^{k,h+1}) ) \leq \beta.$$
> This stability indicates that $\bar{\hat{\psi}}^k(\tau_h^{k,h+1})$ is close to the true feature $\bar{\psi}^*(\tau_h^{k,h+1})$, which allows us to use empirical features for planning and makes our algorithm computationally tractable.
>
> For a clearer comparison, we note that previous works either does not have "conditional distribution" guarantee [2], or only have guarantee on the "expected" distance $\mathbb{E}\_{\tau_h^{k,h+1}}[\mathtt{D}\_{\mathtt{TV}}(\mathbb{P}\_{\theta}(\cdot|\tau_h^{k,h+1}), \mathbb{P}\_{\theta^*}(\cdot|\tau_h^{k,h+1}))]$ [1] instead of on the empirical samples.
>
>
> [2] Liu, et al., Optimistic MLE—A Generic Model-based Algorithm for Partially Observable Sequential Decision Making, 2022.

---

> ### Author Response · Authors · 2023-11-19
> **Response to Reviewer eKGX (Part 2)**
>
> **Q3:** By “computationally efficient”, it in most cases means that the algorithm enjoys polynomial computation efficiency. In some parts of this work, the authors claim that the proposed algorithms are computationally efficient. In my opinion, the proposed algorithm is indeed computationally tractable or oracle-efficient but might not be “computation efficient”, since the polynomial efficiency of the proposed algorithm is not guaranteed (correct me if I am wrong).
>
>
>
> **A3:** We apologize for the inaccurate description of the computational efficiency of our algorithm. The proposed algorithm is indeed oracle-efficient. We have changed the term "computationally efficient" to "computationally tractable" in the revision.
>
>
>
> -----
>
> We thank the reviewer again for the helpful comments and suggestions for our work. If our response resolves your concerns to a satisfactory level, we kindly ask the reviewer to consider raising the rating of our work. Certainly, we are more than happy to address any further questions that you may have.

---

> > ### Comment · Reviewer_eKGX · 2023-11-22
> > **Reply**
> >
> > Thanks for the detailed responses. I have no further questions and I would like to maintain my current evaluation for this work.

---

> > > ### Author Response · Authors · 2023-11-22
> > >
> > > We thank the reviewer very much to check our response and for your effort into this review process.

---

### Official Review · Reviewer_dtbr · 2023-10-30

**Soundness:** 3 good
**Presentation:** 4 excellent
**Contribution:** 3 good
**Rating:** 5
**Confidence:** 4

**Summary:**

This paper adds the UCB (or LCB) term to online or offline POMDP. The modification compared with (Liu et al., 2022) is that 1) a hard threshold $p_\min$ adding to the selection of $\Theta$ and 2) the UCB term used in exploration.

**Strengths:**

- This paper is well-written and easy to follow
- The online algorithm can provide a `last-iteration' bound, and the bonus term can be efficiently calculated.

**Weaknesses:**

- Compare with (Liu et al., 2023), why during the exploration, the action trajectory needs to be sampled from $(\mathcal A \times \mathcal Q_h^{\mathcal A}) \cup Q_{h-1}^{\mathcal A}$, why we need the last term $Q_{h-1}^{\mathcal A}$? In addition, why we do not need to guarantee $\mathcal B_{k+1} \subseteq \mathcal B_{k}$ here?
- The sample complexity of the online algorithm is confusing, it looks more like a 'reward-free' paradigm, in which we are concerned about the *sample complexity* instead of the *regret*. When we talk about the 'last iteration regret', we should assume this regret bound works for all $k$ and the algorithm will not change regarding different $k$, which is not the case in this paper: we need to predefine a $K$ and get the sample complexity bound. Thus I think this result is kind of strange.
- I wonder what will the algorithm look like if we do not only use $V_{theta, b}$ but use the optimistic value function $V_{theta, R} + V_{theta, b}$ in the online algorithm, should the result be the same with (Liu et al., 2023)?
- Why the constrain related to $p_\min$ is easy to hold? Is there any theoretical justification? In other words, what will happen if that constrain does not hold for any $\theta$?

**Questions:**

Please see weakness

---

> ### Author Response · Authors · 2023-11-19
> **Response to Reviewer dtbr (Part 1)**
>
> We thank the reviewer for the helpful comments, and would like to provide the following responses. The paper has been revised accordingly to reflect these responses, where the changes are highlighted in the blue color. We hope that these discussions can clarify the reviewer's concerns, and we will be more than happy to answer any further questions you may have.
>
>
> **Q1 (a):** The action trajectory needs to be sampled from $(\mathcal{A}\times \mathcal{Q}\_{h}^A)\cup \mathcal{Q}\_{h-1}^A$, why we need the last term $\mathcal{Q}\_{h-1}^A$?
>
> **A1 (a):** We emphasize that the **key** difference between our work and [3] is that we try to estimate the feature $\bar{\psi}^*(\tau\_{h-1}^{k,h})$ of the **empirical sample** $\tau\_{h-1}^{k,h}$, while [3] analyzes the estimation error of $\bar{\hat{\psi}}^k(\tau\_{h-1})$ **in expectation**, i.e., $ E = \mathbb{E}\_{\tau\_{h-1}\sim \pi}[ \\|\bar{\psi}^*(\tau\_{h-1}) - \bar{\hat{\psi}}^k(\tau\_{h-1})\\|\_1]$, which can be upper bounded by $$ E\leq |\mathcal{A}|\mathbb{E}\_{\tau\_{h-2}, o\_{h-1} \sim \pi}\mathbb{E}\_{a\_{h-1}\sim U}[\mathtt{D}\_{\mathtt{TV}} (\mathbb{P}\_{\theta^*}(\cdot|\tau\_{h-2}, a\_{h-1},\mathbf{a}\_{h-1}^{\ell}) , \mathbb{P}\_{\hat{\theta}^k}(\cdot|\tau\_{h-2}, a\_{h-1}, \mathbf{a}\_{h-1}^{\ell} ))]$$ using **importance sampling**.  Therefore, it suffices to sample $\mathcal{A}\times \mathcal{Q}\_{h-1}^A$ to obtain an accurate estimate of $ \mathbb{P}\_{\hat{\theta}^k}(\cdot|\tau\_{h-2}, a\_{h-1}, \mathbf{a}\_{h-1}^{\ell} ))$ and ensure the estimation error $E$ shrinks. Such exploration is exactly what the algorithm has done at step $h-1$, i.e. uniformly sampling actions from $\mathcal{A}\times \mathcal{Q}\_{h-1}^A$ at step $h-1$. Therefore, [3] does not need to include additional $\mathcal{Q}\_{h-1}^A$ for exploration.
>
> However, our estimation error $\\|\bar{\psi}^*(\tau\_{h-1}^{k,h}) - \bar{\hat{\psi}}^k(\tau\_{h-1}^{k,h})\\|_1$ (Eq. (b) in Lemma 2, page 22) cannot be controlled in a similar way as [3]. This is the reason that we need to explore $\mathcal{Q}\_{h-1}^A$ explicitly. The importance of uniformly selecting actions from $(\mathcal{A}\times \mathcal{Q}\_{h}^A)\cup Q\_{h-1}^A$ has been explained in the second paragraph on page 6. We restate it as follows. First, the actions in $\mathcal{Q}\_{h-1}^A$ assist us to learn the prediction feature $x = \bar{\psi}^*(\tau\_{h-1}^{k,h})$ since the $\ell$-th coordinate of $x$ equals $\mathbb{P}\_{\theta^*}(\mathbf{o}\_{h-1}^{\ell} | \tau\_{h-1}^{k,h}, \mathbf{a}\_{h-1}^{\ell})$. Selecting $\mathbf{a}\_{h-1}^{\ell}\in\mathcal{Q}\_{h-1}^A$ improves the probability of observing $\mathbf{o}\_{h-1}^{\ell}$, which will provide more information to estimate $\bar{\psi}^*(\tau\_{h-1}^{k,h})$.  Second, the action sequence $(a_h,\mathbf{a}\_h^{\ell}) \in \mathcal{A}\times \mathcal{Q}\_h^A$ helps us to estimate $y=\mathbf{M}\_h^*(o_h,a_h)\bar{\psi}^*(\tau\_{h-1}^{k,h})$ because the $\ell$-th coordinate of $y$ represents $\mathbb{P}\_{\theta^*}( \mathbf{o}\_h^{\ell},o\_h |\tau\_{h-1}^{k,h}, a\_h, \mathbf{a}\_h^{\ell} )$. Similarly, selecting $(a_h,\mathbf{a}\_h^{\ell})\in\mathcal{A}\times\mathcal{Q}\_h^A$ might provide more data to estimate $\mathbf{M}\_h^*(o_h,a_h)\bar{\psi}^*(\tau\_{h-1}^{k,h})$. Thus, by uniformly selecting actions from $(\mathcal{A}\times \mathcal{Q}\_{h}^A)\cup \mathcal{Q}\_{h-1}^A$, we are able to collect the most informative samples for estimating the true model $\theta^*$.
>
> In summary, the novel design that uses empirical features $\bar{\hat{\psi}}(\tau\_{h-1}^{k,h})$ to quantify the uncertainty level requires a new exploration strategy, i.e. uniform sampling from $(\mathcal{A}\times \mathcal{Q}\_{h}^A)\cup \mathcal{Q}\_{h-1}^A$. In addition, from the analysis for $I_3$ in Lemma 2 on page 22, the final result depends on the size of the set of exploration action sequences. Thus, this new strategy does not affect the overall sample complexity, since $$\max\_h|\mathcal{A}\times \mathcal{Q}\_h^A| = |\mathcal{A}| Q\_A\leq \max\_h|(\mathcal{A}\times \mathcal{Q}\_h^A) \cup\mathcal{Q}\_{h-1}^A| \leq 2|\mathcal{A}|Q\_A.$$
>
> [1] Guo et al., Provably Efficient Representation Learning with Tractable Planning in Low-Rank POMDP.
>
> [2] Uehara et a., Representation Learning for Online and Offline RL in Low-rank MDPs.
>
>
> [3] Liu, et al., Optimistic MLE—A Generic Model-based Algorithm for Partially Observable Sequential Decision Making, 2022.
>
> [4] Zhan, et al., PAC Reinforcement Learning for Predictive State Representations

---

> ### Author Response · Authors · 2023-11-19
> **Response to Reviewer dtbr (Part 2)**
>
> **Q1 (b):** In addition, why we do not need to guarantee $\mathcal{B}\_{k+1}\subset \mathcal{B}\_k$ here?
>
> **A1 (b):**The reason that we do not need to guarantee $\mathcal{B}\_{k+1}\subset\mathcal{B}\_k$ is because of the bonus-based design and the stable MLE oracle. First, we note that for *reward-known* design, there is no need to guarantee $\mathcal{B}\_{k+1}\subset\mathcal{B}\_k$ even for confidence-set-based design (Eqn. (6) in [4]). For *reward-free* design in model-based algorithms, a key requirement is to output a model that is sufficiently accurate.  In bonus-based algorithms [1,2], we typically just select **an arbitrary** model from $\mathcal{B}\_k$, where the only property we need is that the model is close to the MLE solution (see Eqn. (2)), and the estimation error of the model is controlled by the empirical bonus term as
>
> $\mathtt{D}\_{\mathtt{TV}} (\mathbb{P}\_{\theta^*}^{\pi}, \mathbb{P}\_{\theta}^{\pi}) \leq V\_{\theta, b}^{\pi}$.
>
> However, in confidence-set-based algorithms [3], the estimated model is **optimized within** $\mathcal{B}\_k$ at each iteration, and there is no ``computable'' control on the accuracy of the models within a confidence set, since the accuracy of those models in $\mathcal{B}\_k$ depends on ground-truth bonus function, which is unknown. Under this situation, the only guarantee they have is that the summation of the estimation errors of any sequence of models selected from $\mathcal{B}_t, t=1,2,\ldots,k$ grows sublinearly in $k$. Mathematically, $\sum\_{t=1}^k\mathtt{D}\_{\mathtt{TV}}(\mathbb{P}\_{\theta_t}^{\pi}, \mathbb{P}\_{\theta^*}^{\pi}) \leq O(\sqrt{k})$ holds for any $\theta\_t\in\mathcal{B}_t$ and $k\in[K]$.   Therefore, to guarantee the accuracy of the final output model $\hat{\theta}$, the analysis in [3] requires $ \mathcal{B}\_{k+1}\subset\mathcal{B}\_k = \cap\_{k=1}^K\mathcal{B}\_k$, so that the accuracy of the model $\hat{\theta}\in\mathcal{B}_K$ in the final confidence set enjoys all the the guarantees in previous iterations, i.e. $K\mathtt{D}\_{\mathtt{TV}}(\mathbb{P}\_{\hat{\theta}}^{\pi}, \mathbb{P}\_{\theta^*}^{\pi}) \leq O(\sqrt{K})$.
>
>
>
>
> **Q2:** The sample complexity of the online algorithm is confusing, it looks more like a 'reward-free' paradigm, in which we are concerned about the sample complexity instead of the regret. When we talk about the 'last iteration regret', we should assume this regret bound works for all $k$ and the algorithm will not change regarding different
> $k$, which is not the case in this paper: we need to predefine a $K$ and get the sample complexity bound. Thus I think this result is kind of strange.
>
>
> **A2:** We would like to clarify that our performance metric is the **sample complexity**, **not the regret**.
> The last-iterate guarantee refers to the sample complexity needed to guarantee the performance of the final output policy and estimated model of our algorithm.
>
> In the current version of our algorithm, we use a predefined $K$ to control the maximum number of episodes run by the algorithm, which is just for ease of analysis. We can actually remove $K$ from line 2 in Alg. 1, since the algorithm is guaranteed to stop when line 10 is satisfied. For this case, $K$ only serves as a hyperparameter used to tune other parameters, such as $p_{\min} = O(\delta/(KH|\mathcal{O}|^H|\mathcal{A}|^H))$ and $\beta$ in Theorem 1, so that the algorithm is guaranteed to stop within $K$ episodes.
>
> We emphasize that assuming a prefixed $K$ is standard for RL algorithms when sample complexity is considered [1,2,3,4]. As a separate note, none of algorithm in [2,3,4] can relax the assumption of a prefixed $K$, while our algorithm can afford such flexibility.
>
>
> [1] Guo et al., Provably Efficient Representation Learning with Tractable Planning in Low-Rank POMDP.
>
> [2] Uehara et a., Representation Learning for Online and Offline RL in Low-rank MDPs.
>
>
> [3] Liu, et al., Optimistic MLE—A Generic Model-based Algorithm for Partially Observable Sequential Decision Making, 2022.
>
> [4] Zhan, et al., PAC Reinforcement Learning for Predictive State Representations.

---

> ### Author Response · Authors · 2023-11-19
> **Response to Reviewer dtbr (Part 3)**
>
> **Q3:** I wonder what will the algorithm look like if we do not only use $V_{\theta, b}^{\pi}$
>  but use the optimistic value function $V\_{\theta, R}^{\pi} + V\_{\theta, b}^{\pi}$
>  in the online algorithm, should the result be the same with (Liu et al., 2023)?
>
> **A3:** Great point! Using the optimistic value function $V_{\theta, R}^{\pi} + V_{\theta, b}^{\pi}$
>  in the online algorithm will also allow us to find a near-optimal policy at the last iteration. However, the drawback is that it no longer guarantees that the estimated model is accurate. This is because the optimistic policy $\pi^k = \arg\max_{\pi} V_{\theta, R}^{\pi} + V_{\theta, b}^{\pi}$ now depends the reward as well as the uncertainty level, which has less incentive to explore the trajectories generating low rewards. As a result, the accuracy of estimated model over those trajectories cannot be guaranteed.
>
>  In terms of the sample complexity, in the large $r$ regime when $r>Q_A^2H/\gamma^2$, our result $O(r^2dH^3|\mathcal{A}|^2Q_A^4/(\gamma^4\epsilon^2))$ achieves better order on $|\mathcal{A}|,H$ than the result $O(r^2|\mathcal{A}|^5Q_A^3H^4/(\gamma^4\epsilon^2) )$ in Liu et al. (2023), but has an additional factor $d$. In the low rank regime where $r<Q_A^2H/\gamma^2$, our result $O(rdH^4|\mathcal{A}|^2Q_A^6/(\gamma^6\epsilon^2))$ has better order on $r, |\mathcal{A}|$.
>
> **Q4:** Why the constrain related to $p_{\min}$
> is easy to hold? Is there any theoretical justification? In other words, what will happen if that constraint does not hold for any $\theta$?
>
> **A4:** We prove that with high probability, the true model $\theta^*$ always satisfies this constraint (See Proposition 7 on page 19). Moreover, a simple model $\theta_0$ that makes trajectories uniformly distributed also satisfies this constraint. Specifically, let $\mathbb{P}_{\theta_0}(o_H,\ldots,o_1|a_1,\ldots,a_H) = 1/|\mathcal{O}|^H$. Then $\theta_0$ clearly satisfies the constraint.
>
>
> In practice, we can set $p\_{\min} = (\delta/(|\mathcal{O}|^H|\mathcal{A}|^HKH))^c$ for any absolute constant $c>1$. Clearly, such a threshold is approximately zero (note that $\delta$ is typically chosen to be a small constant, and all variables in the denominators are large in practical systems). This means that as long as we guarantee $\mathbb{P}\_{\theta}^{\pi}(\tau_h)$ is nonzero, the constraint $\mathbb{P}\_{\theta}^{\pi}(\tau_h)\ge p\_{\min}$ is satisfied. Hence, in practice, we can simply parameterize $\mathbb{P}\_{\theta}$ to have nonzero elements to satisfy the constraint, which is needed anyway to prevent infinite value of $\log \mathbb{P}\_{\theta}^{\pi}(\tau_H)$ in the objective function of Eq (2). Then Eq (2) will become a standard unconstrained optimization to maximize the likelihood value $\sum\_{(\tau_H,\pi)\in\mathcal{D}}\log \mathbb{P}\_{\theta}^{\pi}(\tau_H)$ (MLE problem). Such an unconstrained problem can be solved efficiently through standard
> gradient descent based approaches.
>
> -----
>
> We thank the reviewer again for your feedback and suggestions.  We also would like to bring to your attention that your individual ratings on Soundness (3 good), Presentation (4 excellent) and Contribution (3 good) don't seem to be quite consistent with your final rating 5 of our paper. If our response resolves your concerns, could you please kindly consider raising your rating of the paper, which will also make it aligned with your individual ratings? Of course, if you have any further questions, we would be very happy to address them.

---

> ### Author Response · Authors · 2023-11-22
>
> Dear Reviewer dtbr,
>
> As the author-reviewer discussion period will end soon, we will appreciate it if you could check our response to your review comments. This way, if you have further questions and comments, we can still reply before the author-reviewer discussion period ends.
>
> We also would like to bring to your attention that your individual ratings on Soundness (3 good), Presentation (4 excellent) and Contribution (3 good) don't seem to be quite consistent with your final rating 5 of our paper. If our response resolves your concerns, we kindly ask you to consider raising the rating of our work. Thank you very much for your time and efforts!

---

### Official Review · Reviewer_4GGq · 2023-10-30

**Soundness:** 3 good
**Presentation:** 3 good
**Contribution:** 3 good
**Rating:** 6
**Confidence:** 3

**Summary:**

This paper studies learning predictive state representations (PSRs) using UCB-type approaches. The proposed PSR-UCB algorithm is based on an upper bound for the total variance distance between estimated and true models. The algorithm requires supervised learning oracles based on which it is computationally efficient and has a last iterate guarantee. An offline variant PSR-LCB is also proposed.

**Strengths:**

- This paper is in general well-written and smooth to follow, with the exception of some proofs in the supplementary material.
- The originality of the paper is high as it proposes a novel UCB-based algorithm for PSRs. Related works are covered in detail.
- The theoretical proofs seem to be rigorous.
- The proposed UCB-based algorithm enjoys strong theoretical guarantees and might be of interest to researchers.

**Weaknesses:**

My main concerns with this paper are as follows.
- First, the theoretical results crucially depend on the MLE oracle, which is called $H$ times during the implementation. However, the paper falls short in discussing the difficulty of solving the MLE estimation. At least, it would be better if, for some special class of $\theta$'s, the authors could show how the estimation could be implemented and what's computationally costly.
- This paper is purely theoretical and lacks experiments even on synthetic data. I would appreciate simulation results validating the sample complexity presented in the theory.

**Questions:**

I've detailed my primary concerns and suggestions above. However, I have a couple of specific questions regarding certain aspects:

1. On Page 6, just below equation (2), could the authors elaborate on why learning \$\bar{\psi}^*(\tau_h^{k, h+1}) $ might be harmful when $\mathbb{P}_{\theta^*}^{\pi^{k-1}}(\tau_h^{k, h+1}) $ is very small?

2. In Algorithm 1, the UCB-approach seems to serve as a pure exploration mechanism aimed at enhancing the estimation of $\theta $. Yet, in Algorithm 2, the upper confidence bound for $ D_{TV} $ is directly employed to construct a lower confidence bound for $V_{\theta^*, R}^\pi $, which the output policy then optimizes. Could you shed light on what drives this distinction between the designs of the two algorithms? Or, is there an underlying equivalence between them?

---

> ### Author Response · Authors · 2023-11-19
> **Response to Reviewer 4GGq (Part 1)**
>
> We thank the reviewer for the helpful comments, and would like to provide the following responses. The paper has been revised accordingly to reflect these responses, where the changes are highlighted in the blue color. We hope that these discussions can clarify the reviewer's concerns, and we will be more than happy to answer any further questions you may have.
>
>
>
> **Q1:** Solving MLE oracle. It would be better if, for some special class of $\theta$'s, the authors could show how the estimation could be implemented and what's computationally costly.
>
> **A1:** We note that Eq (2) can be solved as an unconstrained optimization problem in practice as we explain below.
>
> In the constraint, the threshold $p_{\min}$ can be set as small as $O((\frac{\delta}{KHO^HA^H})^c)$, where $c\geq 1$ is an arbitrary positive constant (see Theorem 1). Clearly, such a threshold is approximately zero (note that $\delta$ is typically chosen to be a small constant, and all variables in the denominator are large in practical systems). This means that as long as we guarantee $\mathbb{P}\_{\theta}^{\pi}(\tau_h)$ is nonzero, the constraint $\mathbb{P}\_{\theta}^{\pi}(\tau_h)\ge p\_{\min}$ is satisfied. Hence, in practice, we can simply parameterize $\mathbb{P}\_{\theta}$ to have nonzero elements to satisfy the constraint, which is needed anyway to prevent infinite value of $\log \mathbb{P}\_{\theta}^{\pi}(\tau_H)$ in the objective function of Eq (2). Then Eq (2) will become a standard unconstrained optimization to maximize the likelihood value $\sum\_{(\tau_H,\pi)\in\mathcal{D}}\log \mathbb{P}\_{\theta}^{\pi}(\tau_H)$ (MLE problem). The computational cost is thus equivalent to the cost of performing supervised learning with loss function $\sum\_{(\tau_H,\pi)\in\mathcal{D}}\log \mathbb{P}\_{\theta}^{\pi}(\tau_H)$, which can be solved via methods such as stochastic gradient descent.
>
>
> **Q2:** This paper is purely theoretical and lacks experiments even on synthetic data. I would appreciate simulation results validating the sample complexity presented in the theory.
>
> **A2:** Thanks for the suggestion.  We are actively exploring some synthetic settings to validate our algorithm and theoretical results. We hope to produce some results soon.
>
>
> **Q3:** Could the authors elaborate on why learning $\bar{\psi}^*(\tau_h^{k,h+1})$ might be harmful when $\mathbb{P}\_{\theta^*}^{\pi^{k-1}}(\tau_h^{k,h+1})$ is very small?
>
> **A3:** We apologize for the confusion here. This statement should be interpreted as ``since`` $\mathbb{P}\_{\theta^*}^{\pi^{k-1}}(\tau_h^{k,h+1})$ ``is not small with high probability, it is harmful to have an estimated model`` $\hat{\theta}$ ``where `` $\mathbb{P}\_{\hat{\theta}}^{\pi^{k-1}}(\tau_h^{k,h+1})$ ``is very small.`` In Proposition 7 (page 19), we prove that $\mathbb{P}\_{\theta^*}^{\pi^{k-1}}(\tau_h^{k,h+1})$ must be greater than $p\_{\min}$ with high probability. In other words, it is highly unlikely that $\mathbb{P}\_{\theta^*}^{\pi^{k-1}}(\tau_h^{k,h+1})$ is very small. Therefore, under this high probability event, if a model $\hat{\theta}$ assigns a small probability on $\tau_h^{k,h+1}$, then this model is unlikely to be close to the true model. In other words, using $\hat{\theta}$ for future planning is harmful because it differs from the true model $\theta^*$ too much.

---

> ### Author Response · Authors · 2023-11-19
> **Response to Reviewer 4GGq (Part 2)**
>
> **Q4:** In Algorithm 1, the UCB-approach seems to serve as a pure exploration mechanism aimed at enhancing the estimation of $\theta$. Yet, in Algorithm 2, the upper confidence bound for
> $D_{TV}$ is directly employed to construct a lower confidence bound for $V_{\theta^*, R}^{\pi}$, which the output policy then optimizes. Could you shed light on what drives this distinction between the designs of the two algorithms? Or, is there an underlying equivalence between them?
>
> **A4:** We note that the ultimate goal of both algorithms is to output a near-optimal policy.
>
> In Algorithm 1, we choose a reward-free design for **online** exploration, since it can provide stronger performance guarantees including a near-accurate estimated model. The bonus value $V_{\hat{\theta},\hat{b}}^{\pi}$ also serves as an upper confidence bound (UCB) for $\mathtt{D}\_{\mathtt{TV}}(\mathbb{P}\_{\theta^*}^{\pi},\mathbb{P}\_{\hat{\theta}}^{\pi})$. By deploying the policy that maximizes the UCB of $\mathtt{D}\_{\mathtt{TV}}$, we are collecting data where the estimated model $\hat{\theta}$ has the most uncertainty. Thus, the accuracy of the estimated model can be enhanced.
>
> In Algorithm 2, it is hard to estimate the model accurately since the **offline** dataset has partial coverage of the target policy, or the optimal policy. Following the commonly adopted pessimism principle in offline RL [1], we directly construct a lower confidence bound for $V\_{\theta^*,R}^{\pi}$ in light of the inequality $V\_{\theta^*,R}^{\pi}\geq V\_{\hat{\theta},R}^{\pi} - \mathtt{D}\_{\mathtt{TV}}(\mathbb{P}\_{\theta^*}^{\pi},\mathbb{P}\_{\hat{\theta}}^{\pi})$ and an UCB of $\mathtt{D}\_{\mathtt{TV}}(\mathbb{P}\_{\theta^*}^{\pi},\mathbb{P}\_{\hat{\theta}}^{\pi})$.
>
> In summary, both Alg.1 and Alg. 2 utilize the UCB of $\mathtt{D}\_{\mathtt{TV}}(\mathbb{P}\_{\theta^*}^{\pi},\mathbb{P}\_{\hat{\theta}}^{\pi})$, which is $V\_{\hat{\theta},\hat{b}}^{\pi}$. But due to different purposes, Alg. 1 maximizes $V\_{\hat{\theta},\hat{b}}^{\pi}$ to enhance the accuracy of the estimated model under reward-free design, and Alg. 2 maximizes the LCB $V\_{\hat{\theta},R}^{\pi} -V\_{\hat{\theta},\hat{b}}^{\pi}$ under the pessimism principle to handle partial coverage.
>
> [1] Uehara et a., Representation Learning for Online and Offline RL in Low-rank MDPs.
>
> -----
>
> We thank the reviewer again for your feedback and suggestions. Hopefully, our responses have addressed your main concerns. If so, we kindly ask that you consider raising the score of your evaluation. If you still have remaining questions, please let us know, and we will be very happy to provide further clarification.

---

> > ### Comment · Reviewer_4GGq · 2023-11-22
> >
> > I appreciate the authors' detailed responses to my questions. I'll keep my current rating as there are still no simulation results.

---

> > > ### Author Response · Authors · 2023-11-22
> > >
> > > We thank the reviewer for checking our response. We understand the reviewer's decision. We are working on the numerical experiments, which take some time. Since most of the studies along this line of research on theoretical POMDP don't have numerical experiments, we need to code from scratch. We are doing our best.

---

### Official Review · Reviewer_YPuy · 2023-10-31

**Soundness:** 3 good
**Presentation:** 2 fair
**Contribution:** 3 good
**Rating:** 8
**Confidence:** 3

**Summary:**

This paper presents an UCB-based algorithm to learn parameters in POMDP using predictive state representations (PSRs) to simplify the number of potential historical and future trajectories. PSRs assume a smaller subset of possible trajectories with a low-rank basis. Therefore, PSRs can efficiently reduce the dimensionality required to represent all the historical or future trajectories.

Using PSRs, the authors show how to use maximum likelihood method to find the most likely PSR parameters, and then use an additional bonus function to quantify the uncertainty of the learned PSR parameters. The algorithm leads to no-regret guarantees with a sample complexity bound induced from the regret analysis. The UCB construction is quite different from other UCB algorithms that I have seen in the literature of multi-armed bandits.

**Strengths:**

**Solid theoretical analysis**: I checked the main paper but didn’t check the appendix. The main paper and proof sketches are theoretically sound to me.

**UCB algorithm for learning parameters in POMDP (reduction in learning space)**: careful and detailed analysis of a complex problem of learning parameters of POMDP from historical trajectories. Previous online learning in sequential problems usually learns the tabular-from parameters directly with easily constructed UCBs. This paper analyzes UCBs in predictive state representations (PSRs), which can reduce the number of potential historical trajectories in POMDP to maintain UCBs.

**New analysis**: the analysis is indeed new to me. I appreciate the novelty in theoretical contributions.

**Weaknesses:**

**Computation**: the computation cost is still expensive with a dependency on how good the PSR is (the size of $Q_h$). Could you elaborate more on the overall computation complexity and point out what is the bottleneck?

**Reward-free claim**: the theoretical results and the analysis are reward-free by design. The major learning problem is on learning the transition probability using PSR. The reward function is bounded within [0,1] so I am not surprised that the algorithm and results are reward-free.

**Overly complicated notations**: I understand this is a difficult problem with many notations needed for the analysis. But the presentation is very difficult to follow and its clarity can be significantly improved. I believe there is a better way to present and help readers understand the analysis better.

**Sample complexity**: I thought a major advantage of PSRs is to discard the dependency on the number of all possible history with size $O(|A|^H |O|^H)$. However, this term still appears in your sample complexity with a dependency on $O(\epsilon^{-2})$. The dependency on the action and observation space is hidden in $\epsilon = p_{min}$ where $p_{min} = O(\delta / {|O|^H |A|^H})$. Could you explain how your algorithm improves and compares with the online learning algorithms without using PSRs?

**Offline learning**: the offline learning part directly follows the same proof as the online learning part. Could you elaborate if there is any new insights brought by the offline learning analysis?

**Questions:**

**Questions**: please answer my questions mentioned in the weaknesses section. I am happy to raise the score if some of my questions are clarified.


**Typos or confusions**:
- Equation 1: what does $\pi(\omega_h)$ mean? Your definition of $\pi$ measures the probability of choosing a sequence of actions conditioned on a sequence of observations. $\pi(\omega_h)$ only specifies the future trajectory after time $h$ but does not specify the history prior to $h$. Please clarify.
- After finishing reading the paper, I still don’t see what does the “physical meaning” mean in Section 4.1.
- Line 4 in Algorithm 1: notations of $\omega_{h-1}^{k,h}$ and $\tau_{h-1}^{k,h}$ are confusing. You have a different use of superscript on $\omega$ and $\tau$ defined previously. The following discussion in Page 6 is helpful though.
- Equation 3: $\tau_h \in D_h^k$, abused notation which collides with the definition of the bonus function with a different $\tau_h$ involved. Similarly, there is a typo in the following sentence: $\{ \hat{\phi}^k(\tau_h) \}_{\tau \in D^k_h}$ where $tau_h$ should be $\tau$ unless I misunderstood.

---

> ### Author Response · Authors · 2023-11-19
> **Response to Reviewer YPuy (Part 1)**
>
> We thank the reviewer for the helpful comments, and would like to provide the following responses. The paper has been revised accordingly to reflect these responses, where the changes are highlighted in the blue color. We hope that these discussions can clarify the reviewer's concerns, and we will be more than happy to answer any further questions you may have.
>
>
> **Q1:** The computation cost is still expensive with a dependency on how good the PSR is (the size of $Q_h$). Could you elaborate more on the overall computation complexity and point out what is the bottleneck?
>
> **A1:** The computational cost of our algorithm can be summarized as follows. Our algorithm requires calling MLE computation oracle once in every iteration (line 7 in Alg. 1). Calculating the bonus term (line 8 in Alg. 1) takes $O(Hkd^2)$ flops in iteration $k$ with the Sherman-Morrison formula. Finally, a quasipolynomial time complexity $H(|O||A|)^{C\log(rH/\epsilon)/\gamma^4}$ has been shown in [1] for planning (line 9 in Alg. 1). Therefore, the overall computation complexity is $O(HK^2d^2 + KH(|O||A|)^{C\log(rH/\epsilon)/\gamma^4})$ if $K$ MLE oracles are available. The computational bottleneck is therefore determined by the MLE oracle and the planning procedure.
>
> The above MLE oracle and the planning procedure are what our algorithm shares in common with the existing confidence-set-based POMDP approaches. Beyond those, the existing confidence-set-based approaches also rely on an optimistic planning procedure that iterates over a large function class $\mathcal{B}_k$, which may require combinatorial search, leading to additional exponential computational complexity. In contrast, our bonus-based approach avoids such optimistic planning procedure, and thus is more advantageous for practical implementation.
>
> [1] Golowich, et al., Planning in Observable POMDPs in Quasipolynomial Time.
>
> [2] Liu, et al., Optimistic MLE—A Generic Model-based Algorithm for Partially Observable Sequential Decision Making, 2022.
>
>
> **Q2:** I am not surprised that the algorithm and results are reward-free.
>
> **A2:** Our algorithm and results are indeed reward-free by design. The main advantage of such a reward-free design over reward-based design is that it guarantees to output a near-accurate model for model-based algorithms. We can also modify Line 9 in Alg. 1 to let $\pi^k = \arg\max\_{\pi} V\_{\hat{\theta}^k, R}^{\pi} + V\_{\hat{\theta}^k,\hat{b}^k}^{\pi} $, so that the policy depends on $\hat{b}^k$ as well as the reward information $R$. We can show that such a design can also identify a near-optimal policy under the same sample complexity: The current proof stays almost the same, except that we remove the claim $\forall \pi,  \mathtt{D}\_{\mathtt{TV}}(\mathbb{P}\_{\theta^*}^{\pi}, \mathbb{P}\_{\hat{\theta}^{\epsilon}}^{\pi}) \leq \epsilon$.  However, it lacks the guarantee for the estimated model. This is because the optimistic policy $\pi^k$ now depends on the reward as well as the uncertainty level, which has less incentive to explore the trajectories generating low rewards. As a result, the accuracy of estimated model over those trajectories cannot be guaranteed.
>
> **Q3:** Complicated notations.
>
> **A3:** Thanks for the suggestion. We will try our best to simplify the notations and make our presentation as clear as possible in the revision.
>
> **Q4:** I thought a major advantage of PSRs is to discard the dependency on the number of all possible history with size
> $O(|A|^H|O|^H)$. However, this term still appears in your sample complexity with a dependency on $O(\epsilon^{-2})$. The dependency on the action and observation space is hidden in $\epsilon = p_{\min}$ where
> $p_{\min} = O(\delta/|O|^H|A|^H)$. Could you explain how your algorithm improves and compares with the online learning algorithms without using PSRs?
>
> **A4:**
> We would like to clarify that the term $p_{\min}$ only appears in the **logarithm factor**, as suggested by $\beta = \log|\Theta_{\varepsilon}|$, and $\varepsilon = p_{\min}/(KH)$. In fact, when we plug in $p_{\min} =  O(\delta/|O|^H|A|^H)$, we have $\beta = O(H^2r^2|O||A|\log (KH|O||A|))$ (See Proposition 3).
> Therefore, compared with previous PSR work [2] and other online learning algorithms without using PSRs [3], our algorithm **improves the computation tractability using bonus-based design**  (see detailed explanation in the second paragraph of A1). Meanwhile, our sample complexity depends polynomially on $|\mathcal{O}|, |\mathcal{A}|, H$, and not on $|\mathcal{O}|^H|\mathcal{A}|^H$, which is the same as previous works [1,2,3].
>
> [1] Golowich, et al., Planning in Observable POMDPs in Quasipolynomial Time.
>
> [2] Liu, et al., Optimistic MLE—A Generic Model-based Algorithm for Partially Observable Sequential Decision Making, 2022.
>
>
> [3] Liu, et al., When Is Partially Observable Reinforcement Learning Not Scary?

---

> ### Author Response · Authors · 2023-11-19
> **Response to Reviewer YPuy (Part 2)**
>
> **Q5:** Could you elaborate if there is any new insights brought by the offline learning analysis?
>
> **A5:** One fundamental difference between online and offline learning is that, during online learning, the learner is able to decide the exploration policies adaptively, while in the offline setting, the behavior policy $\pi^b$ adopted to collect the dataset cannot be controlled by the learner. As a result, it leads to two major differences in the offline learning analysis.
>
> First, offline learning requires some **coverage assumption** on the behavior policy adopted to collect the dataset in order to ensure that the information contained in the dataset is sufficient to recover a near-optimal policy. Specifically, our analysis relies on the newly proposed coverage coefficient (Sec. 5.2) for PSRs. Unlike the analysis for online learning, where we use elliptical potential lemma to show the closeness between the output policy and the optimal policy, this coverage coefficient helps us to establish the relationship between the optimal policy and the behavior policy (See Appendix D.4). Second, the action sequence chosen by the behavior policy $\pi^b$ may also be different from the online exploration policy. Specifically, online exploration policies have the freedom to uniformly explore core action sequence, while $\pi^b$ does not necessarily do so. To handle such difference, **importance sampling** is used in the offline learning analysis to control the estimation error $\\|\bar{\psi}^*(\tau_h) - \bar{\hat{\psi}}(\tau_h)\\|_1$ by $\frac{1}{\iota}\mathtt{D}\_{\mathtt{TV}}(\mathbb{P}\_{\theta^*}^{\pi^b}, \mathbb{P}\_{\hat{\theta}}^{\pi^b})$ (See Appendix D.2).
>
> Those differences in the analysis highlight **the importance of coverage assumption and core actions sequence in offline learning for PSRs**. As offline PSR learning is still an active research area, an interesting direction is to weaken the coverage assumption such that it is more suitable for the dataset in real applications.
>
> We then address the typos and confusions as follows.
>
> **Q6:** Equation 1: what does $\pi(\omega_h)$
>  mean? Your definition of $\pi$
>  measures the probability of choosing a sequence of actions conditioned on a sequence of observations. $\pi(\omega_h)$
>  only specifies the future trajectory after time $h$
>  but does not specify the history prior to $h$
> . Please clarify.
>
> **A6:** We apologize for the confusion. The maximization in Eqn. (1) should also be over all possible $\tau_h$. In other words, $\max_{\pi} \sum_{\omega_h}\pi(\omega_h)|\mathbf{m}(\omega_h)^{\top}x| = \max_{\pi}\max_{\tau_h} \sum_{\omega_h} \pi(\omega_h|\tau_h)|\mathbf{m}(\omega_h)^{\top}x|$, where the optimization over $\pi$ is over all possible distributions on future action sequences $a_{h+1},\ldots,a_H$. We have clarified this in the revision.
>
> **Q7:** After finishing reading the paper, I still don’t see what does the “physical meaning” mean in Section 4.1.
>
> **A7:** By ``physical meaning``, we are referring to the sentence ``each coordinate of a prediction feature of `` $\tau_h$ ``represents the probability of visiting a core test conditioned on ``$\tau_h$ ``and taking the corresponding core action sequence`` right after the term ''physical meaning'' in Sec 4.1. We have revised this statement to make it clearer in the revision.
>
>
> **Q8:** Line 4 in Algorithm 1: notations of $\omega_{h-1}^{k,h}$ and $\tau_{h-1}^{k,h}$ are confusing.
>
> **A8:** The superscripts $k,h$ represent the index of episodes during online exploration. The superscripts $o,a$ indicate the state sequence and the action sequence, respectively. We have updated the presentation of Sec. 4.1 (change the order of algorithm description and the pseudo-code) to avoid potential confusion.
>
>
> **Q9:** Equation 3: $\tau_h\in\mathcal{D}\_h^k$
> , abused notation which collides with the definition of the bonus function with a different $\tau_h$
>  involved. Similarly, there is a typo in the following sentence: $\hat{\psi}^k(\tau_h)\_{\tau\in\mathcal{D}_h^k}$.
>
> **A9:** We apologize for the typo. Note that each element in $\mathcal{D}\_h^k$ has the form $(\tau_H^{t,h+1}, \nu_{h+1}(\pi^{t-1}, \mathtt{u}\_{Q_{h}^{\exp}}))$. Thus, $\tau_h\in\mathcal{D}\_h^k$ means that we are extracting all the history before the step $h+1$ in the dataset $\mathcal{D}\_h^k$, i.e. $\tau_h \in\\{\tau_h^{1,h+1}, \tau_h^{2,h+1},\ldots,\tau\_h^{k,h+1}\\} $. We have corrected it as $\hat{\psi}^k(\tau_h)\_{\tau_h\in\mathcal{D}\_h^k}$ in the revision.
> To avoid the notation collision, we now use $\tau_h' \in\mathcal{D}_h^k$ to make it different from that used in the bonus function.
>
> -----
>
> We thank the reviewer again for your feedback and suggestions. Hopefully, our responses have addressed your main concerns. If so, we kindly ask that you consider raising the score of your evaluation. If you still have remaining questions, please let us know, and we will be very happy to provide further clarification.

---

> > ### Comment · Reviewer_YPuy · 2023-11-22
> > **Response to authors**
> >
> > I would like to thank the authors for clarifying my questions and for revising the paper. I also agree with other reviewers that there is no simulation or evaluation and it is certainly a weakness of the paper, but I would still like to increase my score based on the theoretical contribution of the paper.

---

> > > ### Author Response · Authors · 2023-11-22
> > >
> > > We thank the reviewer very much for checking our response and kindly increasing the rating. Many thanks for your effort into this review process.

---

### Meta-Review · Area_Chair_FVm4 · 2023-12-11

**Metareview:**

In this paper, the authors introduced UCB exploration for predictive state representation. This is theory-oriented paper. With MLE oracle and the planning oracle, the authors provided computation and sample complexity for general sequential decision-making problem, which includes Markov decision processes (MDPs) and partially observable MDPs (POMDPs).

**Justification For Why Not Higher Score:**

There are several issues should be addressed:

1, Polynomial computational-complexity relies on computation oracle, which should be clearly stated as "oracle-efficient".

2, Comparing to Optimistic-MLE, the UCB planning oracle is required additionally.

3, Comparing to Golowich, et al, in which no oracle is required, the current claim on computation complexity is unfair.

Considering this, the necessity of UCB upon optimistic MLE should be carefully discussed: the proposed algorithm requires both optimistic MLE oracle and UCB planner oracle, while optimistic MLE only requires MLE estimator.

**Justification For Why Not Lower Score:**

The major confusing part of the paper lies in  1) , the motivation of using UCB upon optimistic MLE, and 2) the claim in "computation-efficient", other than that, the paper analysis is solid.

---

### Decision · Program_Chairs · 2024-01-16

Accept (poster)